# Large Language Models are Demonstration Pre-Selectors for Themselves

## Abstract

In-context learning with large language models (LLMs) delivers strong few-shot performance by choosing few-shot demonstrations from the entire training dataset. However, previous few-shot in-context learning methods, which calculate similarity scores for choosing demonstrations, incur high computational costs by repeatedly retrieving large-scale datasets for each query. This is due to their failure to recognize that not all demonstrations are equally informative, and many less informative demonstrations can be inferred from a core set of highly informative ones. To this end, we propose FEEDER (FEw yet Essential Demonstration prE-selectoR), a novel *pre-selection* framework that identifies a core subset of demonstrations containing the most informative examples. This subset, referred to as the FEEDER set ($\mathcal{D}_{\text{FEEDER}}$), consists of demonstrations that capture both the "sufficiency" and "necessity" information to infer the entire dataset. Notice that $\mathcal{D}_{\text{FEEDER}}$ is selected before the few-shot in-context learning, enabling more efficient few-shot demonstrations choosing in a smaller set ($\mathcal{D}_{\text{FEEDER}}$). To identify $\mathcal{D}_{\text{FEEDER}}$, we propose a novel effective tree based algorithm. Once selected, it can replace the original dataset, leading to improved efficiency and prediction accuracy in few-shot in-context learning. Additionally, $\mathcal{D}_{\text{FEEDER}}$ also benefit fine-tuning LLMs, we propose a bi-level optimization method enabling more efficient training without sacrificing performance when datasets become smaller. Our experiments are on 6 text classification datasets, 1 reasoning dataset, and 1 semantic-parsing dataset, across 8 LLMs (ranging from 335M to 8B parameters), demonstrate that: (i) In few-shot inference, FEEDER achieves superior (or comparable) performance while utilizing only half the input training data. (ii) In fine-tuning, FEEDER significantly boosts the performance of LLMs.

## 1 Introduction

Large language models (LLMs), e.g., GPT (Brown et al., 2020), Gemma (Team et al., 2024), and Llama (Touvron et al., 2023), have demonstrated impressive performance across a wide range of tasks by employing few-shot inference, often referred as in-context learning (Brown et al., 2020; Dong et al., 2022). This approach avoids the computational expense associated with fine-tuning LLMs. Here, the core challenge is how to select the most effective demonstrations from a large training set. Early methods (Qiu et al., 2022; Liu et al., 2021; Rubin et al., 2021; Wang et al., 2022) primarily selected demonstrations based on relevance, using similarity scores between each demonstration and the input question. Recent studies (Levy et al., 2022; Köksal et al., 2022; Zhou et al., 2023) have also incorporated diversity, uncertainty, or clustering based metrics along with similarity, acknowledging that measuring each example in isolation is inefficient. This is because previous methods fail to recognize that not all demonstrations contribute equally across different LLMs and domains. A small set of highly informative examples can often capture enough information to infer many of the less informative ones. By not focusing on this core set, prior approaches end up processing unnecessary data, resulting in higher computational costs and lower efficiency in few-shot inference.

Our main idea is to identify the most informative subset that can effectively replace the entire original dataset, which is grounded in the consistency of LLMs. As observed by (Jang & Lukasiewicz, 2023), LLMs demonstrate strong performance in tasks such as transitive inference. On this promise, we propose a demonstration *pre-selector* named FEEDER (FEw yet Essential Demonstration prE-selectoR). Concretely, our FEEDER, served as a core subset selector over the training dataset, examines input demonstrations in terms of "sufficiency" and "necessity". Sufficiency investigates whether prompting

Figure 1: Overview of FEEDER that operates effectively within both in-context learning and fine-tuning settings. In the in-context learning setting, depicted in (a), we first *pre*-select a core set termed FEEDER from the training dataset, and then incorporate existing demonstration retrievers to get samples regarding specific test input. This selected set is characterized by its sufficiency and necessity conditioned on the frozen LLM. In the fine-tuning setting, shown in (b), FEEDER allows the LLM to be tuned on the fixed subset, and this subset is intentionally selected to be a faithful representation of the training dataset, with the dual objectives of maintaining data quality and minimizing computational expenses. The above two processes can be encapsulated into a bi-level optimization framework, allowing for iterative refinement of both the selected FEEDER and the fine-tuned LLM.

a demonstration enhances LLM performance on domain-specific tasks, while necessity assesses whether a newly considered demonstration offers redundant information compared to those already included. The resulting sets of selected demonstrations, identified as sufficient and necessary, form what we term FEEDER sets.

To efficiently select a FEEDER set from the training dataset, the exhaustive enumeration and evaluation of all possible subsets is impractical. Therefore, we devise a tree based approximation algorithm to examine whether each demonstration is sufficient and necessary to represent other demonstrations. Our identification of FEEDER sets can be characterized as a core-set selection approach, producing a subset of training instances that is highly informative for downstream tasks, including in-context learning and fine-tuning. In the in-context learning setting, our FEEDER can also benefit from the use of various demonstration selectors, by utilizing a pre-selected FEEDER set as the retrieval pool instead of the entire training dataset to generate n-shot demonstrations. Additionally, we demonstrate that a FEEDER set also can enhance the fine-tuning process. Specifically, we show that fine-tuning the performance of LLMs with a single epoch on the pre-selected subset proves to be more effective than doing so on the entire training dataset. The above observations collectively give rise to a novel bi-level framework, wherein we formulate the pre-selection of FEEDER sets and the fine-tuning of LLMs on the pre-selected subset as a unified bi-level optimization problem. It comprises an outer level for extracting a FEEDER set using a frozen LLM and an inner level for fine-tuning the LLM with the fixed FEEDER set. This iterative process involves utilizing the tuned LLM for the new FEEDER selection in the subsequent iteration.

Our empirical evaluations span 6 text classification datasets, 6 LLM bases ranging from 335M to 7B, and 6 existing demonstration selectors (e.g., random, similarity-based, and diversity-based). Results consistently demonstrate that efficiency and effectiveness of FEEDER: In terms of efficiency, our pre-selected FEEDER saves nearly half of the data size. In terms of effectiveness, using FEEDER rather than the full training dataset, consistently yields superior (or comparable) performance in the few-shot inference. Moreover, results also indicate that fine-tuning LLMs on FEEDER consistently leads to significant improvements compared to fine-tuning on the entire training dataset. The evaluation of FEEDER is further expanded to 1 reasoning task and 1 semantic-parsing task, providing consistent results with trends observed in the text classification task.

## 2 A DATA-CENTRIC PERSPECTIVE FROM IN-CONTEXT LEARNING TO FINE-TUNING

We begin by delineating two distinct contexts where FEEDER operates: in-context learning setting and fine-tuning setting. Throughout this paper, we approach both scenarios from a data-centric perspective (Strickland, 2022), emphasizing the significance of *data quality* over *data quantity*.

In the in-context learning setting, we are given a training dataset $\mathcal{D}_{\text{TRAIN}} = \{(\boldsymbol{x}_n, \boldsymbol{y}_n)\}_{n=1}^N$ consisting of pairs of input data (e.g., questions) and output labels (e.g., answers). We are also given a test dataset $\mathcal{D}_{\text{TEST}} = \{(\boldsymbol{x}_m, \boldsymbol{y}_m)\}_{m=1}^M$, where we assume that $\mathcal{D}_{\text{TRAIN}}$ share the same support set (Yosida, 2012) with $\mathcal{D}_{\text{TEST}}$. Our goal is to develop a demonstration selector that extracts n-shot demonstrations from the training dataset, denoted as $\mathcal{D}_{\text{DEMO}} \subseteq \mathcal{D}_{\text{TRAIN}}$. We use $\Psi_{\text{LLM}} : \mathbb{X} \times \mathbb{D} \to \mathbb{Y}$ to represent a

LLM using selected demonstrations as the context. Here, $\boldsymbol{x}. \in \mathbb{X}$ is an input text, $\boldsymbol{y}. \in \mathbb{Y}$ is the corresponding output, and $(\boldsymbol{x}., \boldsymbol{y}.) \in \mathbb{D}$ is one demonstration. Formally, our objective is to minimize:

$$\mathcal{L}(\mathcal{D}_{\text{DEMO}}, \mathcal{D}_{\text{TEST}}) = \sum_{(\boldsymbol{x}_m, \boldsymbol{y}_m) \in \mathcal{D}_{\text{TEST}}} \ell\Big(\Psi^*_{\text{LLM}}(\boldsymbol{x}_m, \mathcal{D}_{\text{DEMO}}), \boldsymbol{y}_m\Big), \tag{1}$$

where $\ell(\cdot, \cdot)$ is the task-specific loss function, and $\Psi^*_{\text{LLM}}(\cdot)$ means that the LLM is frozen. However, since we do not have access to $\mathcal{D}_{\text{TEST}}$ during the training phase, it is impractical to optimize the demonstration selection directly by minimizing $\mathcal{L}(\mathcal{D}_{\text{DEMO}}, \mathcal{D}_{\text{TEST}})$.

Instead, we re-consider the demonstration selection task as a two-stage problem, where we first *pre-select* a subset of *high-quality* demonstrations from $\mathcal{D}_{\text{TRAIN}}$ as the retrieval pool, i.e., a FEEDER set denoted as $\mathcal{D}_{\text{FEEDER}}$; and then we apply existing demonstration selectors such as random or similarity-based retrievers on $\mathcal{D}_{\text{FEEDER}}$, to choose the corresponding demonstrations as context for a specific test instance. Our key idea is that a high-quality training dataset $\mathcal{D}_{\text{FEEDER}}$ should be both representative of the entire training dataset $\mathcal{D}_{\text{TRAIN}}$ and as minimal in size as possible. Formally, we use the loss function $\mathcal{L}(\mathcal{D}_{\text{FEEDER}}, \mathcal{D}_{\text{TRAIN}})$ from Eq. (1) to evaluate our *pre-selector*, i.e., how well the representation of $\mathcal{D}_{\text{FEEDER}}$ aligns with $\mathcal{D}_{\text{TRAIN}}$. Then, our objective can be written as:

$$\min_{\mathcal{D}_{\text{FEEDER}} \subseteq \mathcal{D}_{\text{TRAIN}}} |\mathcal{D}_{\text{FEEDER}}|, \text{ s.t. } \mathcal{L}(\mathcal{D}_{\text{FEEDER}}, \mathcal{D}_{\text{TRAIN}}) \leq \mathcal{L}(\mathcal{D}_{\text{TRAIN}}, \mathcal{D}_{\text{TRAIN}}). \tag{2}$$

This formulation indices that $\mathcal{D}_{\text{FEEDER}}$ should be not only sufficient but also necessary to represent $\mathcal{D}_{\text{TRAIN}}$, thus removing redundant data points to save computation costs meanwhile maintaining LLM performance.

Our pre-selected set of high-quality data $\mathcal{D}_{\text{FEEDER}}$ also can be applied to fine-tune LLMs. Concretely, instead of fine-tuning LLMs on the entire training dataset $\mathcal{D}_{\text{TRAIN}}$, $\mathcal{D}_{\text{FEEDER}}$ allows us to fine-tune LLMs with few but high-quality data, reducing computation costs. In this case, the LLM $\Psi_{\text{LLM}}$ is usually trainable, and our goal can be formulated as:

$$\min_{\Psi_{\text{LLM}}} \mathbb{E}_{(\boldsymbol{x}_n, \boldsymbol{y}_n) \in \mathcal{D}^*_{\text{FEEDER}}}[\ell\Big(\Psi_{\text{LLM}}(\boldsymbol{x}_n, \emptyset), \boldsymbol{y}_n\Big)], \tag{3}$$

where $\mathcal{D}^*_{\text{FEEDER}}$ means that the selected $\mathcal{D}_{\text{FEEDER}}$ is fixed during fine-tuning.

Given the above analysis, we can further bridge the (pre)-selection of $\mathcal{D}_{\text{FEEDER}}$ and the LLM fine-tuning on $\mathcal{D}_{\text{FEEDER}}$ into a bi-level optimization framework. On the outer level, following Eq. (2), we optimize the selection of $\mathcal{D}_{\text{FEEDER}}$ in the context of a frozen LLM $\Psi^*_{\text{LLM}}$; while on the inner level, following Eq. (3), we optimize the LLM $\Psi_{\text{LLM}}$ using the fixed dataset $\mathcal{D}^*_{\text{FEEDER}}$. The bi-level optimization procedure described above is amenable to repetition, enabling iterative refinement of both the selected $\mathcal{D}_{\text{FEEDER}}$ and the tuned LLM. The overall process is summarized in Algorithm 1, and the construction of our FEEDER set is detailed in the subsequent sections.

---

**Algorithm 1:** Bi-level Optimization

**Input:** Training dataset $\mathcal{D}_{\text{TRAIN}}$, LLM $\Psi_{\text{LLM}}$.

**Output:** Approximated set $\widetilde{\mathcal{D}}_{\text{FEEDER}}$, tuned LLM $\Psi_{\text{LLM}}$.

Initialize $\widetilde{\mathcal{D}}_{\text{FEEDER}} = \mathcal{D}_{\text{TRAIN}}$.

**for** *each iteration* **do**

    Update $\widetilde{\mathcal{D}}_{\text{FEEDER}}$ by using our approximation algorithm with frozen LLM $\Psi_{\text{LLM}}$.

    Tune LLM $\Psi_{\text{LLM}}$ by using Eq. (3) as our loss function on fixed $\widetilde{\mathcal{D}}_{\text{FEEDER}}$.

**end**

---

## 3 CONNECTIONS TO EXISTING WORK

With the growing capabilities of LLMs, data (often referred to as "demonstrations") selection has gained prominence, which involves selecting suitable examples as the context for in-context learning (Dong et al., 2022; Yang et al., 2023; Zhou et al., 2022) or filtering a subset from training examples for fine-tuning (Sachdeva et al., 2024; Zhou et al., 2024). Previous solutions have revolved around constructing either parameter-free selection mechanisms (Wang et al., 2022; Zemlyanskiy et al., 2022; Gao et al., 2023) or neural-based selection methods (Pasupat et al., 2021; Liu et al., 2021; Gupta et al., 2021; Rubin et al., 2021; Li et al., 2023). Recent investigations (Xia et al., 2024; Marion et al., 2023) focus on mining training examples for fine-tuning specific tasks, with (Wang et al., 2024) extending this approach to in-context learning. In contrast to previous methods that use LLMs as demonstration

selectors, our work leverages the powerful few-shot inference capabilities of LLMs by employing them as *pre-selectors*. Building on the observation from (Jang & Lukasiewicz, 2023) that LLMs excel at high-level logical reasoning such as transitive inference, our approach examines "sufficiency" and "necessity" to identify a core set of training examples. This pre-selection process remains consistent regardless of test datasets, thereby eliminating the need for re-computation across different test sets. The resulting FEEDER sets can serve a dual purpose: they can be used as candidate input contexts or to fine-tune the LLM. In both scenarios, FEEDER can significantly reduce the computation costs by substituting the entire training dataset with FEEDER sets.

## 4 FEEDER: PRE-SELECTING SUFFICIENT AND NECESSARY DEMONSTRATIONS

Let $X, C$ denote variables for the input and the context (i.e., selected demonstrations). We introduce $Y$, a boolean variable, to represent whether the corresponding output is correct. For simplicity, we use $Y_{\boldsymbol{x}_n} = 1$ to denote $Y = 1|X = \boldsymbol{x}_n$, meaning that the LLM generates the correct output for the input $\boldsymbol{x}_n$. Similarly, $Y_{\boldsymbol{x}_n} = 0$, equivalent to $Y = 0|X = \boldsymbol{x}_n$, indicates that LLM produces an incorrect output for $\boldsymbol{x}_n$.

For convenience, we introduce $S$, a variable to record the original status of the LLM before new plug-in and unplug operations (denoted as $\texttt{plug}(\cdot)$ and $\texttt{unplug}(\cdot)$ respectively). The connections between the above operations and the $\texttt{do}(\cdot)$ operation in causality are discussed in Appendix A1.

### 4.1 RELATIONSHIP BETWEEN DEMONSTRATIONS: FROM INSTANCE LEVEL TO SET LEVEL

We begin by considering the relationship between two examples, denoted as $(\boldsymbol{x}_n, \boldsymbol{y}_n)$ and $(\boldsymbol{x}_m, \boldsymbol{y}_m)$.

Sufficiency relationship is introduced to assess whether plugging in one data point is adequate for the LLM to produce the correct answer to another data point. Formally, we define sufficiency as:

**Definition 1** (Sufficient Instance). *Given tuple $(X, Y, C, S)$, a training sample $(\boldsymbol{x}_n, \boldsymbol{y}_n)$ is considered sufficient for another one $(\boldsymbol{x}_m, \boldsymbol{y}_m)$, if the following equation holds:*

$$Y_{\boldsymbol{x}_m} = 1|\texttt{plug}((\boldsymbol{x}_n, \boldsymbol{y}_n)); C = \emptyset, S = (Y_{\boldsymbol{x}_m} = 0). \tag{4}$$

*It means that when plugging in $(\boldsymbol{x}_n, \boldsymbol{y}_n)$, it would correct the LLM's answer to $\boldsymbol{x}_m$.*

Necessity relationship is introduced to assess whether it is necessary to retain a particular plugged-in data point to maintain the correct output of another data point. Its formal definition can be written as:

**Definition 2** (Necessary Instance). *Given tuple $(X, Y, C, S)$, a training sample $(\boldsymbol{x}_n, \boldsymbol{y}_n)$ is considered necessary for another one $(\boldsymbol{x}_m, \boldsymbol{y}_m)$, if the following equation holds:*

$$Y_{\boldsymbol{x}_m} = 0|\texttt{unplug}((\boldsymbol{x}_n, \boldsymbol{y}_n)); C = ((\boldsymbol{x}_n, \boldsymbol{y}_n)), S = (Y_{\boldsymbol{x}_m} = 1). \tag{5}$$

*It means that prior to unplugging $(\boldsymbol{x}_n, \boldsymbol{y}_n)$, the LLM's output is correct. However, when we do unplug $(\boldsymbol{x}_n, \boldsymbol{y}_n)$ from the context, it causes the LLM to offer an incorrect output.*

The above definitions of sufficiency and necessity metrics, operating on the instance level, are further clarified with examples in Appendix A2.1. Extending these definitions to the set level, a sufficient set signifies that plugging in a specific set is adequate to ensure the correct outputs for all examples in another set, while a necessary set implies that removing any example from this set would result in incorrect answers for at least one example within another set. Formal definitions for the above set-level metrics, along with examples, are available in Appendix A2.2.

Taking into account both the sufficiency and necessity metrics, we define a subset of the training dataset $\mathcal{D}_{\texttt{TRAIN}}$ as $\mathcal{D}_{\texttt{FEEDER}}$, if it can be both sufficient and necessary to represent $\mathcal{D}_{\texttt{TRAIN}}$. Formally, we describe $\mathcal{D}_{\texttt{FEEDER}}$ as follows:

**Definition 3** (FEEDER Set). *Given tuple $(X, Y, C, S)$ and $\mathcal{D}_{\texttt{TRAIN}}$, a subset of $\mathcal{D}_{\texttt{TRAIN}}$, is considered as a FEEDER set (denoted as $\mathcal{D}_{\texttt{FEEDER}}$), if the following conditions are satisfied:*

*(i) $Y_{(\boldsymbol{x}_1..., \boldsymbol{x}_N)} = \boldsymbol{1}_N|\texttt{plug}(\mathcal{D}_{\texttt{FEEDER}}); C = \emptyset, S = (Y_{(\boldsymbol{x}_1..., \boldsymbol{x}_N)} \neq \boldsymbol{1}_N)$ holds.*

*(ii) $Y_{(\boldsymbol{x}_1..., \boldsymbol{x}_N)} \neq \boldsymbol{1}_N|\texttt{unplug}(\mathcal{D}'_{\texttt{FEEDER}}); C = \mathcal{D}_{\texttt{TRAIN}}, S = (Y_{(\boldsymbol{x}_1..., \boldsymbol{x}_N)} = \boldsymbol{1}_N)$ holds for any subset of $\mathcal{D}_{\texttt{FEEDER}}$ (denoted as $\mathcal{D}'_{\texttt{FEEDER}}$).*

$\mathbf{1}_N$ denotes $N$-*dimensional vectors whose elements are all 1s. (i) and (ii) respectively imply that plugging in* $\mathcal{D}_{\text{FEEDER}}$ *is sufficient and necessary to maintain the LLM generating correct output.*

We illustrate the concept of FEEDER via specific examples in Appendix A2.3. Strictly following the above definition to discover a FEEDER set is impractical because the constraints are too stringent and the computational costs are prohibitively high with $O(2^N)$ computational complexity. Therefore, we propose an approximation algorithm for discovering a FEEDER set in the following subsection.

## 4.2 AN APPROXIMATION ALGORITHM FOR DISCOVERING FEEDER

Grounded in the observation by (Jang & Lukasiewicz, 2023) that LLMs excel at transitive inference, we hypothesize that sufficiency is transitive among sets. Specifically, if $\mathcal{D}_{\text{A}}$ is a sufficient set for $\mathcal{D}_{\text{B}}$, and $\mathcal{D}_{\text{B}}$ is a sufficient set for $\mathcal{D}_{\text{C}}$, then $\mathcal{D}_{\text{A}}$ is also a sufficient set for $\mathcal{D}_{\text{C}}$. We provide case studies in Appendix A11.1 to verify the feasibility of this assumption. Based on this, we design a tree-based algorithm to filter out unnecessary portions of $\mathcal{D}_{\text{TRAIN}}$, while retaining the sufficient subset to represent the entire $\mathcal{D}_{\text{TRAIN}}$.

Concretely, we exploit the transitivity to construct a tree, where each node represents a set of instances; and our tree expands from the bottom to the top. Formally, we use the variable $K$ to represent the depth of the tree, corresponding to the number of iterations. Specifically, we use $k = 1, 2, \ldots, K$ to refer to each $k$-th iteration; and during each $k$-th iteration, we generate the $(k+1)$-th of the tree. We denote $\mathscr{W}_k$ as the set of nodes after the $k$-th iteration. We initialize $\mathscr{W}_0$ by assigning all the samples in $\mathcal{D}_{\text{TRAIN}}$ as the bottom nodes:

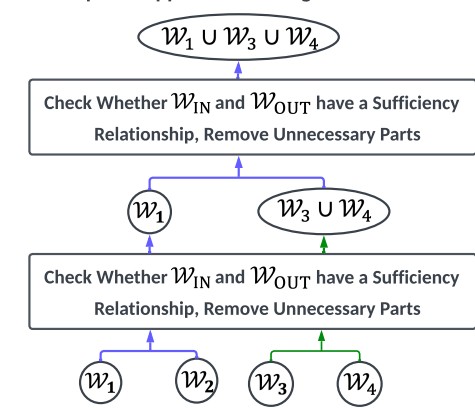

An Example of Approximation Algorithm for FEEDER

Figure 2: An illustrated example of our approximation algorithm for FEEDER. At each iteration (corresponding to each layer of the tree), we check whether there is a *sufficiency* relationship between each pair of nodes. After each check, we remove those *unnecessary* parts from $\mathscr{W}_.$.

$$\mathscr{W}_0 := \{\mathcal{W}_n := \{(\boldsymbol{x}_n, \boldsymbol{y}_n)\} | (\boldsymbol{x}_n, \boldsymbol{y}_n) \in \mathcal{D}_{\text{TRAIN}}\}. \tag{6}$$

During each $k$-th iteration, we generate $\mathscr{W}_k$ from $\mathscr{W}_{k-1}$. This is achieved by examining the sufficiency relationship between every pair of nodes in $\mathscr{W}_{k-1}$, denoted as $\mathcal{W}_i, \mathcal{W}_j \in \mathscr{W}_{k-1}$. In this evaluation, we assess whether the following equation holds true by assigning $\mathcal{W}_i$ and $\mathcal{W}_j$ as $\mathcal{W}_{\text{IN}}$ and $\mathcal{W}_{\text{OUT}}$, or vice versa:

$$Y_{(\{\boldsymbol{x}_n | \boldsymbol{x}_n \in \mathcal{W}_{\text{OUT}}\})} = \mathbf{1}_{|\mathcal{W}_{\text{OUT}}|} | \text{plug}(\mathcal{W}_{\text{IN}}); C = \emptyset, S, \tag{7}$$

where $S$ is loosened to allow for any value. If the above equation holds, it signifies that plugging in $\mathcal{W}_{\text{IN}}$ is *sufficient* for the LLM to generate the correct output to any input in $\mathcal{W}_{\text{OUT}}$. In other words, once we have $\mathcal{W}_{\text{IN}}$ included in the plugged-in context, it is *unnecessary* to further include $\mathcal{W}_{\text{OUT}}$. Formally, we can derive the following equation from Eq. (7):

$$Y_{(\{\boldsymbol{x}_n | \boldsymbol{x}_n \in \mathcal{W}_{\text{OUT}}\})} = \mathbf{1}_{|\mathcal{W}_{\text{OUT}}|} | \text{unplug}(\mathcal{W}_{\text{OUT}}); C = (\mathcal{W}_{\text{IN}} \cup \mathcal{W}_{\text{OUT}}), S, \tag{8}$$

where $S$ is loosened to be any value. Concretely, there are three possible scenarios by examining each pair of nodes in $\mathscr{W}_{k-1}$: (i) If both $\mathcal{W}_i$ and $\mathcal{W}_j$ are sufficient sets for each other, then we select the one with fewer elements to append to $\mathscr{W}_k$. (ii) If only one of $\mathcal{W}_i$ and $\mathcal{W}_j$ is a sufficient set for the other, then we append the sufficient set to $\mathscr{W}_k$. (iii) If neither $\mathcal{W}_i$ nor $\mathcal{W}_j$ is a sufficient set, we append $\mathcal{W}_i \cup \mathcal{W}_j$ to $\mathscr{W}_k$. After performing the above calculations for each pair of nodes, we remove them from $\mathscr{W}_{k-1}$. When there is only one element left in $\mathscr{W}_{k-1}$, it is directly appended to $\mathscr{W}_k$. This process continues until $\mathscr{W}_.$ contains only one element.

We can effectively remove unnecessary samples from $\mathcal{D}_{\text{TRAIN}}$ by extending the above tree structure from the bottom to the top. Simultaneously, the complexity of the above algorithm with $K$ iterations (corresponding to a tree depth of $K + 1$) is $O(K \log_2^{|\mathcal{D}_{\text{TRAIN}}|})$. In practice, we investigate the impact of varying $K$ and find that setting $K = 1$ already yields excellent performance. This indicates that one-shot inference by the LLM to assess sufficiency between each pair of samples is sufficient. Once the results are computed, we merge them to form the resulting set. Figure 2 illustrates the process for $K = 2$. When $K = 1$, the top-level check between $\mathcal{W}_1$ and $\mathcal{W}_1 \cup \mathcal{W}_2$ is no longer required.

Table 1: Performance comparisons on text classification datasets are conducted in the in-context learning setting. We report both the mean and variance of accuracy using 8 different seeds and 5 different permutations of n-shots. Refer to Appendix A5.2 for more extended results on datasets FPB, SST-5, TREC.

| $\Psi_{LLM}(\cdot)$ | $\mathcal{D}$ | $n$ | SUBJ | | | SST-2 | | | COLA | | |
|---|---|---|---|---|---|---|---|---|---|---|---|
| | | | RAN | SIM | DIV | RAN | SIM | DIV | RAN | SIM | DIV |
| SMA (0.3B) | $\mathcal{D}_{TRAIN}$ | 1 | 41.3 (7.2) | 41.1 (0.1) | 41.1 (0.1) | 48.9 (4.6) | 24.5 (0.2) | 24.5 (0.2) | 29.0 (5.4) | 38.8 (0.1) | 38.8 (0.1) |
| | | 2 | 47.3 (7.2) | 62.8 (0.1) | 71.9 (0.2) | 51.2 (5.8) | 65.7 (0.1) | 62.5 (0.2) | 30.9 (4.6) | 38.5 (0.2) | 36.2 (0.1) |
| | | 5 | 51.8 (5.5) | 85.8 (0.3) | 70.1 (0.2) | 62.6 (5.6) | 79.4 (0.2) | 61.7 (0.1) | 39.4 (5.8) | 49.3 (0.1) | 47.0 (0.2) |
| | | 10 | 62.4 (5.0) | 88.0 (0.2) | 78.2 (0.1) | 50.9 (4.9) | 83.8 (0.3) | 76.9 (0.2) | 31.6 (4.6) | 52.5 (0.2) | 58.8 (0.2) |
| | $\mathcal{D}_{FEEDER}$ | 1 | **42.8** (2.4) | **44.9** (1.1) | **44.9** (1.1) | **49.8** (4.2) | 48.1 (1.9) | 48.1 (1.9) | **29.6** (4.1) | 35.1 (1.5) | 35.1 (1.5) |
| | | 2 | **55.9** (3.3) | 63.4 (1.6) | **74.7** (0.9) | 67.3 (2.4) | 67.7 (1.4) | **64.7** (1.5) | 31.3 (2.2) | **41.7** (1.2) | 34.9 (1.9) |
| | | 5 | **57.5** (4.0) | **86.9** (0.7) | 69.8 (1.0) | **70.3** (4.4) | 77.9 (1.4) | **68.5** (1.9) | 35.2 (2.0) | **57.3** (1.2) | **54.6** (1.7) |
| | | 10 | **63.5** (4.4) | **88.7** (1.5) | **79.7** (2.0) | 75.2 (6.2) | 83.0 (1.7) | **77.2** (1.5) | **59.3** (3.8) | **68.7** (2.4) | **68.5** (2.9) |
| MED (0.8B) | $\mathcal{D}_{TRAIN}$ | 1 | 42.5 (5.2) | 43.6 (0.1) | 43.6 (0.1) | 49.0 (4.3) | 42.3 (0.2) | 42.3 (0.2) | 42.1 (5.7) | 48.3 (0.1) | 48.3 (0.1) |
| | | 2 | 58.1 (6.3) | 88.3 (0.2) | 87.0 (0.3) | 68.0 (5.2) | 70.7 (0.1) | 59.6 (0.2) | 41.1 (4.2) | 36.8 (0.2) | 37.7 (0.1) |
| | | 5 | 66.7 (4.5) | 86.2 (0.2) | 86.7 (0.1) | 49.1 (4.3) | 80.6 (0.1) | 67.5 (0.2) | 46.2 (4.7) | 53.8 (0.2) | 48.5 (0.3) |
| | | 10 | 48.6 (6.0) | 85.9 (0.1) | 73.9 (0.2) | 71.1 (4.5) | 84.6 (0.1) | 73.1 (0.2) | 43.4 (4.5) | 55.5 (0.2) | 56.1 (0.4) |
| | $\mathcal{D}_{FEEDER}$ | 1 | **45.8** (5.1) | **46.4** (0.4) | **46.4** (0.4) | **49.1** (3.0) | **47.7** (1.3) | **47.7** (1.3) | **46.6** (3.8) | 45.1 (1.1) | 45.1 (1.1) |
| | | 2 | **63.1** (4.5) | **89.7** (1.5) | 86.8 (1.3) | **69.8** (3.8) | **73.0** (2.9) | **61.2** (2.1) | 36.6 (3.5) | **37.0** (2.8) | **34.6** (2.0) |
| | | 5 | **73.4** (4.3) | **88.2** (1.9) | **88.8** (1.7) | 59.3 (2.4) | **80.9** (1.3) | **69.6** (1.7) | **59.2** (3.3) | **68.6** (1.6) | **66.6** (1.7) |
| | | 10 | 52.0 (3.8) | **87.4** (1.3) | **75.6** (1.2) | **76.0** (3.0) | **86.7** (1.4) | **75.6** (1.8) | **59.3** (4.8) | **68.8** (2.0) | **68.9** (1.8) |
| NEO (1.3B) | $\mathcal{D}_{TRAIN}$ | 1 | 42.8 (3.9) | 42.1 (0.1) | 42.1 (0.1) | 49.2 (3.7) | 33.8 (0.1) | 33.8 (0.1) | 25.5 (3.4) | 36.5 (0.2) | 36.5 (0.2) |
| | | 2 | 48.5 (4.2) | 88.3 (0.2) | 72.6 (0.3) | 76.8 (3.5) | 81.5 (0.1) | 76.3 (0.4) | 30.7 (3.1) | 55.5 (0.2) | 56.5 (0.4) |
| | | 5 | 51.6 (5.0) | 90.5 (0.2) | 81.7 (0.2) | 65.1 (3.5) | 80.8 (0.2) | 66.1 (0.3) | 40.0 (3.6) | 55.9 (0.1) | 52.5 (0.2) |
| | | 10 | 48.5 (5.8) | 85.9 (0.3) | 81.9 (0.1) | 69.8 (4.8) | 84.1 (0.1) | 69.7 (0.1) | 39.6 (4.5) | 59.3 (0.3) | 63.4 (0.1) |
| | $\mathcal{D}_{FEEDER}$ | 1 | **43.2** (4.0) | **46.3** (1.0) | **46.3** (1.0) | **49.3** (5.1) | **48.3** (1.9) | **48.3** (1.9) | **28.3** (5.4) | 34.8 (1.3) | 34.8 (1.3) |
| | | 2 | **62.6** (3.5) | **89.4** (1.5) | **73.8** (2.1) | 75.1 (2.8) | **82.6** (2.1) | **78.5** (1.9) | **59.3** (3.7) | **64.7** (1.4) | **64.7** (1.4) |
| | | 5 | **69.4** (5.6) | **91.2** (1.8) | **82.9** (1.3) | **73.2** (4.2) | **82.9** (2.7) | **71.6** (2.4) | **58.7** (3.2) | **67.2** (2.4) | **65.8** (1.8) |
| | | 10 | **58.7** (3.3) | **87.2** (1.7) | **84.3** (2.8) | **72.4** (3.4) | **85.8** (2.5) | **71.8** (2.9) | **59.8** (2.8) | **68.8** (1.4) | **68.9** (1.3) |
| GEM (2B) | $\mathcal{D}_{TRAIN}$ | 1 | 45.0 (5.9) | 48.1 (0.6) | 48.1 (0.6) | 51.2 (6.8) | 52.2 (0.8) | 52.2 (0.8) | 37.5 (7.0) | 40.5 (1.3) | 40.5 (1.3) |
| | | 2 | 62.3 (6.9) | 82.5 (1.8) | 74.2 (1.3) | 71.5 (5.6) | 78.5 (1.5) | 75.9 (0.9) | 40.6 (5.9) | 62.5 (1.0) | 61.6 (0.5) |
| | | 5 | 68.0 (7.1) | 91.5 (1.2) | 84.2 (1.6) | 70.2 (5.6) | 80.5 (1.6) | 80.6 (0.7) | 45.9 (5.6) | 67.2 (1.8) | 65.6 (0.6) |
| | | 10 | 50.3 (8.2) | 86.2 (1.9) | 85.6 (0.9) | 68.2 (4.8) | 85.5 (1.5) | 76.3 (1.3) | 50.2 (7.4) | 69.8 (1.5) | 71.5 (1.2) |
| | $\mathcal{D}_{FEEDER}$ | 1 | **48.2** (4.2) | **49.5** (1.0) | **49.5** (1.0) | **52.6** (4.6) | **53.1** (0.8) | **53.1** (0.8) | **38.9** (5.2) | 39.6 (0.8) | 39.6 (0.8) |
| | | 2 | **65.2** (2.9) | **85.2** (1.0) | **80.3** (0.8) | **74.2** (4.9) | **82.1** (1.2) | **83.0** (0.7) | **52.5** (2.5) | **68.9** (2.1) | **67.8** (1.5) |
| | | 5 | **72.2** (6.2) | **94.5** (5.3) | **85.5** (0.7) | **72.0** (4.2) | **83.6** (2.1) | **84.5** (1.7) | **55.2** (4.8) | **77.6** (2.5) | **73.9** (2.3) |
| | | 10 | **60.5** (4.0) | **86.5** (2.5) | **88.4** (2.4) | **70.5** (5.6) | **92.6** (2.6) | **78.5** (5.3) | **58.6** (4.6) | **75.6** (2.9) | **76.6** (2.5) |
| LAR (6B) | $\mathcal{D}_{TRAIN}$ | 1 | 44.9 (6.6) | 49.5 (0.1) | 49.5 (0.1) | 48.2 (2.9) | 47.0 (0.1) | 47.0 (0.1) | 38.9 (6.7) | 41.2 (0.2) | 41.2 (0.2) |
| | | 2 | 55.4 (3.5) | 85.5 (0.2) | 86.5 (0.2) | 68.1 (4.2) | 78.7 (0.2) | 77.5 (0.1) | 42.8 (4.0) | 45.5 (0.3) | 45.6 (0.2) |
| | | 5 | 51.2 (4.4) | 90.8 (0.2) | 82.7 (0.1) | 75.2 (3.3) | 80.7 (0.1) | 77.8 (0.2) | 48.5 (3.3) | 51.8 (0.3) | 52.1 (0.2) |
| | | 10 | 57.7 (4.8) | 87.3 (0.1) | 85.3 (0.1) | 72.1 (3.8) | 77.6 (0.1) | 76.5 (0.2) | 59.1 (4.2) | 60.3 (0.1) | 61.0 (0.2) |
| | $\mathcal{D}_{FEEDER}$ | 1 | **43.9** (4.2) | **51.2** (1.0) | **51.2** (1.0) | **49.6** (2.4) | **51.3** (1.6) | **51.3** (1.6) | **41.2** (2.1) | **43.8** (1.8) | **43.8** (1.8) |
| | | 2 | **65.7** (3.0) | **91.5** (1.1) | **88.8** (1.6) | **73.5** (2.5) | **85.7** (4.2) | 76.1 (2.1) | **61.8** (2.1) | **63.1** (1.5) | **60.1** (1.4) |
| | | 5 | **53.7** (3.8) | **92.9** (0.8) | **91.5** (1.4) | **77.6** (4.0) | **81.0** (1.2) | **79.4** (1.0) | **62.4** (2.7) | **63.3** (1.4) | **65.8** (1.4) |
| | | 10 | **58.0** (3.4) | **88.8** (0.9) | **87.8** (1.2) | **83.8** (2.8) | **86.4** (2.0) | **87.2** (1.3) | **59.7** (3.0) | **67.5** (1.9) | **68.4** (2.2) |
| LLA (7B) | $\mathcal{D}_{TRAIN}$ | 1 | 42.9 (6.6) | 48.5 (0.1) | 48.5 (0.1) | 46.2 (2.7) | 49.1 (0.1) | 49.1 (0.1) | 40.1 (6.1) | 42.0 (0.2) | 42.0 (0.2) |
| | | 2 | 51.9 (4.4) | 90.7 (0.1) | 85.2 (0.2) | 67.8 (3.2) | 73.5 (0.2) | 74.5 (0.2) | 43.5 (4.5) | 47.4 (0.2) | 49.6 (0.1) |
| | | 5 | 51.6 (3.2) | 86.8 (0.2) | 82.9 (0.1) | 74.8 (3.8) | 81.2 (0.2) | 78.7 (0.2) | 50.2 (3.7) | 52.6 (0.2) | 48.2 (0.3) |
| | | 10 | 56.1 (4.6) | 81.3 (0.1) | 85.7 (0.1) | 73.2 (3.1) | 76.3 (0.1) | 77.1 (0.1) | 55.3 (4.2) | 55.3 (0.2) | 60.0 (0.4) |
| | $\mathcal{D}_{FEEDER}$ | 1 | **43.8** (4.3) | **49.7** (1.0) | **49.7** (1.0) | **47.2** (2.4) | **50.8** (1.7) | **50.8** (1.7) | **41.2** (2.1) | **43.8** (1.8) | **43.8** (1.8) |
| | | 2 | **54.8** (3.0) | **92.5** (1.1) | 84.8 (0.7) | **72.2** (3.1) | **82.5** (4.0) | **80.1** (2.6) | **50.8** (2.3) | **58.6** (1.7) | **53.5** (1.3) |
| | | 5 | **53.7** (3.8) | **87.9** (1.8) | **91.5** (1.4) | **78.3** (4.6) | **83.2** (1.1) | **80.1** (1.4) | **53.8** (2.8) | **65.3** (1.6) | **61.8** (1.4) |
| | | 10 | **58.0** (3.4) | **85.8** (0.9) | **87.8** (1.2) | **85.0** (2.2) | **87.1** (2.2) | **86.9** (1.0) | **60.5** (3.1) | **68.0** (1.7) | **68.4** (2.0) |

Table 2: A complementary table to Table 1 presents the corresponding results for the demonstration selectors UNC, CLU, LVM.

| $\Psi_{LLM}(\cdot)$ | $\mathcal{D}$ | $n$ | SUBJ | | | SST-2 | | | COLA | | |
|---|---|---|---|---|---|---|---|---|---|---|---|
| | | | UNC | CLU | LVM | UNC | CLU | LVM | UNC | CLU | LVM |
| LAR (6B) | $\mathcal{D}_{TRAIN}$ | 1 | 53.5 (6.3) | 49.3 (4.4) | 51.5 (2.1) | 49.0 (2.9) | 47.5 (1.5) | 47.8 (1.1) | 42.0 (6.5) | 39.8 (1.5) | 40.2 (1.2) |
| | | 2 | 87.8 (3.7) | 86.5 (4.1) | 86.3 (3.5) | 75.6 (4.2) | 80.1 (2.2) | 79.0 (2.4) | 49.6 (4.0) | 46.8 (5.0) | 47.5 (3.3) |
| | | 5 | 90.7 (4.5) | 88.2 (4.4) | 89.4 (4.2) | 81.8 (3.3) | 82.2 (3.3) | 80.7 (4.4) | 55.4 (3.5) | 56.4 (4.3) | 58.8 (3.3) |
| | | 10 | 88.3 (4.8) | 90.7 (3.8) | 91.3 (4.1) | 80.5 (3.8) | 78.8 (3.9) | 76.8 (4.1) | 58.4 (4.2) | 62.1 (3.6) | 61.5 (4.5) |
| | $\mathcal{D}_{FEEDER}$ | 1 | **55.3** (4.2) | **50.9** (4.4) | 50.2 (3.2) | **50.3** (2.4) | **48.4** (3.4) | **48.3** (2.6) | **43.8** (2.1) | **40.8** (3.5) | **42.5** (5.1) |
| | | 2 | **89.8** (3.0) | **89.7** (3.5) | **89.5** (2.5) | **77.1** (2.5) | **82.5** (3.5) | **83.0** (3.2) | **60.0** (2.1) | **57.8** (4.4) | **58.1** (3.5) |
| | | 5 | **92.3** (3.8) | **92.0** (2.4) | **91.8** (2.9) | 81.2 (4.0) | 80.8 (3.8) | 80.4 (2.9) | **62.4** (2.7) | **61.6** (3.7) | **62.3** (2.4) |
| | | 10 | **90.8** (3.4) | **92.0** (2.4) | **91.8** (2.9) | **81.2** (2.8) | **80.8** (3.8) | **80.4** (2.9) | **62.4** (3.0) | **62.7** (3.1) | **62.5** (2.5) |
| LLA (7B) | $\mathcal{D}_{TRAIN}$ | 1 | 49.0 (6.6) | 48.5 (5.6) | 47.5 (5.1) | 49.2 (2.7) | 48.2 (3.7) | 48.7 (3.1) | 40.1 (6.1) | 41.1 (4.1) | 41.0 (3.2) |
| | | 2 | 89.2 (4.4) | 87.8 (3.5) | 88.7 (4.1) | 75.1 (3.2) | 72.5 (2.2) | 74.7 (4.2) | 48.5 (4.5) | 45.2 (4.0) | 46.4 (1.2) |
| | | 5 | 82.9 (3.2) | 80.1 (2.2) | 83.8 (1.2) | 83.7 (3.8) | 81.5 (3.0) | 82.2 (1.2) | 53.2 (3.7) | 51.2 (2.5) | 52.6 (2.2) |
| | | 10 | 86.2 (4.6) | 82.1 (4.4) | 83.3 (2.1) | 76.4 (3.1) | 75.2 (3.7) | 74.8 (4.1) | 63.5 (4.3) | 62.6 (4.0) | 60.3 (2.2) |
| | $\mathcal{D}_{FEEDER}$ | 1 | **49.7** (4.3) | 45.8 (4.3) | **48.7** (5.1) | **51.8** (2.4) | **48.4** (3.5) | **50.3** (2.7) | **43.0** (2.1) | **42.2** (2.5) | **42.8** (1.6) |
| | | 2 | **91.8** (3.0) | **90.8** (3.4) | **91.5** (2.4) | **78.1** (3.1) | **73.5** (3.1) | **76.5** (4.0) | **49.5** (2.3) | **48.8** (2.3) | **50.6** (2.7) |
| | | 5 | **89.5** (3.8) | **88.7** (4.8) | **86.9** (2.8) | **84.1** (4.6) | **82.3** (4.5) | **83.8** (4.1) | **60.8** (2.8) | **58.8** (3.8) | **59.3** (2.6) |
| | | 10 | **88.8** (3.4) | **88.0** (4.4) | **86.8** (2.9) | 80.9 (2.2) | **85.1** (2.0) | **83.4** (2.2) | **67.4** (3.1) | **64.5** (3.4) | **66.0** (2.7) |

Table 3: Performance comparisons on reasoning GSM8K dataset and semantic-parsing SMCALFlow dataset are conducted in the in-context learning setting. We report both the mean and variance of accuracy using 8 different seeds and 5 different permutations of n-shots. Refer to Appendix A5.3 for more extended results on demonstration selectors CLU, LVM.

| $\Psi_{\text{LLM}}(\cdot)$ | $\mathcal{D}$ | $n$ | GSM8K | | | | SMCALFlow | | | |
|---|---|---|---|---|---|---|---|---|---|---|
| | | | RAN | SIM | DIV | UNC | RAN | SIM | DIV | UNC |
| GEM (2B) | $\mathcal{D}_{\text{TRAIN}}$ | 1 | 6.54 (1.56) | 15.16 (0.17) | 15.16 (0.17) | 10.51 (0.78) | 8.54 (1.64) | 19.12 (0.15) | 19.12 (0.15) | 11.21 (0.89) |
| | | 2 | 8.56 (0.85) | 18.89 (0.85) | 19.52 (0.45) | 17.58 (0.27) | 9.56 (0.84) | 20.05 (0.36) | 22.50 (0.41) | 13.58 (0.77) |
| | | 5 | 15.30 (2.89) | 20.31 (0.58) | 21.56 (0.78) | 19.30 (0.90) | 18.56 (4.58) | 28.65 (0.95) | 27.89 (1.85) | 25.22 (3.56) |
| | | 10 | 17.45 (4.21) | 21.52 (0.49) | 20.85 (0.55) | 20.66 (1.84) | 19.85 (5.21) | 30.58 (1.04) | 28.56 (0.58) | 31.00 (0.88) |
| | $\mathcal{D}_{\text{FEEDER}}$ | 1 | **10.25** (0.51) | **16.25** (0.21) | **16.25** (0.21) | **11.12** (1.78) | **9.64** (0.55) | **20.54** (0.66) | **20.54** (0.66) | **15.25** (0.87) |
| | | 2 | **13.76** (0.48) | **19.68** (0.13) | **20.51** (1.55) | 16.85 (3.65) | **10.25** (0.52) | 20.03 (0.18) | **24.25** (2.65) | **17.58** (6.58) |
| | | 5 | **18.52** (5.21) | **22.58** (0.85) | **22.05** (0.77) | **20.20** (2.05) | **20.44** (5.12) | **30.54** (4.58) | **32.54** (5.21) | **28.95** (3.66) |
| | | 10 | **19.20** (5.22) | **22.20** (1.45) | **23.52** (2.20) | **22.10** (6.21) | **21.52** (2.01) | **31.48** (1.52) | **31.02** (2.54) | 30.01 (1.20) |
| LAR (6B) | $\mathcal{D}_{\text{TRAIN}}$ | 1 | 1.21 (0.83) | 2.84 (0.25) | 2.84 (0.25) | 2.54 (0.21) | 1.78 (0.72) | 10.21 (0.85) | 10.21 (0.85) | 9.25 (0.77) |
| | | 2 | 1.44 (0.65) | 4.01 (0.13) | 5.21 (0.25) | 4.25 (0.85) | 2.67 (0.98) | 9.91 (0.20) | 10.02 (0.88) | 8.54 (0.74) |
| | | 5 | 2.58 (0.85) | 6.85 (0.78) | 8.02 (1.84) | 7.88 (1.95) | 6.20 (0.84) | 14.02 (1.58) | 12.05 (1.88) | 10.88 (2.01) |
| | | 10 | 3.20 (0.77) | 7.05 (1.20) | 8.14 (1.65) | 8.01 (1.01) | 8.05 (0.84) | 15.25 (1.77) | 13.33 (1.54) | 11.99 (1.65) |
| | $\mathcal{D}_{\text{FEEDER}}$ | 1 | **2.27** (0.49) | **3.11** (0.15) | **3.11** (0.15) | **3.00** (0.56) | **2.35** (0.59) | **11.52** (1.85) | **11.52** (1.85) | **10.42** (1.02) |
| | | 2 | **2.80** (0.53) | **4.16** (0.14) | **5.55** (0.82) | **4.85** (1.20) | **3.51** (0.71) | **10.73** (0.07) | **11.05** (0.80) | **9.22** (1.03) |
| | | 5 | **3.24** (0.84) | **8.25** (1.58) | **8.47** (0.77) | **7.99** (1.25) | **6.88** (0.66) | **15.20** (1.58) | **14.44** (1.69) | **12.00** (2.03) |
| | | 10 | **3.66** (0.80) | **7.52** (1.88) | **8.55** (2.21) | **8.10** (2.28) | **8.66** (1.03) | **16.85** (3.21) | **15.55** (2.90) | **13.50** (2.25) |
| LLA (7B) | $\mathcal{D}_{\text{TRAIN}}$ | 1 | 2.45 (0.83) | 3.52 (0.88) | 3.52 (0.88) | 3.05 (0.25) | 2.25 (0.64) | 10.25 (0.85) | 10.25 (0.85) | 9.01 (0.33) |
| | | 2 | 2.65 (0.77) | 4.97 (0.18) | 5.62 (0.85) | 4.12 (0.47) | 4.97 (0.84) | 10.05 (2.36) | 10.52 (1.45) | 11.20 (1.54) |
| | | 5 | 3.54 (0.88) | 8.25 (0.89) | 7.25 (0.96) | 7.88 (0.64) | 7.52 (0.85) | 16.20 (1.85) | 15.28 (1.75) | 15.33 (1.30) |
| | | 10 | 4.25 (0.36) | 8.85 (0.85) | 9.21 (1.98) | 8.10 (1.11) | 8.70 (1.05) | 18.95 (1.25) | 19.55 (2.01) | 17.52 (2.66) |
| | $\mathcal{D}_{\text{FEEDER}}$ | 1 | **3.54** (0.51) | **4.44** (0.89) | **4.44** (0.89) | **3.36** (0.66) | **3.64** (0.55) | **10.89** (0.63) | **10.89** (0.63) | **10.02** (0.69) |
| | | 2 | **3.76** (0.48) | **5.68** (0.13) | **6.66** (0.58) | **4.85** (0.88) | **4.25** (0.52) | **12.03** (0.16) | **11.13** (1.10) | **12.50** (2.01) |
| | | 5 | **4.20** (1.23) | **9.22** (1.01) | **8.81** (0.98) | **8.20** (1.14) | **8.25** (1.25) | **17.20** (3.66) | **16.66** (5.20) | **16.06** (2.22) |
| | | 10 | **5.02** (1.51) | **10.22** (1.32) | **9.25** (0.79) | **9.45** (0.66) | **9.20** (0.77) | **20.11** (2.02) | **21.25** (3.36) | **20.22** (4.02) |
| LLA-3 (8B) | $\mathcal{D}_{\text{TRAIN}}$ | 1 | 78.24 (6.56) | 79.56 (3.42) | 79.56 (3.42) | 78.42 (3.76) | 12.37 (6.65) | 15.64 (2.34) | 15.64 (2.34) | 14.35 (4.56) |
| | | 2 | 79.55 (7.29) | 83.40 (4.53) | 83.67 (4.05) | 81.23 (3.53) | 13.21 (4.34) | 16.74 (3.45) | 17.43 (3.65) | 16.60 (4.62) |
| | | 5 | 81.45 (5.43) | 83.47 (5.63) | 84.52 (4.76) | 82.34 (5.34) | 14.53 (5.23) | 16.54 (2.35) | 17.87 (1.35) | 16.52 (3.21) |
| | | 10 | 82.31 (6.34) | 84.42 (3.24) | 84.53 (4.45) | 84.12 (4.44) | 14.63 (4.53) | 16.50 (2.21) | 18.64 (2.34) | 17.87 (2.23) |
| | $\mathcal{D}_{\text{FEEDER}}$ | 1 | **80.23** (4.43) | **81.21** (3.45) | **81.21** (3.45) | **79.64** (2.34) | **13.56** (3.22) | **16.55** (2.31) | **16.55** (2.31) | **15.40** (2.44) |
| | | 2 | **82.13** (4.76) | **84.43** (3.23) | **83.88** (3.33) | **82.22** (3.43) | **14.03** (3.35) | **17.45** (3.64) | **17.77** (3.20) | **17.00** (4.57) |
| | | 5 | **82.55** (5.96) | **85.03** (3.66) | **84.77** (3.77) | **83.56** (3.76) | **14.58** (3.45) | **18.22** (2.78) | **18.12** (2.01) | **17.53** (2.55) |
| | | 10 | **84.56** (2.33) | **85.79** (3.56) | **85.43** (4.55) | **84.98** (4.76) | **14.99** (4.65) | **16.66** (2.33) | **18.78** (3.42) | **18.01** (2.44) |

Our tree based approximation algorithm can also maintain the remaining set to be sufficient to represent the entire $\mathcal{D}_{\text{TRAIN}}$, as verified in the following proposition.

**Proposition 1** ($\widetilde{\mathcal{D}}_{\text{FEEDER}}$ is an Approximation of $\mathcal{D}_{\text{FEEDER}}$). *If we successively apply our tree based approximation algorithm on $\mathcal{D}_{\text{TRAIN}}$ for multiple runs to obtain a subset (denoted as $\widetilde{\mathcal{D}}_{\text{FEEDER}}$), then $\widetilde{\mathcal{D}}_{\text{FEEDER}}$ is sufficient to represent $\mathcal{D}_{\text{TRAIN}}$.*

We provide the proof of the above proposition in Appendix A3, which demonstrates that our approximation algorithm can effectively remove unnecessary samples from $\mathcal{D}_{\text{TRAIN}}$ while ensuring that the resulting set remains sufficient to represent the entire training dataset. The above tree based approximation algorithm is summarized in Algorithm 2 in Appendix A3.

Additionally, we present another algorithm for finding an exact sufficient and necessary subset from $\mathcal{D}_{\text{TRAIN}}$, along with its proof and deployment discussion, in Appendices A4.1 and A7. Moreover, our above tree-based algorithm can be iterated across multiple rounds to further reduce the necessary components. Specifically, the resulting FEEDER set from one round can be used as the input for the subsequent round. This iterative process can also yield an exact sufficient and necessary subset, as demonstrated in Appendix A4.2. Through empirical investigation, we examine the impact of varying the number of rounds $R$ and find that a single round ($R = 1$) already achieves great performance.

## 5 EVALUATING FEEDER INTO REAL-WORLD APPLICATIONS

Our primary focus is on the in-context learning setting, and we also extend it to the fine-tuning setting, where our *pre-selected* $\mathcal{D}_{\text{FEEDER}}$ can represent and replace the entire training dataset $\mathcal{D}_{\text{TRAIN}}$ to reduce the computation cost. Our evaluations are mainly conducted on 6 text classification datasets: SST-2 (Socher et al., 2013), SST-5 (Socher et al., 2013), COLA (Warstadt et al., 2018), TREC (Voorhees & Tice, 2000), SUBJ (Pang & Lee, 2004), and FPB (Malo et al., 2014). These datasets cover a range of tasks from sentiment classification and linguistic analysis to textual entailment. We also further assess FEEDER on reasoning dataset GSM8K (Cobbe et al., 2021) and semantic-parsing dataset

Table 4: Performance comparisons on text classification datasets are conducted in the fine-tuning setting, where we tune the LLMs and evaluate their few-shot inference performance. We report both the mean and variance of accuracy using 8 different seeds and 5 different permutations of n-shots. Refer to Appendix A8.2 for more extended results on datasets FPB, SST-5, TREC.

| $\Psi_{LLM}(\cdot)$ | $\mathcal{D}$ | $n$ | SUBJ | | | SST-2 | | | COLA | | |
|---|---|---|---|---|---|---|---|---|---|---|---|
| | | | RAN | SIM | DIV | RAN | SIM | DIV | RAN | SIM | DIV |
| SMA (0.3B) | $\mathcal{D}_{TRAIN}$ | 1 | 67.8 (7.2) | 83.7 (0.1) | 83.7 (0.1) | 61.3 (8.1) | 71.6 (0.2) | 71.6 (0.2) | 59.3 (5.2) | 69.4 (0.2) | 69.4 (0.2) |
| | | 2 | 69.1 (4.3) | 88.7 (0.2) | 86.9 (0.2) | 73.5 (3.2) | 75.8 (0.5) | 74.2 (0.3) | 64.1 (5.7) | 74.1 (0.2) | 74.0 (0.3) |
| | | 5 | 70.8 (5.1) | 73.3 (0.1) | 72.7 (0.2) | 74.6 (4.1) | 82.8 (0.3) | 75.3 (0.2) | 60.9 (4.6) | 76.7 (0.3) | 76.4 (0.3) |
| | | 10 | 89.2 (4.1) | 94.0 (0.2) | 91.6 (0.2) | 70.8 (2.9) | 84.5 (0.2) | 77.4 (0.2) | 70.7 (3.8) | 75.7 (0.3) | 77.6 (0.5) |
| | $\mathcal{D}_{FEEDER}$ | 1 | **93.0** (4.3) | **93.5** (1.8) | **93.5** (1.8) | **89.5** (4.3) | **88.4** (1.6) | **88.4** (1.6) | **81.5** (3.3) | **82.6** (1.4) | **82.6** (1.4) |
| | | 2 | **96.1** (3.8) | **94.1** (1.3) | **92.6** (1.2) | **92.6** (2.8) | **94.4** (0.6) | **93.8** (0.7) | **90.2** (3.8) | **91.2** (1.7) | **90.8** (0.9) |
| | | 5 | **85.7** (3.5) | **94.7** (1.5) | **94.1** (1.1) | **87.5** (4.1) | **92.5** (1.7) | **93.7** (1.7) | **87.7** (3.2) | **89.6** (2.7) | **90.0** (3.9) |
| | | 10 | **90.5** (3.3) | **95.5** (1.3) | **95.6** (1.4) | **91.9** (2.9) | **93.1** (2.1) | **89.0** (1.4) | **91.3** (3.5) | **92.4** (1.8) | **93.5** (1.9) |
| MED (0.8B) | $\mathcal{D}_{TRAIN}$ | 1 | 67.8 (7.2) | 83.7 (0.1) | 83.7 (0.1) | 61.3 (8.1) | 71.6 (0.2) | 71.6 (0.2) | 59.3 (5.2) | 69.4 (0.2) | 69.4 (0.2) |
| | | 2 | 69.1 (4.3) | 88.7 (0.2) | 86.9 (0.2) | 73.5 (3.2) | 75.8 (0.5) | 74.2 (0.3) | 64.1 (5.7) | 74.1 (0.2) | 74.0 (0.3) |
| | | 5 | 70.8 (5.1) | 73.3 (0.1) | 72.7 (0.2) | 74.6 (4.1) | 82.8 (0.3) | 75.3 (0.2) | 60.9 (4.6) | 76.7 (0.3) | 76.4 (0.3) |
| | | 10 | 89.2 (4.1) | 94.0 (0.2) | 91.6 (0.2) | 70.8 (2.9) | 84.5 (0.2) | 77.4 (0.2) | 70.7 (3.8) | 75.7 (0.3) | 77.6 (0.5) |
| | $\mathcal{D}_{FEEDER}$ | 1 | **93.0** (4.3) | **93.5** (1.8) | **93.5** (1.8) | **89.5** (4.3) | **88.4** (1.6) | **88.4** (1.6) | **81.5** (3.3) | **82.6** (1.4) | **82.6** (1.4) |
| | | 2 | **96.1** (3.8) | **94.1** (1.3) | **92.6** (1.2) | **92.6** (2.8) | **94.4** (0.6) | **93.8** (0.7) | **90.2** (3.8) | **91.2** (1.7) | **90.8** (0.9) |
| | | 5 | **85.7** (3.5) | **94.7** (1.5) | **94.1** (1.1) | **87.5** (4.1) | **92.5** (1.7) | **93.7** (1.7) | **87.7** (3.2) | **89.6** (2.7) | **90.0** (3.9) |
| | | 10 | **90.5** (3.3) | **95.5** (1.3) | **95.6** (1.4) | **91.9** (2.9) | **93.1** (2.1) | **89.0** (1.4) | **91.3** (3.5) | **92.4** (1.8) | **93.5** (1.9) |
| NEO (1.3B) | $\mathcal{D}_{TRAIN}$ | 1 | 72.7 (5.2) | 91.0 (0.1) | 91.0 (0.1) | 65.4 (4.4) | 72.5 (0.2) | 72.5 (0.2) | 61.8 (5.2) | 68.5 (0.2) | 68.5 (0.2) |
| | | 2 | 74.1 (4.3) | 93.7 (0.2) | 92.1 (0.3) | 74.5 (3.2) | 75.8 (0.4) | 76.4 (0.5) | 70.8 (5.7) | 63.9 (0.2) | 64.3 (0.4) |
| | | 5 | 71.8 (5.5) | 74.8 (0.3) | 75.8 (0.4) | 73.6 (4.1) | 77.8 (0.3) | 76.3 (0.2) | 68.7 (4.7) | 75.4 (0.8) | 74.9 (0.4) |
| | | 10 | 90.2 (4.0) | 93.6 (0.4) | 92.5 (0.4) | 72.8 (2.9) | 81.5 (0.2) | 78.8 (0.2) | 72.7 (3.4) | 76.7 (0.4) | 77.5 (0.7) |
| | $\mathcal{D}_{FEEDER}$ | 1 | **93.5** (4.3) | **94.1** (1.4) | **94.1** (1.4) | **91.2** (3.8) | **92.7** (1.5) | **92.7** (1.5) | **86.8** (3.3) | **89.6** (0.9) | **89.6** (0.9) |
| | | 2 | **95.5** (3.9) | **95.1** (1.3) | **96.6** (1.8) | **88.6** (2.4) | **93.4** (0.6) | **94.2** (0.5) | **84.2** (3.7) | **87.3** (0.7) | **89.5** (0.9) |
| | | 5 | **91.5** (3.8) | **95.7** (1.0) | **95.3** (1.4) | **89.4** (2.7) | **92.5** (1.8) | **93.7** (1.9) | **89.7** (3.2) | **92.4** (2.3) | **90.8** (1.8) |
| | | 10 | **92.8** (3.1) | **96.0** (1.4) | **94.8** (1.2) | **90.9** (2.0) | **93.6** (1.6) | **92.2** (1.8) | **89.3** (3.9) | **93.5** (1.7) | **94.4** (1.6) |

SMCALFlow (Andreas et al., 2020). For each dataset, we directly follow the official splits to obtain $\mathcal{D}_{TRAIN}$ and $\mathcal{D}_{TEST}$.

To evaluate the performance of our approach, we employed two GPT-2 variants (Radford et al., 2019): one with 335M parameters denoted as SMA, and the other with 774M parameters denoted as MED; one GPT-neo with 1.3B parameters denoted as NEO; one GPT-3 variant (Brown et al., 2020) with 6B parameters denoted as LAR; one Gemma-2 variant (Team et al., 2024) with 2B parameters denoted as GEM, one Llama 2 variant (Touvron et al., 2023) with 7B parameters denoted as LLA, and Llama 3 variant (Meta, 2024) with 8B parameters, as the LLM base.

## 5.1 EVALUATING FEEDER IN THE IN-CONTEXT LEARNING SETTING

Since our $\mathcal{D}_{FEEDER}$ works as a pre-selector, when applied in the in-context learning setting, we propose incorporating demonstration selectors into FEEDER. In other words, our evaluations follow an ablative approach, with the baseline involving the direct application of these demonstration selectors on $\mathcal{D}_{TRAIN}$. This baseline can be regarded as treating these methods both as pre-selectors and demonstration selectors. For ease of deployment, our $\mathcal{D}_{FEEDER}$ is identified using only a one-shot inference check (i.e., $K = 1$) and a single-round run (i.e., $R = 1$), unless otherwise stated.

Concretely, we conducted an evaluation of FEEDER in conjunction with following 6 selectors: (i) RAN is the random selector, which selects input demonstration randomly from the retrieval pool; (ii) SIM is the similarity-based selector (Sorensen et al., 2022; Gonen et al., 2022), which selects relevant demonstrations in terms of the cosine similarity metric over the embedding vectors generated by a sentence transformer (Reimers & Gurevych, 2019); (iii) DIV is the diversity-based selector (Ye et al., 2022), which selects similar and diverse demonstrations in terms of maximal marginal relevance (Carbonell & Goldstein, 1998); (iv) UNC is the uncertainty-based selector (Köksal et al., 2022) that conducts selections according to their uncertainty metric; (v) CLU is the clustering-based selector (Zhou et al., 2023) that searches demonstrations by clustering. (vi) LVM uses LLMs as latent variable models (Wang et al., 2024) to learn latent variables for down-streaming in-context learning. Please refer to Appendix A5.1 for detailed descriptions of the above demonstration selectors.

Experimental results regarding in-context learning performance are reported in Tables 1, 2 and 3. We also present the reduction of our FEEDER in Figure 4. Our findings are summarized as follows.

**FEEDER is an effective demonstration pre-selector (i.e., compressor) and can benefit from diverse demonstration selectors.** By combining the results from Table 1 and Figure 4, it is evident that

FEEDER enables the retention of almost half of the training samples while consistently achieving superior or comparable performance across popular demonstration selectors, including RAN, SIM, and DIV. Experimental results using UNC, CLU, and LVM as demonstration selectors are depicted in Table 2, providing additional evidence supporting the efficacy of FEEDER as a proficient data pre-selection method for in-context learning. We also evaluate the few-shot performance on more complex tasks using LLMs GEM, LAR, and LLA, with the corresponding results reported in Table 3. The table demonstrates that, even though LLMs may not perform well on these tasks, our FEEDER can consistently enhance their performance.

FEEDER **performs well with a large number of shots.** In Table 1, we can observe many cases where the LLM performance drops when the number of shots increases from 5 to 10 (e.g., SMA and MED on COLA dataset). This may be caused by the introduction of noisy and redundant shots. Our FEEDER addresses this issue by evaluating the sufficiency and necessity of each demonstration. To further verify this claim, in Appendix A9.3, we duplicate the training dataset and evaluate NEO's performance. Our results show that FEEDER minimizes the negative impact on the LLM, supporting its effectiveness in managing demonstration quality.

## 5.2 EVALUATING FEEDER IN THE FINE-TUNING SETTING

Here, we extend our FEEDER to the fine-tuning setting. As formulated in Section 2, our pre-selection and the LLM fine-tuning can be integrated into a bi-level optimization framework. Specifically, in our evaluation, we assess the performance of FEEDER by initially fine-tuning the LLM on the pre-selected $\mathcal{D}_{\text{FEEDER}}$. Subsequently, we use the tuned LLM to generate a new $\mathcal{D}_{\text{FEEDER}}$, and evaluate the LLM within the in-context learning setting, using the new $\mathcal{D}_{\text{FEEDER}}$ as the retrieval pool.

For comparison, our baseline is to initially fine-tune the LLM with $\mathcal{D}_{\text{TRAIN}}$ and then evaluate the LLM within the in-context learning setting, using $\mathcal{D}_{\text{TRAIN}}$ as the retrieval pool. Due to budget constraints, we limit our evaluation to LLMs with up to 2B parameters (i.e., SMA, MED, NEO).

Experimental results are reported in Table 4. Our findings are summarized as follows.

FEEDER **achieves substantial improvements when compared to fine-tuning with** $\mathcal{D}_{\text{TRAIN}}$**.** As illustrated in Table 4, using FEEDER sets consistently yields substantial improvements compared to using $\mathcal{D}_{\text{TRAIN}}$ for fine-tuning. This emphasizes the potential for achieving enhanced performance by utilizing a small yet high-quality dataset for fine-tuning, while simultaneously reducing computational expenses. By combining the results from Table 1 and Table 4, we can see that fine-tuning LLMs provides greater performance improvements compared to augmenting LLMs with contexts. Furthermore, our FEEDER achieves even better performance gains in the fine-tuning setting. One potential explanation is that in this scenario, fine-tuning can leverage input demonstrations more effectively than prompting can, and our high-quality FEEDER can therefore provide greater benefits.

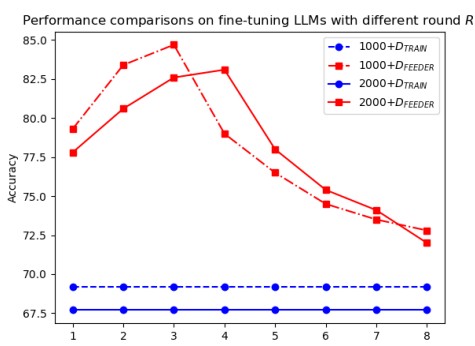

Figure 3: Performance comparisons on fine-tuning NEO with running our approximation algorithm to pre-select $\mathcal{D}_{\text{FEEDER}}$ with different run $R$. Our evaluation operates on COLA dataset in the zero-shot setting after fine-tuning on 1000 and 2000 batches.

FEEDER**'s performance first rises and then drops with increasing tree algorithm runs** $R$**.** Figure 3 visualizes the impact of employing different numbers of runs of our approximation algorithm (as described in Section 4.2) to derive $\mathcal{D}_{\text{FEEDER}}$ for fine-tuning NEO. For ease of comparison, the results of fine-tuning NEO on $\mathcal{D}_{\text{TRAIN}}$ are also included with the blue line. The observations suggest that fine-tuning with a smaller dataset with high data quality can enhance performance, but excessively reducing the dataset size may not lead to the desired outcomes. Also, it also indicates that fine-tuning LLMs on "unnecessary" data samples would not help. This trend may be summarized as a trade-off between data quantity and data quality, and similar observations are reported in (Chen et al., 2023).

We also investigate the performance of FEEDER with varying tree depths (i.e., the number of iterations $K$), which exhibits a similar trend to increasing the number of tree algorithm runs. Detailed results and discussions are provided in Appendix A9.2. These findings further verify that identifying an

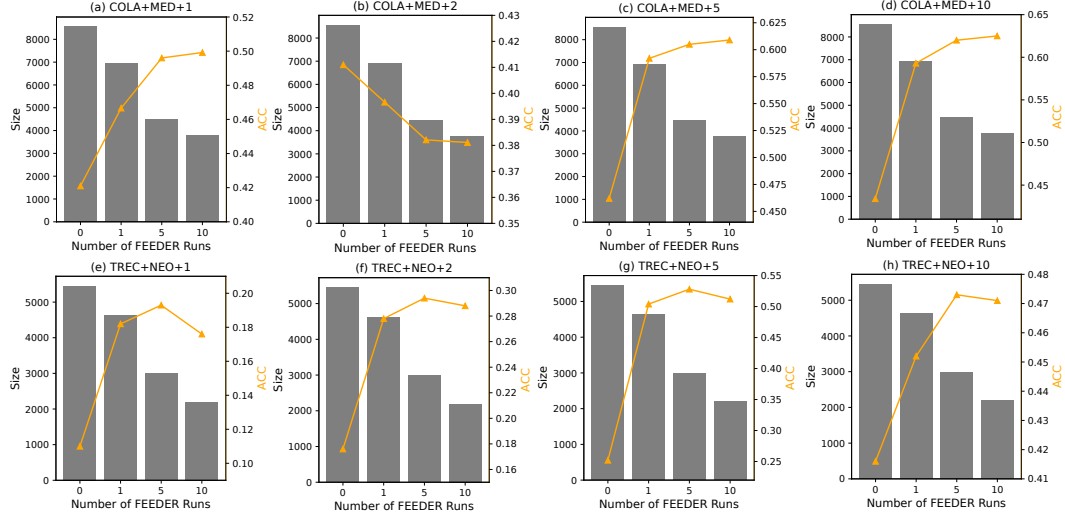

Figure 4: Performance comparisons for running our approximation algorithm to pre-select FEEDER with different runs $R$ are evaluated in terms of accuracy (denoted as ACC) with RAN as the retriever and the size of the resulting FEEDER set (denoted as Size). Each sub-figure is entitled with Dataset+LLM base+n shots.

informative subset from the training dataset-either by increasing the number of rounds or the number of iterations—can significantly enhance the performance of the LLM. However, overly narrow subsets may limit the potential performance gains.

We also provide empirical results of the time complexity associated with FEEDER in Appendix A10, and scaling up FEEDER into larger LLMs and real-world datasets in Appendix A6.

### 5.3 Case Study with Artificial Data Points Generated by LLMs

Subsequently, we conduct a case study to substantiate the central proposition of this paper: whether the assessment of the quality of demonstrations should depend on the specific LLM in use. We consider the factual error made by Google Bard in the first demo[1]. We further prompt gpt-3.5-turbo to generate 5 *sufficient and necessary statements* for the fact. We evaluate separately using these statements as a prompt to gpt-3.5-turbo, and find that either one of the generated statements is sufficient and necessary to answer the question "What took the very first pictures of a planet outside of our own solar system?" We then evaluate the performance of gpt-j-6b with the above 5 statements, and find that only the 1-st or the 5-th statement is sufficient and necessary instance to answer the above question. Combining the results of gpt-j-6b and gpt-3.5-turbo verifies one of the core insights of our paper: the evaluation of prompting a demonstration should consider the specific LLM in use. Please refer to the detailed description of prompts and outputs in Appendix A11.2.

### 6 Conclusion and Future Work

In this paper, we present a novel demonstration *pre-selector* FEEDER, designed to leverage LLMs' powerful transitivity inference capabilities to identify high-quality demonstration and provide an approximate approach for their discovery. Our experimental results showcase the significant advantages of FEEDER across diverse LLM bases in both in-context learning and fine-tuning settings. Due to budget limitations, our paper presents results only for LLMs with up to 10B parameters for in-context learning evaluation and up to 2B parameters for the fine-tuning setting. In the future, it would be valuable to explore the use of larger LLMs and extend the applications of FEEDER to areas such as data safety and data management.

---

[1] https://www.theverge.com/2023/2/8/23590864/google-ai-chatbot-bard-mistake-error-exoplanet-demo

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

## A1 Connections to Existing Approaches

### A1.1 Connections to Causality

The concepts of sufficiency and necessity have a broad application scope, especially in causality (Pearl, 1980; 2009), where sufficiency and necessity are proposed to define the causal relationship between two binary variables. Let $X$ and $Y$ denote a pair of variables. Then, the probability of sufficiency measures the capacity of setting $X = \texttt{true}$ to produce $Y = \texttt{true}$, while the probability of necessity measures the changing the value of $X$ from $X = \texttt{true}$ to $X = \texttt{false}$ would cause the value of $Y$ changing from $Y = \texttt{true}$ to $Y = \texttt{false}$

In this paper, we adopt the concepts of sufficiency and necessity in the context of demonstration selection, where we investigate whether prompting certain data points is sufficient or necessary for the given LLM to generate correct answers for input questions. For this purpose, we introduce the plugging-in operation, denoted as $\texttt{plug}(\cdot)$, to examine sufficiency, and the unplugging operation, denoted as $\texttt{unplug}(\cdot)$, to examine necessity. Both of these operations are analogous to the do operation in causality, denoted as $\texttt{do}(\cdot)$, which indicates that the system operates under the condition that certain variables are controlled by external forces. To be more specific, in our setting, the external force can be explained as follows. We have the choice to either plug in or unplug certain data points, thereby altering what is already plugged into the LLM. Our approach shares similarities with the counterfactual idea in causality, which explores hypothetical scenarios by considering what might happen if certain variables are set with different values. In our case, we investigate the impact of plugged-in data that includes data points differing from the historical (i.e., factual) setting. Notably, a significant distinction between our approach and the counterfactual setting in causality lies in the fact that we do not need to estimate "counterfactual" situations; instead, we can directly conduct evaluations.

### A1.2 Connections to Demonstration Selection

In the context of few-shot inference, a central challenge lies in selecting the appropriate training samples as extra input during inference. These samples are often referred to as demonstrations or prompts (Levy et al., 2022; Liu et al., 2021; Dong et al., 2022). The underlying assumption is that the training dataset serves as a support set (Yosida, 2012) for test samples. Previous studies (Wang et al., 2022; Rubin et al., 2021) have demonstrated that introducing similar training samples can enhance the performance of LLMs on test instances. (Gao et al., 2023) enhances these approaches by retrieving candidates whose ground label lies in top-2 zero-shot predictions. However, as pointed out in (Levy et al., 2022), existing methods often treat each data point in isolation, neglecting the collective impact of multiple data points. For instance, retrievers based on similarity metrics may select redundant data points together. To address this limitation, (Levy et al., 2022) proposes to consider the diversity among the data points, to avoid the case where too "similar" data points are selected together. Further, (Rubin et al., 2021) trains an LLM as a contrastive scorer as well as a demonstration referrer, and (Li et al., 2023) advances this framework through unified training across various datasets.

In this paper, we present a novel perspective, asserting that the quality of demonstrations is contingent on the specific LLM in use. Namely, a high-quality demonstration for one LLM might be deemed low-quality for another. Leveraging this insight, we introduce sufficiency and necessity as new set-level metrics. Our approach offers several advantages: Firstly, sufficiency and necessity measure the quality of data points based on the specific LLM, in contrast to generic similarity and diversity metrics. Secondly, our proposed sufficiency and necessity extend to the set level, enabling the consideration of data points as a cohesive whole. In our framework, "similarity" is akin to "sufficiency" signifying that plugging in data points can enhance LLM performance, while "diversity" is akin to "necessity" suggesting that each data point should play an indispensable role.

Recent studies (Xia et al., 2024; Marion et al., 2023) focus on mining training examples for fine-tuning on specific tasks, while (Wang et al., 2024) extends this idea to in-context learning. Unlike these approaches, which use LLMs to select demonstrations tailored to specific test datasets, our work leverages LLMs as demonstration pre-selectors, identifying a core subset of the training data that remains independent of the test datasets, thus eliminating the need for re-computation across different test datasets.

### A1.3 CONNECTIONS TO CORE SET SELECTION

Core-set selection (Feldman, 2020; Guo et al., 2022), a longstanding problem in machine learning, focuses on identifying a subset of the most informative training samples. Previous research (Dor et al., 2020) has surveyed and evaluated state-of-the-art approaches for models like BERT (Devlin et al., 2018), encompassing strategies such as random sampling, uncertainty-sampling (using entropy metric) (Lewis, 1995; Gal & Ghahramani, 2016) and diversity sampling (using diversity metric) (Gissin & Shalev-Shwartz, 2019).

FEEDER, in contrast to these prior papers mainly using active learning, is designed to select core sets, which can serve as additional input contexts (i.e., in-context learning setting) or be used for fine-tuning LLMs (i.e., fine-tuning setting). FEEDER defines "informative training samples" as those samples that specifically enhance the LLM's performance on a given task.

### A1.4 CONNECTIONS TO PROMPT OPTIMIZATION

Prompting provides a natural way for humans to interact with; and due to its flexibility, prompting has been widely used as a genre method for various natural language processing tasks (Schick & Schütze, 2020; Brown et al., 2020; Sanh et al., 2021). However, using prompting effectively with LLMs requires careful design, either done manually (Reynolds & McDonell, 2021) or automatically (Gao, 2021; Shin et al., 2020), as LLMs do not interpret prompts in the same way humans do (Webson & Pavlick, 2021; Lu et al., 2021). While numerous successful methods (Liu et al., 2021; Lester et al., 2021; Qin & Eisner, 2021) for prompt tuning rely on optimizing a continuous space through gradient-based techniques, this approach becomes impractical as many powerful LLMs are only accessible through APIs that may not offer gradient access.

Our FEEDER approach can be seen as a discrete pre-search method for prompts, distinct from existing methods for prompt generation (Gao, 2021; Ben-David et al., 2021), prompt scoring (Davison et al., 2019), and prompt paraphrasing (Jiang et al., 2020; Yuan et al., 2021), which aim to optimize instructions by directly searching the natural language hypothesis space. Instead, our approach leverages the causal dependencies among candidate demonstrations, focusing on searching for the most informative demonstrations as prompts, in terms of sufficiency and necessity.

## A2 A FAMILY OF ANALYSIS ON DATA RELATIONSHIPS

We begin by introducing some key notations used in the paper.

Let $X, C$ denote variables for the input and the context (i.e., previously plugged-in demonstrations). We use $Y$, a boolean variable, to denote whether the output to the input is correct. Concretely, we use $Y_{\boldsymbol{x}} = 1$ to denote $Y = 1 | X = \boldsymbol{x}$, meaning that the LLM generates the correct output to the input $\boldsymbol{x}$. Similarly, $Y_{\boldsymbol{x}} = 0$, equivalent to $Y = 0 | X = \boldsymbol{x}$, indicates that the LLM produces the incorrect output to $\boldsymbol{x}$. For clarity, we introduce $S$, a variable to record the original status of the LLM before *new* plug-in and unplug operations (denoted as plug($\cdot$) and unplug($\cdot$) respectively), e.g., $C = ((\boldsymbol{x}, \boldsymbol{y})), S = (Y_{\boldsymbol{x}} = 1)$ means that without plugging-in any new data or unplugging any plugged-in data, the plugged-in data is $(\boldsymbol{x}, \boldsymbol{y})$ and the LLM's performance is $Y_{\boldsymbol{x}} = 1$.

### A2.1 DATA RELATIONSHIPS ON INSTANCE LEVEL

Here, two instances are considered, represented as $(\boldsymbol{x}_n, \boldsymbol{y}_n)$ and $(\boldsymbol{x}_m, \boldsymbol{y}_m)$.

Sufficiency relationship is introduce to assess whether plugging in one data point is sufficient to enable the LLM to generate the correct output for the other one. Formally, the sufficiency relationship is defined as follows:

**Definition 4** (Instance-level Sufficiency). *Given tuple* $(X, Y, C, S)$*, data point* $(\boldsymbol{x}_n, \boldsymbol{y}_n)$ *is sufficient for* $(\boldsymbol{x}_m, \boldsymbol{y}_m)$*, if the following equation holds:*

$$Y_{\boldsymbol{x}_m} = 1 | \texttt{plug}((\boldsymbol{x}_n, \boldsymbol{y}_n)); C = \emptyset, S = (Y_{\boldsymbol{x}_m} = 0). \tag{9}$$

*It means that when plugging in* $(\boldsymbol{x}_n, \boldsymbol{y}_n)$*, it would correct the LLM's answer to* $\boldsymbol{x}_m$*.*

**Example A1.** Let $\boldsymbol{x}_m, \boldsymbol{x}_n$ be *Which country does Sherlock Holmes live?* and *Which city does Sherlock Holmes live?* Then, after informing the LLM of the correct answer of $\boldsymbol{x}_n$ (e.g., $\boldsymbol{y}_n$ is *Sherlock Holmes lives in London*), the LLM can deduce the correct answer of $\boldsymbol{x}_m$ (e.g., $\boldsymbol{y}_m$ is *Sherlock Holmes lives in the United Kingdom*). In this case, the LLM is using the city where Sherlock Holmes lives to infer the country in which he lives.

Necessity relationship is introduced to assess whether the presence of one plugged-in data point is necessary for preserving the correct output in relation to another. Formally, this is expressed as:

**Definition 5** (Instance-level Necessity). *Given tuple $(X, Y, C, S)$, we say that data point $(\boldsymbol{x}_n, \boldsymbol{y}_n)$ is necessary for $(\boldsymbol{x}_m, \boldsymbol{y}_m)$, if the following equation holds:*

$$Y_{\boldsymbol{x}_m} = 0 | \texttt{unplug}((\boldsymbol{x}_n, \boldsymbol{y}_n)); C = ((\boldsymbol{x}_n, \boldsymbol{y}_n)), S = (Y_{\boldsymbol{x}_m} = 1). \tag{10}$$

*It means that before unplugging $(\boldsymbol{x}_n, \boldsymbol{y}_n)$, the LLM's answer to $\boldsymbol{x}_m$ is correct. However, when we do unplug $(\boldsymbol{x}_n, \boldsymbol{y}_n)$, it causes the LLM to offer an incorrect output to $\boldsymbol{x}_m$.*

**Example A2.** Consider $\boldsymbol{x}_m$ as *Which city does Sherlock Holmes live?* and $\boldsymbol{x}_n$ as *What is the detailed address of Sherlock Holmes lives?*. Assume the LLM has no prior knowledge about Sherlock Holmes until the introduction of the plugged-in data $(\boldsymbol{x}_n, \boldsymbol{y}_n)$, where $\boldsymbol{y}_n$ is *221B Baker Street, London*. After plugging in $(\boldsymbol{x}_n, \boldsymbol{y}_n)$, the LLM is capable of generating the correct output $\boldsymbol{y}_m$ (e.g., *Sherlock Holmes lives in London*) in response to $\boldsymbol{x}_m$. If we were to unplug $(\boldsymbol{x}_n, \boldsymbol{y}_n)$, the LLM would provide an incorrect output for $\boldsymbol{x}_m$, such as *Sherlock Holmes lives in New York*.

In an ideal scenario, ensuring optimal LLM performance entails the extraction of data points that are both sufficient and necessary.

**Definition 6** (Instance-level Sufficiency and Necessity). *Given tuple $(X, Y, C)$, we say that data point $(\boldsymbol{x}_n, \boldsymbol{y}_n)$ is both sufficient and necessary for $(\boldsymbol{x}_m, \boldsymbol{y}_m)$, if the following equation holds:*

$$\begin{aligned} &\Big( Y_{\boldsymbol{x}_m} = 1 | \texttt{plug}((\boldsymbol{x}_n, \boldsymbol{y}_n)); C = \emptyset \Big) \\ &\wedge \Big( Y_{\boldsymbol{x}_m} = 0 | \texttt{unplug}((\boldsymbol{x}_n, \boldsymbol{y}_n)); C = ((\boldsymbol{x}_n, \boldsymbol{y}_n)) \Big), \end{aligned} \tag{11}$$

*which indicates that plugging in data point $(\boldsymbol{x}_n, \boldsymbol{y}_n)$ can respond to the LLM's answering $\boldsymbol{x}_m$ in both ways. We omit $S$ here, because we can derive the original status of the necessary instance based on the condition of the sufficiency instance.*

We further demonstrate that neither of the aforementioned quantities (i.e., sufficiency and necessity) is adequate for determining the other, indicating that they are not entirely independent. This is illustrated in the following lemma.

**Lemma 1** (Connection between Sufficiency and Necessity). *Supposing that we only consider using the data point $(\boldsymbol{x}_n, \boldsymbol{y}_n)$ as the plug in data, and only care about the LLM's performance regarding the input question $\boldsymbol{x}_m$, then overall there are only two situations here: (i) $(\boldsymbol{x}_n, \boldsymbol{y}_n)$ is plugged-in, and (ii) $(\boldsymbol{x}_n, \boldsymbol{y}_n)$ is not plugged-in. Based on the above assumption, we re-write (i) as plugging-in $(\boldsymbol{x}_n, \boldsymbol{y}_n)$ when there is no plugged-in data (i.e., $\texttt{plug}((\boldsymbol{x}_n, \boldsymbol{y}_n)); C = \emptyset$, and re-write (ii) as unplugging $(\boldsymbol{x}_n, \boldsymbol{y}_n)$ when there is plugged-in data $(\boldsymbol{x}_n, \boldsymbol{y}_n)$ (i.e., $\texttt{unplug}((\boldsymbol{x}_n, \boldsymbol{y}_n)); C = ((\boldsymbol{x}_n, \boldsymbol{y}_n)))$. For convenience, we use $E^*$ and $E$ to denote (i) and (ii) respectively; and we use $Y^*$ and $Y$ to denote $Y_{\boldsymbol{x}_1} = 1$ and $Y_{\boldsymbol{x}_1} = 0$. Then, we have: $E^* \vee E = \texttt{true}$, $E^* \wedge E = \texttt{false}$, $Y^* \vee Y = \texttt{true}$, $Y^* \wedge Y = \texttt{false}$.*

*We define $\texttt{PS}$ as the probability of being sufficient as:*

$$\begin{aligned} \texttt{PS} :=& \Pr\Big( Y_{\boldsymbol{x}_m} = 1 | \texttt{plug}((\boldsymbol{x}_n, \boldsymbol{y}_n)); C = \emptyset \Big) \\ =& \Pr(Y^* | E^*). \end{aligned} \tag{12}$$

*We define $\texttt{PN}$ as the probability of being necessary as:*

$$\begin{aligned} \texttt{PN} :=& \Pr\Big( Y_{\boldsymbol{x}_m} = 0 | \texttt{unplug}((\boldsymbol{x}_n, \boldsymbol{y}_n)); C = ((\boldsymbol{x}_n, \boldsymbol{y}_n)) \Big) \\ =& \Pr(Y | E). \end{aligned} \tag{13}$$

*We further define $\texttt{PNS}$ as the probability of being sufficient and necessary as:*

$$\texttt{PNS} := \Pr(Y^* | E^*, Y | E). \tag{14}$$

*Then,* PS, PN, PSN *satisfy the following relationship:*

$$\text{PSN} = \Pr(Y, E) \cdot \text{PS} + \Pr(Y^*, E^*) \cdot \text{PN}. \tag{15}$$

*Proof.* Based on the earlier delineation of $Y^*$, $Y$, $E^*$, and $E$, we can express:

$$Y^*|E^* \wedge Y|E = (Y^*|E^* \wedge Y|E) \wedge (E \vee C^*)$$
$$= (Y^*|E^* \wedge Y \wedge E) \vee (Y|E \wedge Y^* \wedge E^*). \tag{16}$$

Taking probabilities on both sides and using the disjointedness of $E^*$ and $E$, we have:

$$\begin{aligned}
\text{PSN} &= \Pr(Y^*|E^*, Y|E) \\
&= \Pr(Y|E, Y^*, E^*) + \Pr(Y^*|E^*, Y, E) \\
&= \Pr(Y, E) \cdot \text{PS} + \Pr(Y^*, E^*) \cdot \text{PN}.
\end{aligned} \tag{17}$$

$\square$

## A2.2 DATA RELATIONSHIPS ON SET LEVEL

We extend Definitions 4 and 5 to the set level as:

**Definition 7** (Set-level Sufficiency). *Given tuple $(X, Y, C, S)$, the input set $\mathcal{D}_{\text{IN}}$ is sufficient for output set $\mathcal{D}_{\text{OUT}}$, if the following equation holds:*

$$Y_{(\{\boldsymbol{x}_n|\boldsymbol{x}_n \in \mathcal{D}_{\text{OUT}}\})} = \mathbf{1}_{|\mathcal{D}_{\text{OUT}}|}|\text{plug}(\mathcal{D}_{\text{IN}}); C = \emptyset, S = (Y_{(\{\boldsymbol{x}_n|\boldsymbol{x}_n \in \mathcal{D}_{\text{OUT}}\})} \neq \mathbf{1}_{|\mathcal{D}_{\text{OUT}}|}). \tag{18}$$

$\mathbf{1}_{|\mathcal{D}_{\text{OUT}}|}$ *denotes* $\mathbf{1}_{|\mathcal{D}_{\text{OUT}}|}$*-dimensional vectors whose elements are all 1s. It indicates that when plugging in $\mathcal{D}_{\text{IN}}$, it guarantees that the LLM's output to any input question in $\mathcal{D}_{\text{OUT}}$ is correct.*

**Definition 8** (Set-level Necessity). *Given tuple $(X, Y, C, S)$, the input set $\mathcal{D}_{\text{IN}}$ is necessary for output set $\mathcal{D}_{\text{OUT}}$, if the following equation holds:*

$$Y_{(\{\boldsymbol{x}_n|\boldsymbol{x}_n \in \mathcal{D}_{\text{OUT}}\})} \neq \mathbf{1}_{|\mathcal{D}_{\text{OUT}}|}|\text{unplug}(\mathcal{D}'_{\text{IN}}); C = \mathcal{D}_{\text{IN}}, S = (Y_{(\{\boldsymbol{x}_n|\boldsymbol{x}_n \in \mathcal{D}_{\text{OUT}}\})} = \mathbf{1}_{|\mathcal{D}_{\text{OUT}}|}), \tag{19}$$

*where $\mathcal{D}'_{\text{IN}}$ can be any subset of $\mathcal{D}_{\text{IN}}$. $\mathbf{1}_{|\mathcal{D}_{\text{OUT}}|}$ denotes $\mathbf{1}_{|\mathcal{D}_{\text{OUT}}|}$-dimensional vectors whose elements are all 1s. It means that before unplugging any subset of $\mathcal{D}_{\text{IN}}$, there is plugged-in data $\mathcal{D}_{\text{IN}}$ and the LLM's output to any input in $\mathcal{D}_{\text{OUT}}$ is correct. When we unplug any subset of $\mathcal{D}_{\text{IN}}$, then it would cause the LLM's output to at least one input in $\mathcal{D}_{\text{OUT}}$ to be incorrect.*

From the above description, when we refer to a set as a sufficient set, we are stating that the collective set of data points is sufficient. On the other hand, when we characterize a set as a necessary set, we mean that each individual data point within the set is necessary.

**Example A3.** Let $\mathcal{D}_{\text{OUT}} = \{(\boldsymbol{x}_m, \boldsymbol{y}_m)\}$ and $\mathcal{D}_{\text{IN}} = \{(\boldsymbol{x}_i, \boldsymbol{y}_i), (\boldsymbol{x}_j, \boldsymbol{y}_j)\}$. We assign $\boldsymbol{x}_m$ and $\boldsymbol{y}_m$ as *Which country does Sherlock Holmes live?* and *Sherlock Holmes lives in the United Kingdom.* Let $\boldsymbol{x}_i$ and $\boldsymbol{y}_i$ denote *Which street does Sherlock Holmes live?* and *Baker street.* We assign $\boldsymbol{x}_j$ and $\boldsymbol{y}_j$ as *Where is Baker street?* and *Bake street is located in London.* Supposing that the LLM does not know that Bake Street is located in the United Kingdom, then solely plugging in either $(\boldsymbol{x}_i, \boldsymbol{y}_i)$ or $(\boldsymbol{x}_j, \boldsymbol{y}_j)$ is not sufficient for the LLM to get the right answer to the input question $\boldsymbol{x}_m$. In this regard, it is easy to derive that $\mathcal{D}_{\text{IN}}$ is both a sufficient and necessary set for $\mathcal{D}_{\text{OUT}}$ when both (i) plugging in $\mathcal{D}_{\text{IN}}$ is sufficient to maintain the right answer for $\mathcal{D}_{\text{OUT}}$; and (ii) unplugging any subset of $\mathcal{D}_{\text{IN}}$ can not maintain the right answer for $\mathcal{D}_{\text{OUT}}$, are satisfied.

## A2.3 FEEDER SET

Next, we explore the problem of defining a subset within the given dataset $\mathcal{D}_{\text{TRAIN}}$ that is both sufficient and necessary to represent $\mathcal{D}_{\text{TRAIN}}$. This subset is termed FEEDER (FEw yet Essential DEmonstRations).

**Definition 9** (FEEDER Set). *Given tuple $(X, Y, C, S)$ and $\mathcal{D}_{\text{TRAIN}}$, a subset of $\mathcal{D}_{\text{TRAIN}}$, is considered as a* FEEDER *set (denoted as $\mathcal{D}_{\text{FEEDER}}$), if the following conditions are satisfied:*

*(i)* $Y_{(\boldsymbol{x}_1 \ldots, \boldsymbol{x}_N)} = \mathbf{1}_N|\text{plug}(\mathcal{D}_{\text{FEEDER}}); C = \emptyset, S = (Y_{(\boldsymbol{x}_1 \ldots, \boldsymbol{x}_N)} \neq \mathbf{1}_N)$ *holds.*

*(ii)* $Y_{(\boldsymbol{x}_1 \ldots, \boldsymbol{x}_N)} \neq \mathbf{1}_N|\text{unplug}(\mathcal{D}'_{\text{FEEDER}}); C = \mathcal{D}_{\text{TRAIN}}, S = (Y_{(\boldsymbol{x}_1 \ldots, \boldsymbol{x}_N)} = \mathbf{1}_N)$ *holds for any subset of $\mathcal{D}_{\text{FEEDER}}$ (denoted as $\mathcal{D}'_{\text{FEEDER}}$).*

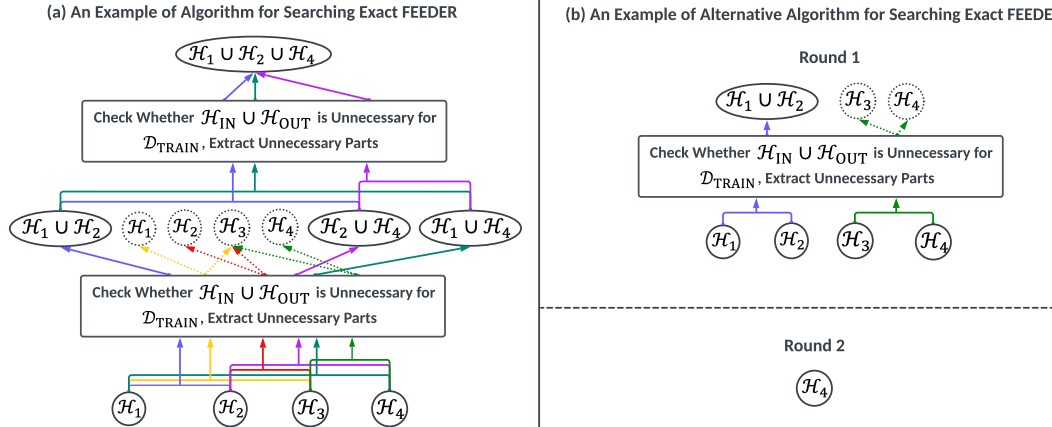

Figure 5: An illustrated example of our algorithm for deriving an exact FEEDER set. As shown in (a), we check the necessity of the conjunction of each pair of nodes, and we do not remove them from $\mathscr{H}$; instead, we assign MAINTAIN signals to newly generated nodes and the node with the maximum size, and those nodes without MAINTAIN signals, circled with dashed lines, would be removed from $\mathcal{H}$.. In (b), we propose an alternative algorithm by removing nodes after checking the necessity, and we repeat the above process for multiple rounds, at the beginning of each round, we unplug all the previously selected data points. The repeat should stop until there is no or only one node in $\mathscr{H}_0$ (i.e., $\mathcal{H}_4$), and therefore, the result in (b) is $\mathcal{H}_1 \cup \mathcal{H}_2 \cup \mathcal{H}_4$, same as the result in (a).

$1_N$ denotes $N$-dimensional vectors whose elements are all 1s. (i) and (ii) respectively imply that plugging in $\mathcal{D}_{\text{FEEDER}}$ is sufficient and necessary to maintain the LLM generating correct output.

**Example A4.** If we merge $\mathcal{D}_{\text{IN}}$ and $\mathcal{D}_{\text{OUT}}$ exemplified in Example A3 into one set $\mathcal{D}$, namely let $\mathcal{D} = \mathcal{D}_{\text{IN}} \cup \mathcal{D}_{\text{OUT}}$, then in this case, it is easy to derive that $\mathcal{D}_{\text{IN}}$ is a FEEDER set (denoted as $\mathcal{D}_{\text{FEEDER}}$) for $\mathcal{D}$.

---

**Algorithm 2:** Approximation Algorithm for FEEDER

---

**Input:** Training dataset $\mathcal{D}_{\text{TRAIN}}$.
**Output:** An approximated FEEDER set $\widetilde{\mathcal{D}}_{\text{FEEDER}}$.
Initialize $k = 1$.
Initialize $\mathscr{W}_0 = \{\mathcal{W}_n = \{(\boldsymbol{x}_n, \boldsymbol{y}_n)\} | (\boldsymbol{x}_n, \boldsymbol{y}_n) \in \mathcal{D}_{\text{TRAIN}}\}$.
**repeat**
    **for** *each pair* $(\mathcal{W}_i, \mathcal{W}_j)$ *where* $\mathcal{W}_i, \mathcal{W}_j \in \mathscr{W}_{k-1}$ **do**
        Check $Y_{(\{\boldsymbol{x}_n | \boldsymbol{x}_n \in \mathcal{W}_j\})} = 1_{|\mathcal{W}_j|} | \text{plug}(\mathcal{W}_i); C, S$ (a), where $C = \emptyset$ and $S$ can be any value.
        Check $Y_{(\{\boldsymbol{x}_n | \boldsymbol{x}_n \in \mathcal{W}_i\})} = 1_{|\mathcal{W}_i|} | \text{plug}(\mathcal{W}_j); C, S$ (b), where $C = \emptyset$ and $S$ can be any value.
        **Case I** (Both (a) and (b) hold), if $|\mathcal{W}_i| \geq |\mathcal{W}_j|$, append $\mathcal{W}_j$ to $\mathscr{W}_k$; otherwise, append $\mathcal{W}_i$ to $\mathscr{W}_k$.
        **Case II** (Either one of (a) and (b) holds), if (a) holds, append $\mathcal{W}_i$ to $\mathscr{W}_k$; otherwise, append $\mathcal{W}_j$ to $\mathscr{W}_k$.
        **Case III** (Neither (a) nor (b) holds), append $\mathcal{W}_i \cup \mathcal{W}_j$ to $\mathscr{W}_k$.
        Remove $\mathcal{W}_i, \mathcal{W}_j$ from $\mathscr{W}_{k-1}$, i.e., $\mathscr{W}_{k-1} = \mathscr{W}_{k-1} - \{\mathcal{W}_i, \mathcal{W}_j\}$.
    **end**
    **if** $|\mathscr{W}_{k-1}| = 1$ **then**
        Append only element in $\mathscr{W}_{k-1}$ to $\mathscr{W}_k$.
    **end**
    Grow tree from bottom to top via $k = k + 1$.
**until** $|\mathscr{W}_k| = 1$, *and we assume the current iteration is* $K$;
Let $\mathcal{W}_{\text{SUFFICIENT}}$ denote only one element (i.e. the root node) in $\mathscr{W}_K$.
Assign $\widetilde{\mathcal{D}}_{\text{FEEDER}}$ as $\mathcal{W}_{\text{SUFFICIENT}}$, i.e., $\mathcal{D}_{\text{OUT}} = \mathcal{W}_{\text{SUFFICIENT}}$.

---

---

**Algorithm 3:** Exact Algorithm for FEEDER

---

**Input:** Training dataset $\mathcal{D}_{\texttt{TRAIN}}$.

**Output:** An exact FEEDER set $\widetilde{\mathcal{D}}_{\texttt{FEEDER}}$.

Initialize $k = 1$.

Initialize $\mathscr{H}_0 = \emptyset$.

**for** *each instance* $(\boldsymbol{x}_n, \boldsymbol{y}_n) \in \mathcal{D}_{\texttt{TRAIN}}$ **do**

    Check $Y_{(\{\boldsymbol{x}_{n'} | \boldsymbol{x}_{n'} \in \mathcal{D}_{\texttt{TRAIN}}\})} = \mathbf{1}_{|\mathcal{D}_{\texttt{TRAIN}}|} | \texttt{unplug}((\boldsymbol{x}_n, \boldsymbol{y}_n)); C, S$ (a), $C = \mathcal{D}_{\texttt{TRAIN}}$,

     $S = (Y_{(\{\boldsymbol{x}_{n'} | \boldsymbol{x}_{n'} \in \mathcal{D}_{\texttt{TRAIN}}\})} = \mathbf{1}_{|\mathcal{D}_{\texttt{TRAIN}}|})$.

    If (a) holds, let $\mathcal{H}_n = \{(\boldsymbol{x}_n, \boldsymbol{y}_n)\}$ and append $\mathcal{H}_n$ to $\mathscr{H}_0$.

**end**

**repeat**

    **for** *each pair* $(\mathcal{H}_i, \mathcal{H}_j)$ *where* $\mathcal{H}_i, \mathcal{H}_j \in \mathscr{H}_{k-1}$ **do**

        Check $Y_{(\{\boldsymbol{x}_n | \boldsymbol{x}_n \in \mathcal{D}_{\texttt{TRAIN}}\})} = \mathbf{1}_{|\mathcal{D}_{\texttt{TRAIN}}|} | \texttt{unplug}(\mathcal{H}_i \cup \mathcal{H}_j); C, S$ (b), where $C = \mathcal{D}_{\texttt{TRAIN}}$ and

         $S = (Y_{(\{\boldsymbol{x}_{n'} | \boldsymbol{x}_{n'} \in \mathcal{D}_{\texttt{TRAIN}}\})} = \mathbf{1}_{|\mathcal{D}_{\texttt{TRAIN}}|})$.

        If (b) holds, generate a new node $\mathcal{H}_i \cup \mathcal{H}_j$, append it to $\mathscr{H}_k$, and assign $\mathcal{H}_i \cup \mathcal{H}_j$ with

         MAINTAIN signals; otherwise, append $\mathcal{H}_i$ and $\mathcal{H}_j$ to $\mathscr{H}_k$.

    **end**

    Assign $\mathcal{H}_{\texttt{MAX}} = \arg\max_{\mathcal{H}. \in \mathscr{H}_k} |\mathcal{H}.|$ with MAINTAIN signal.

    Remove the nodes without MAINTAIN signals in $\mathscr{H}_k$.

    Grow tree from bottom to top via $k = k + 1$.

**until** $|\mathscr{H}_k| = 1$ *where we assume the iteration is* $K$;

Let $\mathcal{H}_{\texttt{UNNCESSARY}}$ denote only one element (i.e. the root node) in $\mathscr{H}_K$.

Assign $\widetilde{\mathcal{D}}_{\texttt{FEEDER}}$ as removing $\mathcal{H}_{\texttt{UNNCESSARY}}$ from $\mathcal{D}_{\texttt{TRAIN}}$, i.e., $\widetilde{\mathcal{D}}_{\texttt{FEEDER}} = \mathcal{D}_{\texttt{TRAIN}} - \mathcal{H}_{\texttt{UNNECESSARY}}$.

---

## A3    APPROXIMATED EXTRACTION OF FEEDER

**Definition 10** (Transitivity Inference). *Noted by (Jang & Lukasiewicz, 2023) that LLMs excel at transitive inference. We assume that sufficiency is transitive among sets. Formally, for any three sets, denoted as $\mathcal{D}_{\texttt{A}}$, $\mathcal{D}_{\texttt{B}}$, and $\mathcal{D}_{\texttt{C}}$, if $\mathcal{D}_{\texttt{A}}$ is a sufficient set of $\mathcal{D}_{\texttt{B}}$ and $\mathcal{D}_{\texttt{B}}$ is a sufficient set of $\mathcal{D}_{\texttt{C}}$, then we can conclude that $\mathcal{D}_{\texttt{A}}$ is a sufficient set of $\mathcal{D}_{\texttt{C}}$.*

We also establish case studies in Appendix A11.1 to verify the feasibility of the above assumption.

For convenience, we use $\mathcal{D}_{\texttt{IN}} = \{(\boldsymbol{x}_n, \boldsymbol{y}_n)\}_{n=1}^{N_{\texttt{IN}}}$ to denote the input set for our tree algorithm, and we use $\mathcal{D}_{\texttt{OUT}}$ to denote the corresponding output. The tree expands from the bottom to the top. We use the variable $K$ to represent the depth of these trees, which corresponds to the number of iterations. To be more specific, we use $k = 1, 2, \ldots, K$ to refer to each $k$-th iteration, and during each $k$-th iteration, we generate the $(k + 1)$-th layer of the tree.

Concretely, we leverage the transitivity of sufficiency to build the tree, where each node is a set of samples. Formally, we denote $\mathscr{W}_k$ as the set of nodes after the $k$-th iteration. We initialize $\mathscr{W}_0$ by assigning all the candidate samples in $\mathcal{D}_{\texttt{IN}}$ as the bottom nodes:

$$\mathscr{W}_0 := \{\mathcal{W}_n := \{(\boldsymbol{x}_n, \boldsymbol{y}_n)\} | (\boldsymbol{x}_n, \boldsymbol{y}_n) \in \mathcal{D}_{\texttt{IN}}\}. \tag{20}$$

During each $k$-th iteration, we generate $\mathscr{W}_k$ by examining the sufficiency relationship between every pair of nodes, denoted as $\mathcal{W}_i, \mathcal{W}_j \in \mathscr{W}_{k-1}$. In this evaluation, we assess whether the following equation holds true by assigning $\mathcal{W}_i$ and $\mathcal{W}_j$ as $\mathcal{W}_{\texttt{IN}}$ and $\mathcal{W}_{\texttt{OUT}}$, or vice versa.

$$Y_{(\{\boldsymbol{x}_n | \boldsymbol{x}_n \in \mathcal{W}_{\texttt{OUT}}\})} = \mathbf{1}_{|\mathcal{W}_{\texttt{OUT}}|} | \texttt{plug}(\mathcal{W}_{\texttt{IN}}); C = \emptyset, S, \tag{21}$$

where $S$ is loosened to allow for any value. $\mathbf{1}_{|\mathcal{W}_{\texttt{OUT}}|}$ is $\mathbf{1}_{|\mathcal{W}_{\texttt{OUT}}|}$-dimensional vectors whose elements are all 1s. It signifies that plugging in $\mathcal{W}_{\texttt{IN}}$ is sufficient for the LLM to generate the correct output to any input in $\mathcal{W}_{\texttt{OUT}}$. In other words, once we have $\mathcal{W}_{\texttt{IN}}$ included in the plugged-in context, it is unnecessary to further include $\mathcal{W}_{\texttt{OUT}}$. Formally, we can derive the following equation from Eq. (21):

$$Y_{(\{\boldsymbol{x}_n | \boldsymbol{x}_n \in \mathcal{W}_{\texttt{OUT}}\})} = \mathbf{1}_{|\mathcal{W}_{\texttt{OUT}}|} | \texttt{unplug}(\mathcal{W}_{\texttt{OUT}}); C = (\mathcal{W}_{\texttt{IN}} \cup \mathcal{W}_{\texttt{OUT}}), S, \tag{22}$$

where $S$ is loosened to be any value. Concretely, there are three possible scenarios by examining each pair of nodes in $\mathscr{W}_{k-1}$: (i) If both $\mathcal{W}_i$ and $\mathcal{W}_j$ are sufficient sets for each other, then we select

the one with fewer elements to append to $\mathcal{W}_k$. (ii) If only one of $\mathcal{W}_i$ and $\mathcal{W}_j$ is a sufficient set for the other, then we append the sufficient set to $\mathcal{W}_k$. (iii) If neither $\mathcal{W}_i$ nor $\mathcal{W}_j$ is a sufficient set, we append $\mathcal{W}_i \cup \mathcal{W}_j$ to $\mathcal{W}_k$. After performing the above calculations for each pair of nodes, we remove them from $\mathscr{W}_{k-1}$. When there is only one element left in $\mathscr{W}_{k-1}$, it is directly appended to $\mathscr{W}_k$. This process continues until $\mathscr{W}$ contains only one element, which is denoted as $\mathcal{W}_{\texttt{SUFFICIENT}} \in \mathscr{W}_K$. We then assign $\mathcal{D}_{\texttt{OUT}}$ as $\mathcal{D}_{\texttt{OUT}} = \mathcal{W}_{\texttt{SUFFICIENT}}$.

The time complexity of running the above tree algorithm for one round is $O(\log_2^{|\mathcal{D}_{\texttt{IN}}|})$.

To effectively remove the unnecessary part, we can repeat the above process for multiple rounds by using the output of the previous round (i.e., $\mathcal{D}_{\texttt{OUT}}$) as the input for the subsequent round (i.e., $\mathcal{D}_{\texttt{IN}}$). Our tree algorithm can also maintain the remaining set to be sufficient to represent the entire $\mathcal{D}_{\texttt{TRAIN}}$, as verified in the following proposition.

**Proposition 2** ($\widetilde{\mathcal{D}}_{\texttt{FEEDER}}$ obtained by Algorithm 2 is an Approximation of $\mathcal{D}_{\texttt{FEEDER}}$). *If we successively apply Algorithm 2 on $\mathcal{D}_{\texttt{TRAIN}}$ for multiple rounds to obtain a subset (denoted as $\widetilde{\mathcal{D}}_{\texttt{FEEDER}}$), then $\widetilde{\mathcal{D}}_{\texttt{FEEDER}}$ is sufficient to represent $\mathcal{D}_{\texttt{TRAIN}}$.*

*Proof.* In the tree generation process, each parent node is established as a sufficient set for every leaf node within the tree. More precisely, as shown in **Case I**, **Case II** and **Case III** of Algorithm 2, three scenarios exist for creating a parent node for each pair of leaf nodes. In cases (i) and (ii), the parent node corresponds to the leaf node which serves as a sufficient set for the other node. In case (iii), the parent node results from the conjunction of two leaf nodes, inherently forming a sufficient set capable of representing either of the two leaf nodes.

According to our assumption of the sufficiency transitivity, for each data point in $\mathcal{D}_{\texttt{TRAIN}}$, the root node of the tree is a sufficient set for each leaf node in the tree. Formally, we have:

$$Y_{\{\boldsymbol{x}_n | \boldsymbol{x}_n \in \mathcal{D}_{\texttt{TRAIN}}\}} = \mathbf{1}_{|\mathcal{D}_{\texttt{TRAIN}}|} | \texttt{plug}(\widetilde{\mathcal{D}}_{\texttt{FEEDER}}); C = \emptyset, S, \tag{23}$$

where $S$ can be any value. This means that the resulting set $\widetilde{\mathcal{D}}_{\texttt{FEEDER}}$ is a sufficient set of $\mathcal{D}_{\texttt{TRAIN}}$. $\square$

## A4    EXACT EXTRACTION OF FEEDER

To extract an exact FEEDER set $\mathcal{D}_{\texttt{FEEDER}}$ from $\mathcal{D}_{\texttt{TRAIN}}$, we need to explicitly check the necessity among all the candidate samples, and remove those unnecessary parts. We do not directly apply this algorithm in practice, due to its high computation costs. We provide a solution for integrating the algorithm into our FEEDER and report the corresponding results in Appendix A7.

### A4.1    EXACT EXTRACTION OF FEEDER VIA NECESSITY CHECKS

Our intuition behind constructing a tree for checking necessity is grounded in the inherent transitivity property of necessity. Formally, it can be expressed as: If unplugging $\mathcal{D}_{\texttt{A}}$ could cause the outputs to at least one input in $\mathcal{D}_{\texttt{C}}$ from correct to incorrect, then unplugging $\mathcal{D}_{\texttt{A}} \cup \mathcal{D}_{\texttt{B}}$ also can not maintain the outputs to all the input in $\mathcal{D}_{\texttt{C}}$ correct. Namely, if unplugging a subset would degrade the performance, then unplugging the whole set would also degrade the performance.

Similar to the tree for explicitly checking sufficiency introduced in Appendix A3, each node in the tree for checking necessity also represents a set of samples. For convenience, we also use $\mathcal{D}_{\texttt{IN}} = \{(\boldsymbol{x}_n, \boldsymbol{y}_n)\}_{n=1}^{N_{\texttt{IN}}}$ to denote the input set and $\mathcal{D}_{\texttt{OUT}}$ for the corresponding output. We use $\mathscr{H}_k$ to denote a set of nodes after the $k$-th iteration.

We initialize $\mathscr{H}_0$ by identifying all samples in $\mathcal{D}_{\texttt{IN}}$ for which unplugging them individually does not affect the LLM's performance. Formally, we construct $\mathscr{H}_0$ as $\mathscr{H}_0 := \{\mathcal{H}_n := \{(\boldsymbol{x}_n, \boldsymbol{y}_n)\}\}$ where $(\boldsymbol{x}_n, \boldsymbol{y}_n) \in \mathcal{D}_{\texttt{IN}}$ satisfies:

$$Y_{(\{\boldsymbol{x}_{n'} | \boldsymbol{x}_{n'} \in \mathcal{D}_{\texttt{IN}}\})} = \mathbf{1}_{|\mathcal{D}_{\texttt{IN}}|} | \texttt{unplug}((\boldsymbol{x}_n, \boldsymbol{y}_n)); C = \mathcal{D}_{\texttt{IN}}, S, \tag{24}$$

where $S$ is loosened to allow for any value. During each $k$-th iteration, we generate $\mathscr{H}_k$ by examining the necessity relationship between each pair of nodes (denoted as $\mathcal{H}_i, \mathcal{H}_j \in \mathscr{H}_{k-1}$). Here, we further

verify whether solely unplugging $\mathcal{H}_i \cup \mathcal{H}_j$ does not impact the LLM's performance. Formally, we check whether the following equation holds:

$$Y_{(\{\boldsymbol{x}_{n'}|\boldsymbol{x}_{n'}\in\mathcal{D}_{\text{IN}}\})} = \mathbf{1}_{|\mathcal{D}_{\text{IN}}|}|\text{unplug}(\mathcal{H}_i \cup \mathcal{H}_j); C = \mathcal{D}_{\text{IN}}, S, \tag{25}$$

where $S$ is loosened to allow for any value. This determines whether plugging $\mathcal{H}_i \cup \mathcal{H}_j$ is unnecessary for maintaining the correct outputs to all inputs in $\mathcal{D}_{\text{IN}}$. If the above equation holds, we create a new node $\mathcal{H}_i \cup \mathcal{H}_j$ and add it to $\mathscr{H}_k$, labeling it with a MAINTAIN signal. Otherwise, we add both $\mathcal{H}_i$ and $\mathcal{H}_j$ to $\mathscr{H}_k$. After this computation, we identify $\mathcal{H}_{\text{MAX}} = \arg\max_{\mathcal{H}_. \in \mathscr{H}_k} |\mathcal{H}_.|$ and label it with a MAINTAIN signal. Subsequently, we remove the nodes in $\mathscr{H}_k$ that lack MAINTAIN signals. This process continues until $\mathscr{H}_.$ contains only one element, denoted as $\mathcal{H}_{\text{UNNECESSARY}} \in \mathscr{H}_K$. Finally, we calculate $\mathcal{D}_{\text{OUT}}$ as $\mathcal{D}_{\text{OUT}} = \mathcal{D}_{\text{IN}} - \mathcal{H}_{\text{UNNECESSARY}}$.

## A4.2 EXACT EXTRACTION OF FEEDER VIA ITERATIVE SUFFICIENCY CHECKS

Consider that at each iteration, we need to check the necessity for $O(\mathtt{C}_{N_{\text{IN}}}^2)$ times (where $\mathtt{C}_.$ denotes a combination operator), this becomes impractical. To this end, we develop an alternative algorithm. Specifically, at each $k$-th iteration, we remove all the checked nodes (i.e., $\mathcal{H}_i$ and $\mathcal{H}_j$ from $\mathscr{H}_k$, similar to our approximation algorithm in Appendix A3). Then, it requires $O(\log_2^{|\mathcal{D}_{\text{IN}}|})$ computations to finish one round. To obtain an exact FEEDER, we need to keep repeating the above process until there is no or only one left in $\mathcal{H}_0$. While practical, we also can set a maximum number of rounds to approximate.

**Proposition 3** ($\widetilde{\mathcal{D}}_{\text{FEEDER}}$ obtained by either Algorithm 3 or Algorithm 4 is an Exact $\mathcal{D}_{\text{FEEDER}}$). *If we successively apply either Algorithm 3 or Algorithm 4 on $\mathcal{D}_{\text{TRAIN}}$ for multiple rounds to obtain a subset (denoted as $\widetilde{\mathcal{D}}_{\text{FEEDER}}$), then $\widetilde{\mathcal{D}}_{\text{FEEDER}}$ is sufficient and necessary to represent $\mathcal{D}_{\text{TRAIN}}$.*

*Proof.* According to Definition 3, it is straightforward to see that to prove the above proposition is equivalent to proving that $\widetilde{\mathcal{D}}_{\text{FEEDER}}$ is a sufficient set of $\mathcal{D}_{\text{TRAIN}}$ and a necessary set of $\mathcal{D}_{\text{TRAIN}}$.

We begin by proving sufficiency. Either Algorithm 3 or 4 preserves the sufficiency during checking the necessity, as we are always guaranteeing $Y_{(\{\boldsymbol{x}_n|\boldsymbol{x}_n\in\mathcal{D}_{\text{TRAIN}}\})} = \mathbf{1}_{|\mathcal{D}_{\text{TRAIN}}|}$, when removing the unnecessary parts.

In other words, we have:

$$Y_{(\{\boldsymbol{x}_n|\boldsymbol{x}_n\in\mathcal{D}_{\text{TRAIN}}\})} = \mathbf{1}_{|\mathcal{D}_{\text{TRAIN}}|}|\text{unplug}(\mathcal{D}_{\text{TRAIN}} - \mathcal{H}_{\text{UNNECESSARY}}); C = \mathcal{D}_{\text{TRAIN}}, S, \tag{26}$$

where $S$ can be any value. It can be rewritten as:

$$Y_{(\{\boldsymbol{x}_n|\boldsymbol{x}_n\in\mathcal{D}_{\text{TRAIN}}\})} = \mathbf{1}_{|\mathcal{D}_{\text{TRAIN}}|}|\text{plug}(\widetilde{\mathcal{D}}_{\text{FEEDER}}); C = \emptyset, S, \tag{27}$$

where $S$ can be any value. It shows that plugging in $\widetilde{\mathcal{D}}_{\text{FEEDER}}$ is sufficient for representing $\mathcal{D}_{\text{TRAIN}}$.

Next, we investigate necessity. Our goal is to prove unplugging any data point in $\widetilde{\mathcal{D}}_{\text{FEEDER}}$ would lead to a degradation of the LLM's performance. For convenience, we use $(\boldsymbol{x}_n, \boldsymbol{y}_n) \in \mathcal{D}_{\text{TRAIN}}$ to denote an arbitrary data point. If we applying Algorithm 3 to execute the search for an exact $\mathcal{D}_{\text{FEEDER}}$, then $(\boldsymbol{x}_n, \boldsymbol{y}_n)$ must be in $\mathscr{H}_0$, or out of $\mathscr{H}_0$.

If $(\boldsymbol{x}_n, \boldsymbol{y}_n)$ is not an element in $\mathscr{H}_0$, then according to the computing process of $\mathscr{H}_0$ (i.e., lines 3 to 3 in Algorithm 3), unplugging $(\boldsymbol{x}_n, \boldsymbol{y}_n)$ it would definitively cause the LLM's performance on $\mathcal{D}_{\text{TRAIN}}$ from $Y_{(\{\boldsymbol{x}_n|\boldsymbol{x}_n\in\mathcal{D}_{\text{TRAIN}}\})} = \mathbf{1}_{|\mathcal{D}_{\text{TRAIN}}|}$ to $Y_{(\{\boldsymbol{x}_n|\boldsymbol{x}_n\in\mathcal{D}_{\text{TRAIN}}\})} \neq \mathbf{1}_{|\mathcal{D}_{\text{TRAIN}}|}$.

If $(\boldsymbol{x}_n, \boldsymbol{y}_n)$ is an element in $\mathscr{H}_0$, then $(\boldsymbol{x}_n, \boldsymbol{y}_n)$ must be in $\mathcal{H}_{\text{UNNECESSARY}}$; otherwise, according to lines 3 to 3 in Algorithm 3, $\mathcal{H}_{\text{UNNECESSARY}} \cup \{(\boldsymbol{x}_n, \boldsymbol{y}_n)\}$ should be $\mathcal{H}_{\text{MAX}}$ and always stay in $\mathcal{H}_.$ until becoming the root node (i.e., $\mathcal{H}_{\text{UNNECESSARY}}$ should be updated to be $\mathcal{H}_{\text{UNNECESSARY}} \cup \{(\boldsymbol{x}_n, \boldsymbol{y}_n)\}$). Thus, $(\boldsymbol{x}_n, \boldsymbol{y}_n)$ must be in $\mathcal{H}_{\text{UNNECESSARY}}$. However, all the data points in $\mathcal{H}_{\text{UNNECESSARY}}$ are removed from $\mathcal{D}_{\text{TRAIN}}$, causing a contradiction. Hence, unplugging $(\boldsymbol{x}_n, \boldsymbol{y}_n)$ would change the LLM's performance, namely necessity holds.

Then, we consider applying Algorithm 4 for searching an exact $\mathcal{D}_{\text{FEEDER}}$. Similarly, if $(\boldsymbol{x}_n, \boldsymbol{y}_n)$ is not selected when checking the necessity, then unplugging $(\boldsymbol{x}_n, \boldsymbol{y}_n)$ would definitively cause a degradation of the LLM's performance.

Table 5: Performance comparisons on text classification datasets are conducted in the in-context learning setting. We report both the mean and variance of accuracy using 8 different seeds and 5 different permutations of n-shots. This table is extended from Table 1.

| $\Psi_{LLM}(\cdot)$ | $\mathcal{D}$ | $n$ | FPB | | | SST-5 | | | TREC | | |
|---|---|---|---|---|---|---|---|---|---|---|---|
| | | | RAN | SIM | DIV | RAN | SIM | DIV | RAN | SIM | DIV |
| SMA (0.3B) | $\mathcal{D}_{TRAIN}$ | 1 | 27.2 (6.1) | 25.3 (0.1) | 25.3 (0.1) | 14.5 (6.1) | 22.7 (0.2) | 22.7 (0.2) | 19.4 (6.4) | 42.8 (0.1) | 42.8 (0.1) |
| | | 2 | 27.4 (6.2) | 45.8 (0.2) | 40.4 (0.1) | 18.0 (5.8) | 25.6 (0.1) | 23.7 (0.2) | 21.4 (4.7) | 57.2 (0.2) | 51.4 (0.1) |
| | | 5 | 26.3 (4.5) | 55.9 (0.1) | 44.7 (0.2) | 26.5 (5.3) | 32.3 (0.2) | 27.8 (0.1) | 37.6 (5.1) | 66.0 (0.1) | 61.4 (0.3) |
| | | 10 | 27.8 (5.1) | 63.1 (0.1) | 50.7 (0.1) | 14.9 (3.9) | 35.3 (0.1) | 30.4 (0.2) | 53.0 (5.2) | 71.4 (0.2) | 65.8 (0.3) |
| | $\mathcal{D}_{FEEDER}$ | 1 | **28.4** (3.4) | **28.8** (2.1) | **28.8** (2.1) | **15.4** (5.2) | **23.7** (1.7) | **23.7** (1.7) | **37.4** (3.6) | **48.4** (1.6) | **48.4** (1.6) |
| | | 2 | **35.5** (4.3) | **47.4** (2.6) | 37.9 (1.9) | **20.9** (4.7) | **27.9** (1.1) | **25.8** (1.3) | **27.6** (3.2) | **58.8** (2.2) | **52.1** (1.9) |
| | | 5 | **28.3** (3.0) | 54.6 (1.7) | **47.9** (1.0) | **28.6** (3.4) | **33.2** (1.8) | 27.4 (1.7) | **40.8** (3.0) | **67.4** (1.2) | **61.8** (1.3) |
| | | 10 | **39.6** (3.4) | **66.5** (2.3) | **51.8** (1.2) | **17.6** (2.2) | **36.9** (1.9) | 29.8 (1.7) | 44.6 (2.8) | **74.6** (1.4) | **67.6** (1.9) |
| MED (0.8B) | $\mathcal{D}_{TRAIN}$ | 1 | 33.8 (5.2) | 29.9 (0.1) | 29.9 (0.1) | 14.2 (4.9) | 25.2 (0.1) | 25.2 (0.1) | 21.0 (4.6) | 53.2 (0.2) | 53.2 (0.2) |
| | | 2 | 27.0 (6.1) | 55.4 (0.2) | 49.9 (0.3) | 18.1 (5.1) | 29.7 (0.1) | 24.4 (0.2) | 28.2 (4.4) | 62.6 (0.2) | 60.6 (0.2) |
| | | 5 | 27.2 (4.8) | 64.3 (0.1) | 45.1 (0.3) | 25.6 (4.8) | 34.1 (0.1) | 30.8 (0.1) | 35.4 (5.7) | 63.4 (0.1) | 64.6 (0.1) |
| | | 10 | 47.0 (5.5) | 65.5 (0.2) | 52.9 (0.1) | 28.7 (4.2) | 38.7 (0.1) | 36.6 (0.1) | 43.2 (4.8) | 66.0 (0.1) | 68.8 (0.1) |
| | $\mathcal{D}_{FEEDER}$ | 1 | 33.8 (4.4) | **32.6** (0.7) | **32.6** (0.7) | **18.7** (3.0) | **25.5** (2.2) | **25.5** (2.2) | **22.4** (3.8) | 52.6 (2.1) | 52.6 (2.1) |
| | | 2 | **37.5** (4.7) | 54.8 (1.1) | 47.6 (1.3) | **25.2** (3.8) | 29.7 (1.9) | 24.1 (2.1) | **34.6** (3.5) | **64.2** (1.8) | 59.4 (2.0) |
| | | 5 | **38.9** (3.3) | **64.5** (1.3) | **48.0** (2.7) | **39.3** (2.9) | **35.2** (1.1) | **31.0** (1.2) | **45.4** (3.3) | **65.5** (1.5) | **64.9** (1.7) |
| | | 10 | **63.5** (2.8) | **66.7** (1.6) | **53.1** (1.5) | **39.6** (3.0) | **39.8** (1.8) | **37.8** (1.6) | **55.8** (3.8) | **70.4** (2.0) | 68.6 (1.7) |
| NEO (1.3B) | $\mathcal{D}_{TRAIN}$ | 1 | 54.9 (3.9) | 61.6 (0.1) | 61.6 (0.1) | 12.8 (2.7) | 20.2 (0.1) | 20.2 (0.1) | 11.0 (3.2) | 57.2 (0.2) | 57.2 (0.2) |
| | | 2 | 53.6 (4.0) | 66.8 (0.2) | 60.0 (0.1) | 17.9 (3.6) | 26.9 (0.1) | 22.7 (0.1) | 17.6 (3.1) | 52.6 (0.2) | 42.2 (0.2) |
| | | 5 | 28.2 (4.0) | 68.2 (0.1) | 60.4 (0.1) | 19.0 (3.9) | 29.2 (0.1) | 25.1 (0.1) | 25.2 (3.8) | 66.4 (0.1) | 61.8 (0.1) |
| | | 10 | 49.0 (4.8) | 75.8 (0.1) | 71.1 (0.2) | 12.7 (2.8) | 33.7 (0.2) | 31.9 (0.1) | 41.6 (4.4) | 70.6 (0.1) | 69.0 (0.1) |
| | $\mathcal{D}_{FEEDER}$ | 1 | **58.1** (4.7) | **61.8** (1.4) | **61.8** (1.4) | **18.5** (2.1) | **20.6** (1.8) | **20.6** (1.4) | **18.2** (2.4) | 56.4 (1.3) | 56.4 (1.3) |
| | | 2 | **61.4** (3.3) | 64.1 (1.5) | 58.8 (1.1) | **19.7** (2.7) | **27.4** (2.1) | **22.8** (1.8) | **27.8** (2.7) | **54.0** (1.4) | **44.5** (1.6) |
| | | 5 | **43.2** (2.6) | **68.8** (1.8) | **62.7** (1.3) | **19.2** (3.2) | **30.2** (2.7) | **26.4** (2.4) | **50.4** (2.2) | **68.0** (1.4) | **62.6** (1.9) |
| | | 10 | **61.4** (2.3) | 74.8 (1.9) | **71.9** (1.8) | **15.4** (2.4) | **37.0** (1.5) | **34.5** (1.9) | **45.2** (2.9) | **72.8** (1.4) | **69.8** (1.5) |
| GEM (2B) | $\mathcal{D}_{TRAIN}$ | 1 | 58.2 (5.7) | 62.5 (0.1) | 62.5 (0.1) | 21.5 (3.9) | 22.5 (0.1) | 22.5 (0.1) | 21.9 (3.4) | 52.3 (0.1) | 52.3 (0.1) |
| | | 2 | 59.2 (5.9) | 66.2 (0.4) | 65.8 (0.3) | 26.5 (3.6) | 42.5 (0.6) | 42.2 (0.6) | 35.6 (4.4) | 60.0 (0.2) | 59.1 (0.1) |
| | | 5 | 48.6 (3.6) | 76.6 (0.4) | 78.8 (0.6) | 26.6 (2.5) | 48.8 (0.3) | 41.2 (0.4) | 55.8 (2.9) | 82.2 (0.2) | 71.1 (0.6) |
| | | 10 | 35.2 (6.5) | 79.5 (0.4) | 78.8 (0.2) | 36.6 (4.4) | 50.2 (0.8) | 43.3 (0.4) | 51.1 (3.3) | 84.3 (0.5) | 75.0 (0.4) |
| | $\mathcal{D}_{FEEDER}$ | 1 | **59.9** (4.4) | **64.6** (0.6) | **64.6** (0.6) | **22.6** (4.2) | **25.8** (1.3) | **25.8** (1.3) | **26.2** (1.8) | **55.1** (1.6) | **55.1** (1.6) |
| | | 2 | 55.4 (2.4) | **67.8** (1.8) | **67.0** (1.1) | **28.7** (2.3) | **45.4** (1.4) | **46.8** (1.1) | **40.8** (1.5) | **63.6** (1.3) | **62.8** (1.6) |
| | | 5 | **52.2** (3.4) | **88.0** (4.6) | **80.1** (3.2) | **30.5** (2.0) | **52.6** (1.9) | **54.4** (1.4) | **60.4** (2.5) | **87.8** (1.6) | **73.0** (1.2) |
| | | 10 | **39.1** (5.1) | **81.3** (3.3) | **83.8** (2.4) | **36.8** (2.2) | **62.5** (1.5) | **54.9** (1.3) | **58.1** (5.2) | **88.9** (1.8) | **83.4** (1.4) |
| LAR (6B) | $\mathcal{D}_{TRAIN}$ | 1 | 30.7 (5.5) | 55.3 (0.1) | 55.3 (0.1) | 19.6 (3.6) | 20.5 (0.1) | 20.5 (0.1) | 21.4 (4.4) | 50.7 (0.1) | 50.7 (0.1) |
| | | 2 | 33.4 (4.9) | 64.9 (0.4) | 65.5 (0.3) | 24.1 (3.0) | 30.5 (0.4) | 31.6 (0.3) | 34.4 (4.0) | 58.8 (0.2) | 60.7 (0.1) |
| | | 5 | 40.6 (3.0) | 75.0 (0.4) | 74.9 (0.1) | 24.1 (2.5) | 32.5 (0.3) | 35.6 (0.2) | 51.8 (2.9) | 71.2 (0.2) | 70.6 (0.4) |
| | | 10 | 25.9 (6.5) | 78.5 (0.4) | 79.5 (0.2) | 35.5 (4.2) | 38.9 (0.1) | 40.5 (0.3) | 49.5 (3.6) | 72.5 (0.1) | 73.0 (0.2) |
| | $\mathcal{D}_{FEEDER}$ | 1 | **31.2** (4.8) | 54.8 (0.8) | 54.8 (0.8) | **20.6** (3.1) | **27.8** (1.3) | **27.8** (1.3) | **32.2** (1.8) | **52.1** (1.8) | **52.1** (1.8) |
| | | 2 | **35.4** (2.4) | **65.8** (1.8) | **67.1** (0.9) | **28.7** (2.3) | **33.4** (1.4) | **33.0** (1.1) | **44.8** (2.5) | **60.1** (1.5) | **61.8** (1.4) |
| | | 5 | **42.2** (3.4) | **77.9** (3.6) | **78.4** (3.2) | **28.5** (2.0) | **35.6** (1.3) | **37.4** (1.4) | **53.4** (2.7) | **75.8** (1.6) | **72.2** (1.2) |
| | | 10 | **39.1** (5.1) | **80.3** (3.3) | **82.8** (2.4) | **36.8** (2.2) | **41.5** (1.5) | **40.9** (1.1) | **54.1** (5.2) | **76.9** (1.8) | **80.4** (1.4) |
| LLA (7B) | $\mathcal{D}_{TRAIN}$ | 1 | 29.0 (4.7) | 47.1 (0.1) | 47.1 (0.1) | 28.6 (2.9) | 29.7 (0.1) | 29.7 (0.1) | 35.2 (3.7) | 54.2 (0.1) | 54.2 (0.1) |
| | | 2 | 27.4 (3.4) | 68.4 (0.2) | 67.1 (0.3) | 35.9 (3.1) | 33.9 (0.1) | 33.5 (0.3) | 45.0 (4.0) | 69.4 (0.1) | 63.6 (0.1) |
| | | 5 | 39.7 (3.2) | 80.3 (0.2) | 78.9 (0.1) | 37.9 (2.3) | 38.3 (0.2) | 37.0 (0.1) | 53.0 (3.6) | 79.0 (0.2) | 70.4 (0.3) |
| | | 10 | 37.9 (2.6) | 87.4 (0.3) | 86.5 (0.2) | 38.4 (3.8) | 37.5 (0.1) | 40.0 (0.2) | 58.0 (2.3) | 83.4 (0.1) | 79.2 (0.1) |
| | $\mathcal{D}_{FEEDER}$ | 1 | **33.7** (5.3) | **51.7** (0.8) | **51.7** (0.8) | 27.6 (2.4) | **32.3** (1.5) | **32.3** (1.3) | **41.2** (2.1) | **56.8** (1.8) | **56.8** (1.8) |
| | | 2 | **39.6** (5.0) | **68.7** (1.5) | **69.8** (0.7) | **39.5** (2.5) | 32.6 (1.2) | 32.7 (1.1) | **53.8** (2.3) | 68.6 (1.7) | 63.5 (1.3) |
| | | 5 | **45.6** (4.8) | **87.9** (4.8) | **79.5** (3.5) | **39.2** (2.0) | **38.7** (1.3) | **39.4** (1.0) | **58.2** (2.8) | **82.8** (1.6) | **71.8** (1.4) |
| | | 10 | 37.8 (6.4) | 87.1 (3.9) | **87.8** (2.2) | **39.7** (2.8) | **39.0** (1.0) | **41.6** (1.3) | **59.8** (3.1) | **86.0** (1.9) | **83.4** (2.0) |

Table 6: A complementary table to Table 5 presents the corresponding results for the demonstration selectors UNC, CLU, LVM.

| $\Psi_{LLM}(\cdot)$ | $\mathcal{D}$ | $n$ | FPB | | | SST-5 | | | TREC | | |
|---|---|---|---|---|---|---|---|---|---|---|---|
| | | | UNC | CLU | LVM | UNC | CLU | LVM | UNC | CLU | LVM |
| LAR (6B) | $\mathcal{D}_{TRAIN}$ | 1 | 55.8 (6.3) | 56.3 (4.0) | 58.0 (2.5) | 29.0 (2.9) | 27.5 (1.5) | 25.8 (1.1) | 52.0 (6.5) | 49.8 (1.5) | 50.2 (1.2) |
| | | 2 | 67.8 (3.7) | 66.5 (4.1) | 66.3 (3.5) | 35.6 (4.2) | 36.1 (2.2) | 34.0 (2.4) | 59.6 (4.0) | 60.8 (5.0) | 58.5 (3.3) |
| | | 5 | 76.7 (4.5) | 78.2 (4.4) | 79.4 (4.2) | 41.8 (3.3) | 42.2 (3.3) | 40.7 (4.4) | 65.4 (3.5) | 66.4 (4.3) | 65.8 (3.3) |
| | | 10 | 78.3 (4.8) | 80.7 (3.8) | 81.3 (4.1) | 40.5 (3.8) | 38.8 (3.9) | 36.8 (4.1) | 78.4 (4.2) | 72.1 (3.6) | 71.5 (4.5) |
| | $\mathcal{D}_{FEEDER}$ | 1 | **56.3** (4.2) | **57.9** (4.4) | **58.2** (3.2) | **32.3** (2.4) | **29.4** (3.4) | **28.3** (2.6) | **53.8** (2.1) | **50.8** (3.5) | **52.5** (5.1) |
| | | 2 | **69.8** (3.0) | **69.7** (3.5) | **69.5** (2.5) | **37.1** (2.5) | **42.5** (3.5) | **38.2** (3.2) | **60.1** (2.1) | 57.8 (4.8) | **59.1** (3.5) |
| | | 5 | **82.3** (3.8) | **82.0** (2.4) | **81.8** (2.9) | **44.2** (4.0) | **45.8** (3.8) | **44.4** (2.9) | **68.4** (2.7) | **66.6** (3.7) | **67.3** (2.4) |
| | | 10 | **80.8** (3.4) | **83.0** (2.4) | **83.8** (2.9) | **42.2** (2.8) | **40.8** (3.8) | **40.4** (2.9) | **82.4** (3.0) | **74.7** (3.1) | **73.5** (2.5) |
| LLA (7B) | $\mathcal{D}_{TRAIN}$ | 1 | 49.0 (6.6) | 47.5 (5.6) | 47.5 (5.1) | 36.2 (2.4) | 37.2 (3.7) | 38.7 (4.1) | 55.1 (6.1) | 54.1 (4.0) | 54.0 (3.3) |
| | | 2 | 68.2 (4.8) | 67.8 (3.5) | 68.7 (4.1) | 35.1 (4.2) | 32.5 (2.0) | 34.7 (4.2) | 67.5 (4.5) | 68.2 (4.0) | 66.4 (1.3) |
| | | 5 | 80.9 (3.2) | 81.6 (2.2) | 83.8 (1.2) | 36.7 (3.8) | 38.5 (3.0) | 39.2 (1.2) | 68.2 (3.7) | 69.2 (2.5) | 67.3 (2.2) |
| | | 10 | 86.2 (4.6) | 85.1 (4.4) | 87.3 (2.1) | 36.4 (3.1) | 35.2 (3.7) | 39.8 (4.1) | 86.5 (4.3) | 85.6 (4.0) | 87.3 (2.2) |
| | $\mathcal{D}_{FEEDER}$ | 1 | **51.2** (4.8) | 48.9 (4.3) | 48.7 (5.1) | **41.8** (2.4) | **44.4** (3.5) | **43.3** (2.7) | **58.0** (2.1) | **62.2** (2.5) | **62.8** (1.8) |
| | | 2 | **71.8** (3.0) | **72.8** (3.4) | **73.5** (2.4) | **45.1** (3.1) | **45.3** (3.1) | **46.5** (4.0) | **69.5** (2.3) | **70.8** (2.3) | **70.6** (2.7) |
| | | 5 | **88.5** (3.8) | **85.7** (4.8) | **86.9** (2.8) | **42.1** (4.6) | **42.3** (4.5) | **40.8** (4.1) | **72.8** (2.8) | **75.8** (3.8) | **69.3** (2.6) |
| | | 10 | **88.8** (3.4) | **91.1** (4.4) | **89.2** (2.9) | **46.9** (2.2) | **50.1** (2.0) | **53.0** (2.2) | **87.4** (3.1) | **88.5** (3.4) | **89.0** (2.7) |

Table 7: Performance comparisons on text classification datasets are conducted in the fine-tuning setting, where we tune the LLMs and evaluate their few-shot inference performance. We report both the mean and variance of accuracy using 8 different seeds and 5 different permutations of n-shots. This table is extended from Table 4.

| $\Psi_{\text{LLM}}(\cdot)$ | $\mathcal{D}$ | $n$ | FPB | | | SST-5 | | | TREC | | |
|---|---|---|---|---|---|---|---|---|---|---|---|
| | | | RAN | SIM | DIV | RAN | SIM | DIV | RAN | SIM | DIV |
| SMA (0.3B) | $\mathcal{D}_{\text{TRAIN}}$ | 1 | 58.3 (5.7) | 68.4 (0.1) | 67.4 (0.1) | 55.5 (4.8) | 60.2 (0.4) | 58.4 (0.2) | 59.2 (5.2) | 70.0 (0.1) | 68.0 (0.1) |
| | | 2 | 58.5 (5.2) | 72.3 (0.4) | 70.1 (0.2) | 58.5 (4.2) | 60.4 (0.6) | 61.2 (0.4) | 57.7 (5.2) | 70.1 (0.2) | 70.3 (0.4) |
| | | 5 | 67.8 (5.1) | 66.2 (0.4) | 65.7 (0.3) | 58.6 (5.2) | 60.4 (0.7) | 61.8 (0.5) | 66.3 (4.5) | 72.8 (0.4) | 70.2 (0.5) |
| | | 10 | 58.2 (4.4) | 63.3 (0.6) | 65.6 (0.3) | 61.4 (4.3) | 60.4 (0.4) | 61.8 (0.2) | 60.9 (3.8) | 71.3 (0.5) | 72.5 (0.9) |
| | $\mathcal{D}_{\text{FEEDER}}$ | 1 | **65.0** (5.5) | **77.3** (1.3) | **73.3** (1.3) | **61.7** (4.2) | **74.8** (1.8) | **74.4** (0.8) | **63.9** (4.0) | **74.3** (0.7) | **75.3** (0.7) |
| | | 2 | **62.2** (3.4) | **75.0** (1.1) | **74.3** (1.5) | **62.3** (3.4) | **63.4** (1.8) | **62.6** (1.2) | **60.1** (3.5) | **76.1** (1.7) | **74.4** (0.9) |
| | | 5 | **70.4** (3.2) | **78.8** (1.6) | **76.4** (1.0) | **62.4** (4.2) | **62.2** (1.4) | **66.4** (1.3) | **68.8** (3.2) | **77.2** (3.3) | **76.6** (2.9) |
| | | 10 | **62.3** (3.3) | **80.6** (1.3) | **78.6** (1.9) | **63.9** (4.5) | **78.6** (1.9) | **71.0** (1.2) | **68.7** (2.7) | **72.2** (1.7) | **75.7** (1.9) |
| MED (0.8B) | $\mathcal{D}_{\text{TRAIN}}$ | 1 | 60.3 (4.7) | 73.4 (0.1) | 73.4 (0.1) | 57.5 (5.1) | 64.3 (0.2) | 64.3 (0.2) | 61.1 (5.2) | 77.3 (0.1) | 77.3 (0.1) |
| | | 2 | 62.5 (5.2) | 75.3 (0.4) | 75.1 (0.3) | 62.5 (4.2) | 65.4 (0.6) | 66.2 (0.4) | 62.7 (5.2) | 78.1 (0.2) | 79.3 (0.4) |
| | | 5 | 71.8 (5.1) | 72.2 (0.4) | 70.1 (0.3) | 63.6 (5.2) | 67.4 (0.7) | 68.6 (0.6) | 64.3 (4.5) | 76.8 (0.4) | 74.2 (0.5) |
| | | 10 | 63.2 (4.4) | 67.3 (0.6) | 68.6 (0.3) | 66.4 (4.3) | 68.4 (0.4) | 67.8 (0.2) | 66.9 (3.8) | 78.3 (0.5) | 75.5 (0.9) |
| | $\mathcal{D}_{\text{FEEDER}}$ | 1 | **69.0** (5.3) | **81.3** (1.3) | **81.3** (1.3) | **59.8** (4.2) | **72.8** (0.8) | **72.8** (0.8) | **65.9** (4.0) | **83.3** (0.7) | **83.3** (0.7) |
| | | 2 | **73.2** (3.4) | **82.0** (1.1) | **83.3** (1.5) | **65.3** (3.4) | **73.4** (1.8) | **72.6** (1.2) | **62.1** (3.5) | **80.1** (1.7) | **82.2** (0.9) |
| | | 5 | **74.4** (3.4) | **84.8** (1.6) | **86.4** (1.4) | **67.4** (3.9) | **77.5** (1.0) | **76.7** (1.4) | **69.8** (3.2) | **83.2** (3.3) | **84.6** (2.9) |
| | | 10 | **75.3** (3.3) | **85.6** (1.3) | **87.6** (1.9) | **58.9** (3.5) | **78.6** (1.7) | **79.0** (1.2) | **69.7** (2.7) | **86.2** (1.7) | **85.7** (1.9) |
| NEO (1.3B) | $\mathcal{D}_{\text{TRAIN}}$ | 1 | 62.7 (5.7) | 78.4 (0.1) | 78.4 (0.1) | 60.3 (4.1) | 66.6 (1.4) | 66.6 (1.4) | 63.3 (5.2) | 79.5 (0.4) | 79.5 (0.4) |
| | | 2 | 63.1 (4.6) | 74.2 (0.3) | 73.1 (0.2) | 64.5 (3.2) | 66.8 (0.8) | 68.4 (0.7) | 63.5 (5.7) | 81.2 (0.4) | 81.4 (0.6) |
| | | 5 | 70.8 (5.1) | 73.3 (0.1) | 72.7 (0.2) | 63.6 (4.1) | 70.8 (0.4) | 70.8 (0.4) | 67.8 (4.7) | 80.6 (0.5) | 82.0 (0.4) |
| | | 10 | 62.2 (4.4) | 63.0 (0.6) | 69.6 (0.5) | 65.8 (2.9) | 69.5 (0.3) | 68.8 (0.6) | 68.1 (3.8) | 78.8 (0.4) | 82.4 (0.5) |
| | $\mathcal{D}_{\text{FEEDER}}$ | 1 | **73.0** (4.4) | **83.5** (1.5) | **83.5** (1.5) | **63.3** (3.1) | **72.7** (1.3) | **72.7** (1.3) | **64.6** (3.2) | **84.6** (0.8) | **84.6** (0.8) |
| | | 2 | **76.1** (3.8) | **84.1** (1.4) | **82.5** (1.7) | **65.6** (2.7) | **76.4** (0.7) | **78.6** (0.8) | **64.2** (3.7) | **85.5** (0.7) | **86.3** (0.9) |
| | | 5 | **75.7** (3.5) | **90.7** (1.5) | **88.1** (1.9) | **67.4** (2.9) | **79.5** (1.8) | **79.7** (1.5) | **70.8** (3.2) | **88.2** (2.3) | **89.6** (1.9) |
| | | 10 | **77.5** (3.3) | **92.6** (1.3) | **90.6** (1.8) | **68.9** (2.0) | **82.6** (1.7) | **80.0** (1.6) | **73.7** (2.7) | **91.2** (1.7) | **86.7** (1.9) |

If $(\boldsymbol{x}_n, \boldsymbol{y}_n)$ is selected during checking the necessity, then $(\boldsymbol{x}_n, \boldsymbol{y}_n)$ must be included in $\mathcal{D}_r$; otherwise, $\mathcal{D}_r$ would continue to update, since the condition of stopping iteration is that there is no or only one unnecessary node. However, all the data points are removed from $\mathcal{D}_{\text{TRAIN}}$, causing a contradiction. Hence, unplugging $(\boldsymbol{x}_n, \boldsymbol{y}_n)$ would change the LLM's performance, namely necessity holds.

Combining the above analysis of sufficiency and necessity, we can conclude that $\mathcal{D}_{\text{FEEDER}}$ is an exact FEEDER for $\mathcal{D}_{\text{TRAIN}}$. □

## A5  FEEDER IN IN-CONTEXT LEARNING SETTING

### A5.1  DEMONSTRATION SELECTORS

As described in Section 5.1, when applied in the in-context learning setting, our $\mathcal{D}_{\text{FEEDER}}$ is assessed by serving as the retrieval pool, replacing $\mathcal{D}_{\text{TRAIN}}$ for existing demonstration selectors.

The first one is a random selector, denoted as RAN, which randomly selects samples from the retrieval pool.

The second one is a similarity-based selector, denoted as SIM, which selects samples similar to the test samples. Formally, let $\mathcal{D}_{\text{RETRIEVE}}$ denote the retrieval pool. Then, for each test sample $\boldsymbol{x}_m$, the metric of SIM can be written as:

$$\text{SIM}(\boldsymbol{x}_m, \boldsymbol{x}_n) = \text{COS}(\text{TRANSFORMER}(\boldsymbol{x}_m), \text{TRANSFORMER}(\boldsymbol{x}_n)), \tag{28}$$

where $\boldsymbol{x}_n \in \mathcal{D}_{\text{RETRIEVE}}$, $\text{COS}(\cdot)$ is a cosine similarity metric, and $\text{TRANSFORMER}(\cdot)$ denotes a sentence transformer (Reimers & Gurevych, 2019). Here, we directly use the Sentence Transformers library[2] from Hugging Face in our implementation. Then, we are able to select $N_{\text{shot}}$ samples with maximum SIM values from $\mathcal{D}_{\text{RETRIEVE}}$.

The third one is a diversity based selector, denoted as DIV, where we adopt the maximal marginal relevance method (Carbonell & Goldstein, 1998) as the metric of DIV. Formally, we have:

$$\text{DIV}(\boldsymbol{x}_m, \boldsymbol{x}_n) = \text{SIM}(\boldsymbol{x}_m, \boldsymbol{x}_n) - \eta \cdot \max_{\boldsymbol{x}_{n'} \in \mathcal{D}'_{\text{RETRIEVE}}} \text{SIM}(\boldsymbol{x}_m, \boldsymbol{x}_{n'}), \tag{29}$$

where $\boldsymbol{x}_n \in \mathcal{D}_{\text{RETRIEVE}} - \mathcal{D}'_{\text{RETRIEVE}}$, and $\mathcal{D}'_{\text{RETRIEVE}}$ denotes the set of previously selected instances. We can see that DIV prefers the instance that is both similar to the test samples meanwhile distant to

---

[2] https://huggingface.co/sentence-transformers

---

**Algorithm 4:** Alternative Exact Algorithm for FEEDER

---

**Input:** Training dataset $\mathcal{D}_{\texttt{TRAIN}}$.

**Output:** Exact FEEDER $\widetilde{\mathcal{D}}_{\texttt{FEEDER}}$.

Initialize the number of rounds $r = 0$.

Initialize the set of unnecessary data $\mathcal{D}_r = \emptyset$.

**repeat**

    Initialize $k = 1$.

    Initialize $\mathscr{H}_0 = \emptyset$.

    Update input data by removing the unnecessary part $\mathcal{D}_{\texttt{IN}} = \mathcal{D}_{\texttt{TRAIN}} - \mathcal{D}_r$.

    **for** *each instance* $(\boldsymbol{x}_n, \boldsymbol{y}_n) \in \mathcal{D}_{\texttt{IN}}$ **do**

        Check $Y_{(\{\boldsymbol{x}_{n'}|\boldsymbol{x}_{n'} \in \mathcal{D}_{\texttt{IN}}\})} = \mathbf{1}_{|\mathcal{D}_{\texttt{IN}}|}|\texttt{unplug}((\boldsymbol{x}_n, \boldsymbol{y}_n)); C, S$ (a), $C = \mathcal{D}_{\texttt{IN}}$,

          $S = (Y_{(\{\boldsymbol{x}_{n'}|\boldsymbol{x}_{n'} \in \mathcal{D}_{\texttt{IN}}\})} = \mathbf{1}_{|\mathcal{D}_{\texttt{IN}}|})$.

        If (a) holds, let $\mathcal{H}_n = \{(\boldsymbol{x}_n, \boldsymbol{y}_n)\}$ and append $\mathcal{H}_n$ to $\mathscr{H}_0$.

    **end**

    **repeat**

        **for** *each pair* $(\mathcal{H}_i, \mathcal{H}_j)$ *where* $\mathcal{H}_i, \mathcal{H}_j \in \mathscr{H}_{k-1}$ **do**

            Check $Y_{(\{\boldsymbol{x}_n|\boldsymbol{x}_n \in \mathcal{D}_{\texttt{IN}}\})} = \mathbf{1}_{|\mathcal{D}_{\texttt{IN}}|}|\texttt{unplug}(\mathcal{H}_i \cup \mathcal{H}_j); C, S$ (b), where $C = \mathcal{D}_{\texttt{IN}}$ and

              $S = (Y_{(\{\boldsymbol{x}_{n'}|\boldsymbol{x}_{n'} \in \mathcal{D}_{\texttt{IN}}\})} = \mathbf{1}_{|\mathcal{D}_{\texttt{IN}}|})$.

            If (b) holds, generate a new node $\mathcal{H}_i \cup \mathcal{H}_j$, append it to $\mathscr{H}_k$, and assign $\mathcal{H}_i \cup \mathcal{H}_j$;

              otherwise, append $\mathcal{H}_i$ and $\mathcal{H}_j$ to $\mathscr{H}_k$.

            Remove $\mathcal{H}_i, \mathcal{H}_j$ from $\mathscr{H}_{k-1}$, i.e., $\mathscr{H}_{k-1} = \mathscr{H}_{k-1} - \{\mathcal{H}_i, \mathcal{H}_j\}$.

        **end**

        Grow tree from bottom to top via $k = k + 1$.

    **until** $|\mathscr{H}_k| = 1$ *where we assume the iteration is* $K$;

    Let $\mathcal{H}_{\texttt{UNNCESSARY}}$ denote only one element (i.e. the root node) in $\mathscr{H}_K$.

    Update the number of rounds, i.e., $r = r + 1$.

    Update $\mathcal{D}_r$ to include the unnecessary part $\mathcal{H}_{\texttt{UNNCESSARY}}$, i.e., $\mathcal{D}_r = \mathcal{D}_r \cup \mathcal{H}_{\texttt{UNNCESSARY}}$.

**until** $|\mathcal{H}_{\texttt{UNNCESSARY}}| \leq 1$;

Assign $\widetilde{\mathcal{D}}_{\texttt{FEEDER}}$ as removing $\mathcal{D}_r$ from $\mathcal{D}_{\texttt{TRAIN}}$, i.e., $\widetilde{\mathcal{D}}_{\texttt{FEEDER}} = \mathcal{D}_{\texttt{TRAIN}} - \mathcal{D}_r$.

---

previously selected instances. $\eta$ is a hyper-parameter to balance the above two parts. We set $\eta = 1$ in our experiment.

The fourth one is an uncertainty-based selector (Köksal et al., 2022), denoted as UNC, which conducts selections according to their uncertainty metric;

The fifth one is a clustering-based selector (Zhou et al., 2023), denoted as CLU, which searches demonstrations by clustering.

The sixth one uses LLMs as latent variable models (Wang et al., 2024), denoted as LVM, which learns latent variables for down-streaming in-context learning.

In our experiment, we run our approximation algorithm for 1 run to get $\mathcal{D}_{\texttt{FEEDER}}$, and then treat $\mathcal{D}_{\texttt{FEEDER}}$ as the retrieval pool for the above demonstration selectors. In our results, we report both the mean and variance of accuracy using 8 different seeds and 5 different permutations of n-shots.

We also want to emphasize that since our pre-selector and pre-selection process are novel, we evaluate the performance of FEEDER in an ablation fashion. Specifically, our results (denoted as $\mathcal{D}_{\texttt{FEEDER}}$ in the $\mathcal{D}$ column) can be interpreted as FEEDER + X (where X represents any demonstration retriever described above), meaning that FEEDER is used for pre-selection of input demonstrations, and X is used to select specific demonstrations considering the target inputs. Our baseline (denoted as $\mathcal{D}_{\texttt{TRAIN}}$ in the $\mathcal{D}$ column) can be formulated as X + X, meaning X is used for both pre-selection of input demonstrations and for selecting specific demonstrations with regard to the target inputs.

### A5.2 Additional Results with Diverse Datasets

We report performance comparison results on text classification datasets SUBJ, SST-2, and COLA datasets in Table 1. We include the results of FPB, SST-5, and TREC datasets in Table 5, whose trend

Table 8: Performance comparisons on reasoning GSM8K dataset and semantic-parsing SMCALFlow dataset are conducted in the in-context learning setting. We report both the mean and variance of accuracy using 8 different seeds and 5 different permutations of n-shots. This table is extended from Table 3.

| $\Psi_{\text{LLM}}(\cdot)$ | $\mathcal{D}$ | $n$ | GSM8K | | SMCALFlow | |
|---|---|---|---|---|---|---|
| | | | CLU | LVM | CLU | LVM |
| GEM (2B) | $\mathcal{D}_{\text{TRAIN}}$ | 1 | 16.17 (0.18) | 16.20 (0.19) | 20.02 (0.21) | 19.54 (0.14) |
| | | 2 | 19.89 (0.96) | 20.52 (0.15) | 22.58 (0.45) | 23.05 (0.36) |
| | | 5 | 21.31 (0.84) | 23.56 (0.66) | 29.30 (0.90) | 28.65 (0.95) |
| | | 10 | 22.52 (0.49) | 23.85 (0.65) | 30.12 (1.11) | 31.11 (0.91) |
| | $\mathcal{D}_{\text{FEEDER}}$ | 1 | **17.25** (0.21) | **16.68** (0.24) | **21.12** (1.78) | **20.89** (1.21) |
| | | 2 | **20.68** (0.83) | **21.01** (0.85) | **22.85** (2.65) | **25.03** (0.18) |
| | | 5 | **22.55** (0.75) | 23.05 (0.77) | **31.20** (1.15) | **29.54** (4.58) |
| | | 10 | **22.75** (0.85) | **24.02** (2.20) | **32.10** (2.01) | **32.48** (1.52) |
| LAR (6B) | $\mathcal{D}_{\text{TRAIN}}$ | 1 | 2.95 (0.12) | 2.87 (0.25) | 9.95 (0.79) | 9.21 (0.85) |
| | | 2 | 4.78 (0.33) | 4.21 (0.25) | 10.12 (0.46) | 10.14 (0.88) |
| | | 5 | 7.21 (0.78) | 8.00 (1.05) | 12.31 (1.11) | 12.15 (1.30) |
| | | 10 | 8.05 (1.20) | 7.44 (1.25) | 14.14 (1.57) | 13.99 (1.54) |
| | $\mathcal{D}_{\text{FEEDER}}$ | 1 | **4.10** (0.22) | **3.25** (0.24) | **12.52** (1.13) | **11.42** (1.02) |
| | | 2 | 4.26 (0.64) | **4.55** (0.82) | **11.73** (0.54) | **12.05** (0.80) |
| | | 5 | **8.85** (1.28) | **8.14** (0.87) | **13.58** (1.44) | **12.44** (1.69) |
| | | 10 | **9.52** (1.88) | **8.50** (1.21) | **15.08** (1.91) | **16.50** (1.25) |
| LLA (7B) | $\mathcal{D}_{\text{TRAIN}}$ | 1 | 3.68 (0.89) | 3.98 (0.88) | 10.12 (0.95) | 9.25 (0.85) |
| | | 2 | 5.20 (0.38) | 5.55 (0.85) | 11.05 (1.36) | 12.52 (1.45) |
| | | 5 | 7.58 (0.89) | 7.52 (0.96) | 15.18 (1.15) | 15.30 (1.20) |
| | | 10 | 9.85 (0.85) | 9.21 (0.98) | 17.95 (1.25) | 18.55 (2.01) |
| | $\mathcal{D}_{\text{FEEDER}}$ | 1 | **4.25** (0.21) | **4.17** (0.89) | **11.89** (0.51) | **12.05** (0.63) |
| | | 2 | **5.88** (0.63) | **6.02** (0.58) | **13.03** (0.16) | **14.13** (1.10) |
| | | 5 | **8.22** (1.01) | **9.17** (0.98) | **18.20** (3.66) | **19.66** (5.20) |
| | | 10 | **10.17** (1.22) | **9.65** (0.83) | **22.11** (1.22) | **21.25** (1.26) |

Table 9: Performance comparisons among using different LLMs MED, LAR, NEO as the base for acquiring a FEEDER set and using NEO for inference on COLA dataset are conducted in the in-context learning setting. We report both the mean and variance of accuracy using 8 different seeds and 5 different permutations of n-shots.

| $\Psi_{\text{LLM}}(\cdot)$ | $\mathcal{D}$ | $n$ | MED (0.8B) | | | LAR (6B) | | | NEO (1.3B) | | |
|---|---|---|---|---|---|---|---|---|---|---|---|
| | | | RAN | SIM | DIV | RAN | SIM | DIV | RAN | SIM | DIV |
| NEO (1.3B) | $\mathcal{D}_{\text{FEEDER}}$ | 1 | 23.7 (5.7) | 31.0 (1.3) | 31.0 (1.3) | 25.3 (4.1) | 34.6 (1.8) | 34.6 (1.8) | 28.3 (5.4) | 34.8 (1.3) | 34.8 (1.3) |
| | | 2 | 45.1 (5.6) | 49.7 (1.4) | 46.1 (0.8) | 58.5 (3.2) | 57.8 (1.2) | 56.4 (1.0) | 69.3 (3.7) | 64.7 (1.4) | 64.7 (1.6) |
| | | 5 | 49.4 (4.6) | 58.1 (2.5) | 59.1 (1.9) | 54.6 (3.8) | 64.5 (1.1) | 61.7 (2.4) | 68.7 (3.2) | 67.2 (2.4) | 65.8 (1.8) |
| | | 10 | 59.4 (4.6) | 62.4 (1.5) | 65.8 (1.5) | 60.6 (3.8) | 64.7 (1.8) | 66.0 (1.4) | 69.8 (2.8) | 68.8 (1.4) | 68.9 (1.3) |

is consistent with our results in Table 1. These results further verify the superiority of our FEEDER in the in-context learning setting.

Besides three basic demonstration selectors, denoted as RAN, SIM, and DIV, we also examine the performance of FEEDER with some recently proposed demonstration selectors, denoted as UNC, CLU, VLM. We summarize the corresponding results in Table 6, whose trend is consistent with our results in Table 2. Overall, compared to using the entire training dataset $\mathcal{D}_{\text{TRAIN}}$ as the retrieval pool, treating its core set $\mathcal{D}_{\text{FEEDER}}$ as the retrieval pool can improve the LLM performance at most cases. These results are consistent with the analysis reported in Section 5.1, which together verify that our FEEDER collaborating with various demonstration selectors works well in the in-context learning setting.

### A5.3 ADDITIONAL RESULTS WITH DIVERSE DEMONSTRATION SELECTORS

We report performance comparison results on the reasoning dataset GSM8K and the semantic parsing dataset SMCALFlow in Table 3. The corresponding results for additional demonstration selectors, CLU and LVM, are provided in Table 8, showing a similar trend. Together, these results further demonstrate the superiority of our FEEDER framework in the in-context learning setting.

## A6 SCALING UP FEEDER INTO REAL-WORLD APPLICATIONS

### A6.1 SCALING UP FEEDER TO LARGER LLMS.

As the LLM scales up in size (e.g., scaling up to Llama-65B (Touvron et al., 2023) and Gemma-70B (Team et al., 2024)), the execution of our approximation algorithm for searching $\mathcal{D}_{\text{FEEDER}}$ can become exceedingly time-consuming. In response to this challenge, we propose a strategy wherein a smaller

Table 10: Performance comparisons among using different LLMs MED, LAR, NEO as the base for acquiring a FEEDER set and using NEO for inference on COLA dataset are conducted in the in-context learning setting. We report both the mean and variance of accuracy using 8 different seeds and 5 different permutations of n-shots.

| $\Psi_{\text{LLM}}(\cdot)$ | $\mathcal{D}$ | $n$ | MED (0.8B) | | | LAR (6B) | | | NEO (1.3B) | | |
|---|---|---|---|---|---|---|---|---|---|---|---|
| | | | RAN | SIM | DIV | RAN | SIM | DIV | RAN | SIM | DIV |
| NEO (1.3B) | $\mathcal{D}_{\text{TRAIN}}$ | 2 | 23.7 (5.7) | 31.0 (1.3) | 31.0 (1.3) | 25.3 (4.1) | 34.6 (1.8) | 34.6 (1.8) | 28.3 (5.4) | 34.8 (1.3) | 34.8 (1.3) |
| | | 5 | 49.4 (4.6) | 58.1 (2.5) | 59.1 (1.9) | 54.6 (3.8) | 64.5 (1.1) | 61.7 (2.4) | 68.7 (3.2) | 67.2 (2.4) | 65.8 (1.8) |
| | $\mathcal{D}_{\text{FEEDER}}$ | 2 | 23.7 (5.7) | 31.0 (1.3) | 31.0 (1.3) | 25.3 (4.1) | 34.6 (1.8) | 34.6 (1.8) | 28.3 (5.4) | 34.8 (1.3) | 34.8 (1.3) |
| | | 5 | 49.4 (4.6) | 58.1 (2.5) | 59.1 (1.9) | 54.6 (3.8) | 64.5 (1.1) | 61.7 (2.4) | 68.7 (3.2) | 67.2 (2.4) | 65.8 (1.8) |

**Integrating Exact Extractor of FEEDER into FEEDER**

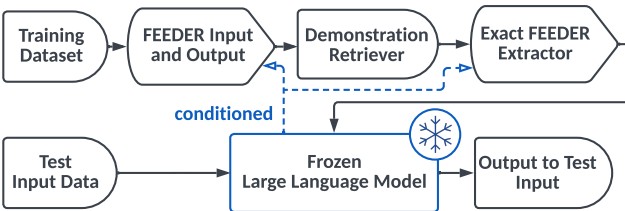

Figure 6: Integrating our extraction algorithm for FEEDER (i.e., Algorithm 4) into our in-context learning framework (as introduced in Figure 1(a)).

LLM is employed to generate a FEEDER set, which is then stored and utilized by the larger LLM. To assess the viability of this approach, we conducted an experiment comparing the performance of using SMA, MED, and NEO as the LLMs for obtaining a FEEDER set, and then we use this set as the retrieval pool to acquire demonstrations for NEO. Results summarized in Table 10 demonstrate that even when $\mathcal{D}_{\text{FEEDER}}$ is pre-selected by a small LLM, it contributes to improved performance, compared to using $\mathcal{D}_{\text{TRAIN}}$, as reported in Table 1. This observation suggests the potential feasibility of employing a more compact LLM for pre-selecting $\mathcal{D}_{\text{FEEDER}}$ to enhance the performance of a larger LLM.

### A6.2 SCALING UP FEEDER BY INCREMENTAL UPDATE.

Notice that numerous real-world datasets are temporal and require frequent updates. Re-running the tree based approximation algorithm for FEEDER over all samples can be excessively time-consuming. To address this, we design an incremental approach, treating the unchanged portion as a plug-and-play FEEDER set and the LLM as a whole, forming a new "LLM". Therefore, we can apply FEEDER solely to compute incremental data for the modified part, encompassing newly added and modified data points. Also, a significant challenge of FEEDER arises from the temporal nature of many real-world datasets, some of which require frequent updates, potentially on a daily basis. The conventional approach of recalculating a FEEDER over all unchanged and changed samples can be time-consuming in such dynamic scenarios. To address this challenge, we introduce an incremental update algorithm for FEEDER, enabling the efficient re-computation of only the changed portions, including newly added and modified samples.

As depicted in Figure 7, once a FEEDER set for the original dataset is generated, we treat the unchanged part of plug-and-play plugged data and the LLM as a whole (depicted by the dashed box) as a new "LLM". Subsequently, we apply FEEDER exclusively to compute incremental data for the changed part, covering newly added and modified data points. This strategy aims to enhance the efficiency and responsiveness of FEEDER in the context of evolving and temporal datasets.

### A7 INTEGRATING ALGORITHM 4 IN FEEDER

One limitation to directly applying Algorithm 3 or 4 is that $\mathcal{D}_{\text{TRAIN}}$ is too large to be directly used as input demonstrations. For this purpose, we incorporate running Algorithm 4 for one round into our FEEDER as follows. As shown in Figure 6, we place Algorithm 4 after the demonstration retriever to filter out the unnecessary parts from the retrieved data. Concretely, we first retrieve $n$ samples from our FEEDER set (i.e., $\mathcal{D}_{\text{FEEDER}}$), then filter retrieved samples by running Algorithm 4 for one round (treating the set of retrieved samples as $\mathcal{D}_{\text{IN}}$). Then, re-retrieve $n - |\mathcal{D}_{\text{OUT}}|$ where $\mathcal{D}_{\text{OUT}}$ indicates the output of Algorithm 4.

## A8 FEEDER IN FINE-TUNING SETTING

### A8.1 IMPLEMENTATION DETAILS

As summarized in Algorithm 1 in Section 2, we can integrate our FEEDER selection and LLM fine-tuning into a bi-level optimization problem. To evaluate the performance of our bi-level optimization, we first run Algorithm 1 for one run to get a pre-selected FEEDER set (i.e., $\mathcal{D}_{\text{FEEDER}}$) and a tuned LLM. Then, we update our FEEDER set with the tuned LLM and evaluate the performance of LLM in the in-context learning setting (i.e., few-shot inference), where we allow the LLM to retrieve relevant information from the pre-selected FEEDER set or the training dataset.

Concretely, our baseline is to first tune the LLM on the entire training dataset (i.e., $\mathcal{D}_{\text{TRAIN}}$) and then do few-shot inference on the test dataset (i.e., $\mathcal{D}_{\text{TEST}}$) with $\mathcal{D}_{\text{TRAIN}}$ as the retrieval pool. In contrast, ours is to first pre-select a FEEDER set (i.e., $\mathcal{D}_{\text{FEEDER}}$) from $\mathcal{D}_{\text{TRAIN}}$ and then tune the LLM on $\mathcal{D}_{\text{FEEDER}}$. Our FEEDER set is updated according to the tuned LLM using Algorithm 2 for 1 run, and our approach is evaluated on $\mathcal{D}_{\text{TEST}}$ with the updated $\mathcal{D}_{\text{FEEDER}}$ as the retrieval pool.

We conduct the fine-tuning pipeline in this manner to not only verify the superiority of our FEEDER but also to validate our bi-level optimization framework, which is able to tune both the FEEDER set and the LLMs in each loop.

We list some key hyper-parameters for fine-tuning as follows. The batch size is set as 32, the warm steps is set as 100, the learning rate is set as $5 \times 10^{-5}$, and the weight decay is set as 0.01. All our experiments are conducted with NVIDIA A100s[3].

### A8.2 ADDITIONAL RESULTS WITH DIVERSE DATASETS

We report performance comparison results on text classification datasets SUBJ, SST-2, and COLA datasets in Table 4. We include the results of FPB, SST-5, and TREC datasets in Table 7, whose trend is consistent with our analysis in Section 5.2. These results further verify the superiority of our FEEDER in the fine-tuning setting.

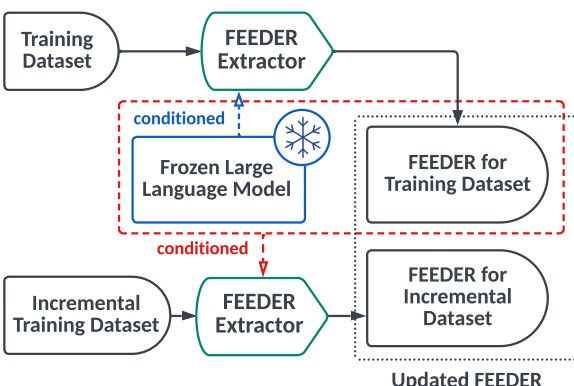

Figure 7: In order to scale up FEEDER for real-world applications dealing with dynamic data, we introduce an incremental update algorithm. This algorithm is designed to efficiently handle changes in training examples, avoiding the need to recompute over unchanged training examples.

## A9 IN-DEPTH ANALYSIS OF FEEDER

### A9.1 PERFORMANCE GAP BETWEEN USING FEEDER AND RAN AS PRE-SELECTOR

As our paper introduces a new pre-selection stage before the demonstration selection process, we also include an ablation study that randomly selects the same number of samples to form a

---

[3]https://www.nvidia.com/en-us/data-center/a100/

Table 11: Performance comparisons between using randomly-selected $\mathcal{D}^*_{\text{TRAIN}}$ (where $|\mathcal{D}^*_{\text{TRAIN}}| = |\mathcal{D}_{\text{TRAIN}}|$) as the base for acquiring a FEEDER set and using NEO for inference on SST-2, SST-5, and COLA datasets are conducted in the in-context learning setting. We report both the mean and variance of accuracy using 8 different seeds and 5 different permutations of n-shots.

| $\Psi_{\text{LLM}}(\cdot)$ | $\mathcal{D}$ | $n$ | SST-2 | | | SST-5 | | | COLA | | |
|---|---|---|---|---|---|---|---|---|---|---|---|
| | | | RAN | SIM | DIV | RAN | SIM | DIV | RAN | SIM | DIV |
| NEO (1.3B) | $\mathcal{D}_{\text{TRAIN}}$ | 2 | 76.8 (3.5) | 81.5 (0.1) | 76.3 (0.4) | 17.9 (3.6) | 26.9 (0.1) | 22.7 (0.1) | 30.7 (3.1) | 55.5 (0.2) | 56.5 (0.4) |
| | | 5 | 65.1 (3.5) | 80.8 (0.2) | 66.1 (0.3) | 19.0 (3.9) | 29.2 (0.1) | 25.1 (0.1) | 40.0 (3.6) | 55.9 (0.1) | 52.5 (0.2) |
| | $\mathcal{D}^*_{\text{TRAIN}}$ | 2 | 73.2 (3.6) | 77.8 (2.3) | 72.4 (2.4) | 14.5 (3.8) | 23.3 (3.6) | 20.0 (1.0) | 28.3 (5.4) | 48.8 (3.3) | 49.7 (3.1) |
| | | 5 | 62.4 (3.5) | 77.6 (3.3) | 62.2 (2.2) | 16.6 (2.8) | 25.5 (2.1) | 27.7 (2.8) | 33.8 (4.4) | 50.2 (3.4) | 48.7 (2.8) |
| | $\mathcal{D}_{\text{FEEDER}}$ | 2 | **75.1** (2.8) | **82.6** (2.1) | **78.5** (1.9) | **19.7** (2.7) | **27.4** (2.1) | **22.8** (1.8) | **59.3** (3.7) | **64.7** (1.4) | **64.7** (1.6) |
| | | 5 | **73.2** (4.2) | **82.9** (2.7) | **71.6** (2.4) | **19.2** (3.2) | **30.2** (1.1) | **26.4** (2.4) | **58.7** (3.2) | **67.2** (2.4) | **65.8** (1.8) |

randomly selected training dataset, denoted as $\mathcal{D}^*_{\text{TRAIN}}$, which matches the sample size of $\mathcal{D}_{\text{FEEDER}}$. The corresponding results are reported in Table 11. A comparison of Table 11 with Tables 1 and 5 indicates that replacing the entire training dataset with randomly selected samples significantly degrades LLM performance. In contrast, the FEEDER-selected samples act as a core set that summarizes the key information of the entire training dataset. By focusing on high-value samples, our approach enables LLMs to achieve better performance, effectively leveraging the essential knowledge within the dataset.

## A9.2 Performance Gap among Using Different Depth of Tree

As described in Section 4.2, we set the tree depth to 2 (corresponding to $K = 1$), utilizing the one-shot inference capability of LLMs as the sufficiency filter to eliminate unnecessary samples. To further explore the performance impact of varying tree depths, we investigate the performance gap associated with different depths of the tree. Similarly to the analysis in Section 5.2, Figure 8 visualizes the impact of employing different numbers of runs of our approximation algorithm (as outlined in Section 4.2) to derive $\mathcal{D}_{\text{FEEDER}}$ for fine-tuning NEO. For ease of comparison, the results of fine-tuning NEO on $\mathcal{D}_{\text{TRAIN}}$ are also presented as a baseline (depicted by the blue line). The results suggest that fine-tuning with a smaller, high-quality dataset can significantly enhance performance. However, when comparing to Figure 3, we observe that increasing the tree depth leads to more "smoothing" changes in the LLM performance. There are two po-

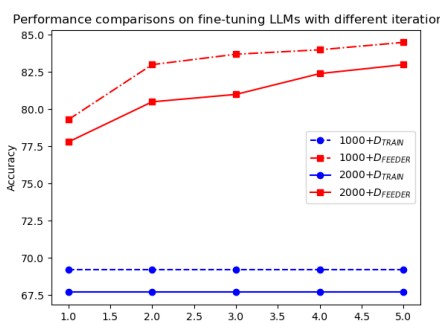

Figure 8: Performance comparisons on fine-tuning NEO with running our approximation algorithm to pre-select $\mathcal{D}_{\text{FEEDER}}$ with different iteration $K$. Our evaluation operates on COLA dataset in the zero-shot setting after fine-tuning on 1000 and 2000 batches.

tential explanations for this phenomenon: (i) The hyper-parameter $K$, which controls the tree depth, typically changes within a relatively small scope compared to $R$ due to its high computational cost and diminishing returns. While increasing $K$ initially enhances the filtering process by leveraging deeper evaluations of sufficiency, the marginal improvements in the quality or size of the resulting FEEDER set decreases as $K$ grows. (ii) Increasing the tree depth corresponds to performing n-shot inference to satisfy the sufficiency condition described in Eq. (7). This is significantly more challenging than a one-shot inference check and results in a much smaller reduction in the number of samples in the training dataset. (iii) Leveraging the n-shot inference capability of LLMs may yield more robust results. Specifically, the unnecessary samples filtered out by an n-shot sufficiency check are more likely to be genuinely unnecessary, thereby ensuring a higher-quality training set for fine-tuning.

## A9.3 Performance Gap between our Approximately Computed FEEDER Set and Exact FEEDER Set

As described in Section 4.2, our approximation algorithm ensures the sufficiency of the resulting FEEDER set but does not guarantee the necessity of each sample within it. To address this, we employ the integration method outlined in Appendix A7, which ensures that the selected demonstrations are both sufficient and necessary. We denote this refined set as $\mathcal{D}^*_{\text{FEEDER}}$. We compare the performance

Table 12: Results of performance difference between using $\mathcal{D}^*_{\text{FEEDER}}$ (derived by using FEEDER version introduced in Appendix A7), we also evaluate the performance of our variants of FEEDER with duplicated training dataset. We evaluate NEO's performance on the n-shot settings.

| $\Psi_{\text{LLM}}(\cdot)$ | $\mathcal{D}$ | $n$ | SST-2 | | | SST-5 | | | COLA | | |
|---|---|---|---|---|---|---|---|---|---|---|---|
| | | | RAN | SIM | DIV | RAN | SIM | DIV | RAN | SIM | DIV |
| NEO (1.3B) | $\mathcal{D}_{\text{TRAIN}}$ | 2 | 76.8 (3.5) | 81.5 (0.1) | 76.3 (0.4) | 17.9 (3.6) | 26.9 (0.1) | 22.7 (0.1) | 30.7 (3.1) | 55.5 (0.2) | 56.5 (0.4) |
| | | 5 | 65.1 (3.5) | 80.8 (0.2) | 66.1 (0.3) | 19.0 (3.9) | 29.2 (0.1) | 25.1 (0.1) | 40.0 (3.6) | 55.9 (0.1) | 52.5 (0.2) |
| | $\mathcal{D}'_{\text{TRAIN}}$ | 2 | 73.4 (6.6) | 78.4 (0.3) | 75.4 (2.4) | 14.9 (3.8) | 22.7 (2.9) | 21.7 (1.0) | 29.3 (5.4) | 49.8 (1.3) | 52.7 (3.3) |
| | | 5 | 59.4 (3.5) | 75.3 (1.3) | 64.1 (3.5) | 17.5 (2.8) | 23.5 (2.1) | 22.7 (2.8) | 37.8 (4.2) | 51.2 (1.4) | 51.0 (2.3) |
| | $\mathcal{D}_{\text{FEEDER}}$ | 2 | 75.1 (2.8) | 82.6 (2.1) | 78.5 (1.9) | 19.7 (2.7) | 27.4 (2.1) | 22.8 (1.8) | 59.3 (3.7) | 64.7 (1.4) | 64.7 (1.6) |
| | | 5 | 73.2 (4.2) | 82.9 (2.7) | 71.6 (2.4) | 19.2 (3.2) | 30.2 (1.1) | 26.4 (2.4) | 58.7 (3.2) | 67.2 (2.4) | 65.8 (1.8) |
| | $\mathcal{D}'_{\text{FEEDER}}$ | 2 | 74.3 (2.9) | 81.3 (1.1) | 76.4 (1.8) | 18.2 (2.2) | 26.1 (2.1) | 21.0 (1.8) | 58.3 (2.7) | 62.5 (1.4) | 63.5 (1.1) |
| | | 5 | 71.1 (3.2) | 80.0 (2.4) | 69.8 (2.1) | 19.0 (2.0) | 29.4 (1.3) | 25.3 (2.1) | 57.5 (3.0) | 65.0 (2.4) | 64.1 (1.8) |
| | $\mathcal{D}^*_{\text{FEEDER}}$ | 2 | 75.6 (1.8) | 83.1 (1.0) | 79.0 (1.1) | 20.1 (2.0) | 27.8 (2.3) | 23.1 (1.2) | 60.2 (3.2) | 64.9 (1.4) | 65.0 (1.1) |
| | | 5 | 73.7 (4.1) | 82.8 (2.2) | 71.8 (2.1) | 19.0 (3.0) | 31.2 (1.0) | 26.3 (2.1) | 59.2 (2.7) | 67.3 (2.1) | 65.4 (2.2) |
| | $\mathcal{D}^{*'}_{\text{FEEDER}}$ | 2 | 75.2 (2.0) | 82.8 (2.0) | 78.4 (1.3) | 19.9 (2.2) | 27.0 (2.1) | 22.7 (1.8) | 59.4 (1.7) | 64.9 (1.2) | 64.5 (1.2) |
| | | 5 | 73.5 (4.2) | 82.4 (2.2) | 71.3 (2.2) | 18.9 (2.2) | 29.9 (1.0) | 26.2 (1.2) | 56.5 (2.2) | 65.5 (2.2) | 64.7 (1.4) |

of few-shot preference using $\mathcal{D}_{\text{FEEDER}}$, $\mathcal{D}^*_{\text{FEEDER}}$, and $\mathcal{D}_{\text{TRAIN}}$, with the results summarized in Table 12. The results indicates that $\mathcal{D}^*_{\text{FEEDER}}$ achieves a slight improvement in LLM performance compared to $\mathcal{D}_{\text{FEEDER}}$, further validating the effectiveness of integrating sufficiency and necessity in the pre-selection process.

We further evaluate the robustness of our $\mathcal{D}^*_{\text{FEEDER}}$ and $\mathcal{D}_{\text{FEEDER}}$ by duplicating the training dataset $\mathcal{D}_{\text{TRAIN}}$. The duplicated dataset is denoted as $\mathcal{D}'_{\text{TRAIN}}$, and the corresponding resulting sets derived using our approximation and integration methods are denoted as $\mathcal{D}'_{\text{FEEDER}}$ and $\mathcal{D}^{*'}_{\text{FEEDER}}$ respectively. The results of this evaluation are summarized in Table 12. From the table, we observe that both random and similarity-based demonstration retrievers are significantly impacted by the duplicated dataset. This is because the retrieved demonstrations can include duplicates, particularly when using a similarity-based retriever, as similarity scores are calculated independently for each sample. In contrast, our $\mathcal{D}'_{\text{FEEDER}}$ and $\mathcal{D}^{*'}_{\text{FEEDER}}$ act as "weak" and "strong" filters, respectively, by effectively removing redundant or unnecessary samples from the input. The "weak" filter provided by $\mathcal{D}'_{\text{FEEDER}}$ ensures sufficiency by eliminating a significant portion of redundant data while maintaining the core information needed for the task. On other hand, the "strong" filter represented by $\mathcal{D}^{*'}_{\text{FEEDER}}$ not only ensures sufficiency but also guarantees necessity, leading to an even more refined dataset that further enhances model robustness and performance. This differentiation highlights the flexibility and effectiveness of our filtering mechanisms in handling noisy or duplicated datasets.

## A10 COMPLEXITY ANALYSIS OF FEEDER

### A10.1 TIME COMPLEXITY FOR ALGORITHM 2

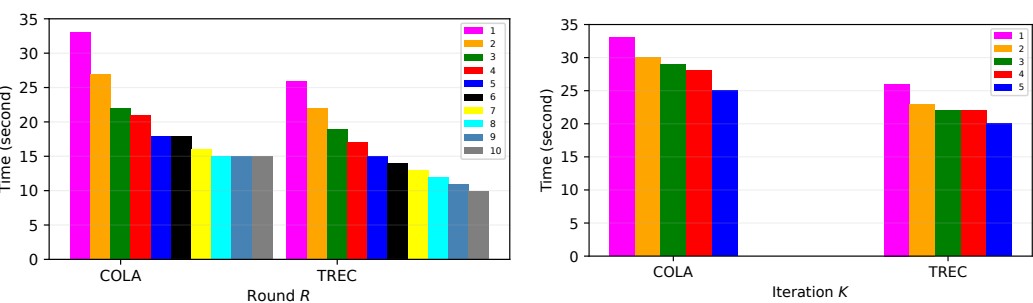

Figure 9: Time complexity of searching FEEDER using our approximation algorithm for different runs on COLA and TREC datasets using varying the number of rounds $R$ and varying the number of iterations $K$.

As summarized in Algorithm 2 and discussed in Section 4.2, there are two key hyperparameter settings for reducing the time cost of Algorithm 2: the number of iterations (i.e., $K$) and the number of rounds (i.e., $R$). In our main experiment, we set $K = 1$ and $R = 1$, meaning that we perform only

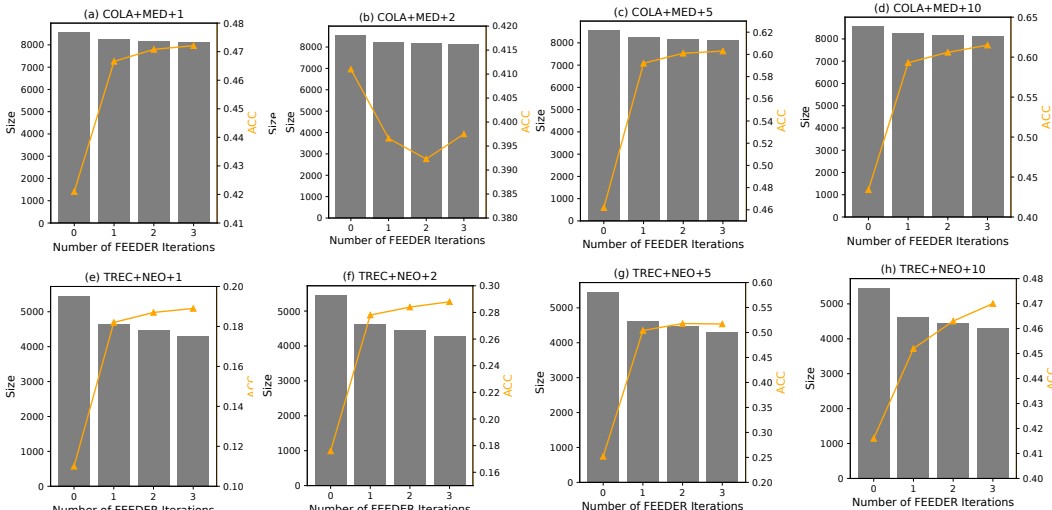

Figure 10: Performance comparisons for running our approximation algorithm to pre-select `FEEDER` with different iterations $K$ are evaluated in terms of accuracy (denoted as ACC) with `RAN` as the retriever and the size of the resulting `FEEDER` set (denoted as Size). Each sub-figure is entitled with Dataset+LLM base+n shots.

one-shot inference for sufficiency checks in each round of Algorithm 2 and execute the algorithm for a single round. We investigate the performance differences arising from varying $K$ and $R$ in Appendix A9.2 and Section 5.2 respectively. Additionally, we report the time complexity associated with different values of $K$ and $R$ on COLA and TREC datasets in Figure 9. From the figure, we observe that as the number of samples decreases, the time consumption of Algorithm 2 also decreases. Furthermore, we note that increasing the number of rounds has a great impact on reducing the time complexity. This may be attributed to the fact that two-shot inference for sufficiency-satisfying Eq. (7)-is significantly more challenging than a one-shot inference check. By further combining Figure 9 and Figure 4 in Section 5.1, we observe that the time consumption is nearly linear with respect to the size of the data samples.

## A10.2 CORRELATIONS BETWEEN TIME COMPLEXITY AND ACCURACY

Consider two hyper-parameter settings in our approximation algorithm: the number of rounds $R$ and the number of iterations $K$, both designed to balance performance and computational efficiency. As detailed in Appendix A10.1, the time complexity of our method scales almost linearly with the number of samples, making these parameters critical for practical applications. Figure 4 illustrates the performance changes across different values of $R$, while Figure 10 explores the impact of varying $K$. Interestingly, Figure 10 reveals a similar but more robust trend compared to Figure 4. This robustness could be attributed to the inherent strength of the two-shot inference process for sufficiency, as defined in Eq. (7). The two-shot inference introduces a more rigorous evaluation mechanism than the one-shot inference check, enabling a stronger filtering of unnecessary samples.

Combining all the above results, we observe that both increasing the tree depth (i.e., the number of iterations $K$) in each round and increasing the number of rounds $R$ contribute to reducing the size of the resulting `FEEDER` set. However, there are notable trade-offs between these two approaches. Increasing the tree depth is computationally more expensive but offers greater robustness, as it minimizes the risk of mistakenly filtering out useful samples. On the other hand, increasing the number of rounds is relatively inexpensive but carries a higher likelihood of discarding valuable data points due to less rigorous evaluations. In practice, we deploy our approximation algorithm with $K = 1$ and $R = 1$, which provides an optimal trade-off between computational efficiency and model performance. This configuration ensures that the pre-selection process remains practical while maintaining competitive accuracy.

## A11  Case Study with Artificial Data Points Generated by LLMs

### A11.1  Case Study for Transitivity of LLMs

To illustrate the transitivity of LLMs, we conducted a simple experiment using `gpt-3.5-turbo`. We prompted the model with the question *which place does Jerry lives in?* LLM responses with *I'm sorry, but I don't have access to personal information about individuals, including your friend Jerry.* Then, let $\mathcal{D}_A, \mathcal{D}_B, \mathcal{D}_C$ denote the city, the country, and the continent he lives in. Then, we can observe that if we tell the LLM about the city (e.g., London), then the LLM can tell about the country (e.g., United Kingdom); if we tell the LLM about the country (e.g., United Kingdom), then the LLM can tell about the continent (e.g., Europe). Also, if we tell the LLM about the city (e.g., London), then we ask about the continent, and the LLM also can tell (e.g., Europe). This observation is one case to demonstrate the transitivity of sufficiency.

### A11.2  Case Study for Demonstrations Selection Should be Aware of LLMs in Use

Subsequently, we conduct a case study to substantiate the central proposition of this paper: Whether the assessment of the quality of a demonstration should depend on the specific LLM in use.

We consider the factual error made by Google Bard in the first demo[4]. Bard said *The James Webb Space Telescope took the very first pictures of a planet outside of our own solar system.* However, the fact is *The Very Large Telescope took the very first pictures of a planet outside of our own solar system.* Based on the above statements, we produce the question *What took the very first pictures of a planet outside of our own solar system?*

We ask the question to `gpt-3.5-turbo`. The corresponding answer is:

*The first pictures of a planet outside of our own solar system were taken by the Hubble Space Telescope in 2008. The planet, named Fomalhaut b, orbits the star Fomalhaut, which is located about 25 light-years away from Earth in the constellation Piscis Austrinus.*

We can see that the above answer is incorrect. We prompt the fact into the LLM, then the answer is:

*The Very Large Telescope took the very first pictures of a planet outside of our own solar system.*

This indicates that prompting the fact works. We further ask `gpt-3.5-turbo` to generate some sufficient statements for the fact. The following is its output:

*1. The Very Large Telescope made history by capturing the first-ever images of a planet beyond our solar system.*

*2. Thanks to the advanced technology of the Very Large Telescope, scientists were able to capture the first-ever photographs of an exoplanet.*

*3. The groundbreaking images captured by the Very Large Telescope marked a major milestone in the study of exoplanets.*

*4. The discovery of a planet outside of our solar system was made possible by the cutting-edge capabilities of the Very Large Telescope.*

*5. The Very Large Telescope's ability to capture images of distant celestial bodies allowed scientists to observe an exoplanet for the first time in history.*

We also evaluate separately using the above statements as a prompt to `gpt-3.5-turbo`, and either one of them can lead to the correct answer. We provide detailed answers as follows.

*1. The Very Large Telescope took the very first pictures of a planet outside of our own solar system.*

*2. The Very Large Telescope took the very first pictures of a planet outside of our own solar system.*

*3. The Very Large Telescope took the very first pictures of a planet outside of our own solar system.*

---

[4]https://www.theverge.com/2023/2/8/23590864/google-ai-chatbot-bard-mistake-error-exoplanet-demo

*4. The Very Large Telescope took the very first pictures of a planet outside of our own solar system.*

*5. The Very Large Telescope took the very first pictures of a planet outside of our own solar system.*

We can see, in this case, that either one of the statements generated by `gpt-3.5-turbo`, is a sufficient and necessary instance to answer *What took the very first pictures of a planet outside of our own solar system?*

We further evaluate the performance of `gpt-j-6b`. Without any prompt (i.e., in the zero-shot setting), its answer is:

*The Hubble Space Telescope.*

We then prompt the above 5 statements provided by `gpt-3.5-turbo` into `gpt-j-6b`, then the answer is:

*1. The first-ever images of a planet beyond our solar system have been captured by the Very Large Telescope in Chile.*

*2. The Hubble Space Telescope.*

*3. A team of astronomers led by the University of Arizona's Michael Liu.*

*4. The Hubble Space Telescope.*

*5. The Very Large Telescope.*

We can see that only the 1-st statement or the 5-th statement is a sufficient and necessary instance. Combining the results of `gpt-j-6b` and the results of `gpt-3.5-turbo` can verify the core insight of our paper: *the measurement over prompting a demonstration should consider what LLM is in use.*

Furthermore, we also evaluate the performance of 3 GPT variants used in the paper. We begin by evaluating the zero-shot performance of `gpt-neo-1.3B` (denoted as `NEO` in the experiment), and its result is:

*The first pictures of a planet outside of our own solar system were taken by the Voyager 1 spacecraft in 1977.*

We then prompt the above 5 statements into `gpt-neo-1.3B`, then the answer is:

*1. The very large telescope was built in the early 1990s by the European Southern Observatory (ESO) in Chile. The Very Large Telescope.*

*2. The Very Large Telescope in Chile.*

*3. The Very Large Telescope (VLT) in Chile.*

*4. The Very Large Telescope.*

*5. The Very Large Telescope in Chile.*

The above results show that either one of the latter 4 statements is a sufficient instance. The results of `gpt2-large` (denoted as `LAR` in the experiment) show that neither of the 5 statements is a sufficient and necessary instance:

*1. The very large telescope was built in the early 1990s by the European Southern Observatory (ESO) in Chile. The Very Large Telescope.*

*2. The Hubble Space Telescope.*

*3. The first pictures of a planet outside of our own solar system were taken by the Hubble Space Telescope in 1990.*

*4. The Hubble Space Telescope.*

*5. The very first pictures of a planet outside of our own solar system were taken by the Hubble Space Telescope.*

The results of `gpt2-medium` (denoted as `MED` in the experiment) show that only the 4-th statement is not a sufficient and necessary instance:

*1. The Very Large Telescope.*

*2. The Very Large Telescope.*

*3. The Very Large Telescope.*

*4. The Hubble Space Telescope.*

*5. The Very Large Telescope.*

All the above results verify that *quality* of one demonstration should be LLM-specific, which is the key idea of our paper.

## A12    LIMITATION AND IMPACT STATEMENTS

Notice that our FEEDER serves as a general demonstration pre-selector capable of enhancing the performance of various LLMs while simultaneously reducing computation costs. Due to budget limitations, our paper presents results only for LLMs with up to 10B parameters for in-context learning evaluation and up to 2B parameters for the fine-tuning setting. It would be worthwhile to investigate the performance of our FEEDER with larger LLMs and employing a greater number of shots. Due to computation limitations and budget constraints, we leave this exploration for future work.

The objective of this paper is to develop a pre-selection method over the training dataset as an intermediary process to enhance the accuracy of factual knowledge in the model's outputs. Consequently, our method is designed to enhance the faithfulness of LLM systems. It is essential to note that our FEEDER, selected from the training dataset without external trustworthy corpora, relies on the capability of the given LLM itself. This characteristic may potentially amplify existing biases in the model weights of LLMs.

