# OpenReview forum: "Large Language Models are Demonstration Pre-Selectors for Themselves"
_ICLR.cc/2025/Conference — Submitted to ICLR 2025_

### Official Review · Reviewer_CV5L · 2024-11-02

**Soundness:** 3
**Presentation:** 3
**Contribution:** 2
**Rating:** 6
**Confidence:** 2

**Summary:**

This paper presents FEEDER, a framework for improving in-context learning with large language models by pre-selecting a core subset of essential demonstrations. Through the concepts of "sufficiency" and "necessity," FEEDER effectively reduces dataset size, leading to enhanced efficiency in both few-shot inference and fine-tuning settings while maintaining or improving model performance.

**Strengths:**

1. The paper introduces FEEDER, a demonstration pre-selector that optimizes in-context learning by identifying a core set of examples that capture "sufficiency" and "necessity." This subset enhances efficiency by reducing data size without sacrificing performance.
2.  Experiments on various datasets (text classification, reasoning, and semantic parsing) demonstrate the framework's effectiveness across multiple LLMs.

**Weaknesses:**

1. The paper's explanations lack clarity and specificity in certain areas. For instance:
(1) When describing the poor demonstration of some examples, the authors provide general statements without detailing specific cases or examples. Adding concrete instances would help readers better understand and contextualize the problem.
(2) The terms "sufficiency" and "necessity" lack clear definitions, which are essential for understanding the FEEDER framework.
2. Although the paper frequently emphasizes the tree-based approximation algorithm, the selection process remains time-consuming. It would be beneficial for the authors to include a comparison of computational efficiency, including time costs and accuracy, between FEEDER-selected samples and randomly selected samples of the same size in an in-context learning setting. This could help justify the trade-offs between computational efficiency and the benefits gained from the FEEDER method.
3. The paper lacks an in-depth analysis of certain observations from the experimental results. For example, Table 1 shows that some datasets perform better with larger sample sizes (i.e. n), while others achieve better results with fewer samples. It is unclear whether these discrepancies arise from dataset characteristics or other factors.
4. The experimental results indicate that the FEEDER set and direct sample selection from the training set yield similar performance trends with respect to changes in n, which is inconsistent with the "necessity" mentioned by the authors. Given that FEEDER sets are meant to fulfill "necessity," it would be logical for performance to improve as the number of examples increases. However, this anticipated trend is not clearly reflected in the results. Further investigation into this issue is warranted, particularly to clarify why the FEEDER set’s performance does not show a clear improvement with increasing sample sizes.

**Questions:**

1. Although the paper frequently emphasizes the tree-based approximation algorithm, the selection process remains time-consuming. It would be beneficial for the authors to include a comparison of computational efficiency, including time costs and accuracy, between FEEDER-selected samples and randomly selected samples of the same size in an in-context learning setting. This could help justify the trade-offs between computational efficiency and the benefits gained from the FEEDER method.
2. The paper lacks an in-depth analysis of certain observations from the experimental results. For example, Table 1 shows that some datasets perform better with larger sample sizes (i.e. n), while others achieve better results with fewer samples. It is unclear whether these discrepancies arise from dataset characteristics or other factors.
3. The experimental results indicate that the FEEDER set and direct sample selection from the training set yield similar performance trends with respect to changes in n, which is inconsistent with the "necessity" mentioned by the authors. Given that FEEDER sets are meant to fulfill "necessity," it would be logical for performance to improve as the number of examples increases. However, this anticipated trend is not clearly reflected in the results. Further investigation into this issue is warranted, particularly to clarify why the FEEDER set’s performance does not show a clear improvement with increasing sample sizes.

---

> ### Author Response · Authors · 2024-11-28
> **Response to Review CVSL 1/2**
>
> We thank reviewer CVSL for the constructive feedback and recognition of our idea, method and experiments on various datasets. We respond to each of the reviewer's questions in the sections below and also please see the main comments above.
>
>
> > The paper's explanations lack clarity and specificity in certain areas. For instance: (1) When describing the poor demonstration of some examples, the authors provide general statements without detailing specific cases or examples. Adding concrete instances would help readers better understand and contextualize the problem. (2) The terms "sufficiency" and "necessity" lack clear definitions, which are essential for understanding the FEEDER framework.
>
> We have included illustrative cases in the appendix to demonstrate our concepts and method: (i) Conceptual cases: in Appendix A2, we present cases for the concepts of "sufficient instance/set", "necessary instance/set", and "sufficienct and necessary instance/set". Additionally, we discuss and exemplify the relationship between our "FEEDER set" and the "sufficient set" and "necessary set", highlighting their theoretical connections. (ii) Case study with artifical data: in Appendix A11, we include a case study using artifical data points generated by LLMs. This study demonstrates the  transitivity property of LLMs and supports our proposition that the selected demonstrations should be tailored to the specific LLM in use.
>
> > Although the paper frequently emphasizes the tree-based approximation algorithm, the selection process remains time-consuming. It would be beneficial for the authors to include a comparison of computational efficiency, including time costs and accuracy, between FEEDER-selected samples and randomly selected samples of the same size in an in-context learning setting. This could help justify the trade-offs between computational efficiency and the benefits gained from the FEEDER method.
>
> Thanks for your suggestion. We have provided a comparison between FEEDER-selected samples and randomly selected samples of the same size in Appendix A9.1 (Table 11). These results further validate the superiority of our FEEDER method in identifying informative samples from the entire training dataset, demonstrating its effectiveness in enhancing model performance.
>
> > The paper lacks an in-depth analysis of certain observations from the experimental results. For example, Table 1 shows that some datasets perform better with larger sample sizes (i.e. n), while others achieve better results with fewer samples. It is unclear whether these discrepancies arise from dataset characteristics or other factors.
>
> As summarized in [1], the performance of in-context learning is highly sensitive to specific settings. It is not solely determined by demonstration selection but is also influenced by factors such as the nature if the task, the dataset, and the order of demonstration examples. Here, in text classification tasks, increasing $n$ (the number of demonstrations) can provide more information but may also introduce noise, negatively affecting performance. Conversely, in reasoning and semantic-parsing task, the data samples are typically clean, but other factors such as the squeezed token limits of response and context window size [2] can also influence the model performance. As our primary focus is on the proposed pre-selection process, we provide an in-depth analysis of the influence of each component, along with their associated time costs in Appendices A9 and A10.
>
> [1] A Survey on In-context Learning. 2022.
>
> [2] Boosting LLM Reasoning: Push the Limits of Few-shot Learning with
> Reinforced In-Context Pruning. 2023.

---

> ### Author Response · Authors · 2024-11-28
> **Response to Review CVSL 2/2**
>
> > The experimental results indicate that the FEEDER set and direct sample selection from the training set yield similar performance trends with respect to changes in n, which is inconsistent with the "necessity" mentioned by the authors. Given that FEEDER sets are meant to fulfill "necessity," it would be logical for performance to improve as the number of examples increases. However, this anticipated trend is not clearly reflected in the results. Further investigation into this issue is warranted, particularly to clarify why the FEEDER set’s performance does not show a clear improvement with increasing sample sizes.
>
> Our paper involves two evaluation tasks. The first is in-context learning, which, as discussed above, is influenced not only by the demonstration selection but also by other factors, such as the order of the demonstrations. In this context, introducing redundant or noisy samples could negatively impact the performance of the LLM. The second task is fine-tuning. As discussed in Section 2, our approach operates as a core-set selection method and aligns with the active learning principle. This allows the LLM to prioritize learning from high-valued (i.e., FEEDER-selected) samples that effectively summarize the knowledge contained in the entire training dataset. By focusing on these representative samples, our method enhances the fine-tuning efficiency and overall model performance. We provide an in-depth analysis of the design choices in Appendix A9 and present case studies in Appendix A11 to further explore the underlying mechanisms of FEEDER.

---

> > ### Comment · Reviewer_CV5L · 2024-11-29
> >
> > Authors well addressed my concerns, so I raise my score. I encourage authors to add these discussion in the final version.

---

> ### Author Response · Authors · 2024-12-02
> **Thank you for your positive feedback and support**
>
> We appreciate your recognition of our efforts in addressing your concerns. In the final version, we will include the discussions and enhancements as suggested to further strengthen the clarity and depth of our work.

---

### Official Review · Reviewer_RMS6 · 2024-11-02

**Soundness:** 1
**Presentation:** 2
**Contribution:** 1
**Rating:** 3
**Confidence:** 4

**Summary:**

This paper introduces a novel pre-selection framework called FEEDER, designed to identify a core subset of examples that exhibit the highest levels of "sufficiency" and "necessity." By using only this subset, FEEDER can effectively replace the original dataset, improving both efficiency and predictive accuracy in few-shot in-context learning.

**Strengths:**

1. This paper presents a new metric system based on "sufficiency" and "necessity" to evaluate the importance of each example.
2. This paper proposes a tree-based search method to identify the core subset, which is novel to the existing literature.

**Weaknesses:**

1. Even in the APPROXIMATION version presented in Section 4.2, achieving precise accuracy with terms like $1_{|W_{in}|}$ and $1_{|W_{out}|}$ in Equations 7 and 8 seems challenging. The current benchmark may very struggle to achieve 100% accuracy, raising questions about this method’s effectiveness and feasibility.

2. As the number of samples within each node grows, conducting few-shot tests on these samples becomes increasingly difficult. For instance, if the limit for few-shot samples is set to n=5, handling a node containing 1,000 samples would be problematic.

3. Most of the benchmarks used in this study, aside from GSM8k, are based on sentiment classification and linguistic analysis, which are proposed before 2018. These benchmarks are somewhat outdated and do not adequately represent the benchmarks currently prioritized by the research community.

4. Besides Llama 7B and Gemma-2 2B, the models tested are relatively outdated (before 2020) and have limited parameter sizes. This choice of models and benchmarks may reduce the impact and relevance of the paper’s conclusions for the present-day community.

**Questions:**

N/A

---

> ### Author Response · Authors · 2024-11-28
> **Response to Reviewer RMS6**
>
> We thank reviewer RMS6 for their suggestions and appreciate their recognition that our method is novel. We will address each of the questions posed by the reviewer in the sections below and also please see the main comment above.
>
> > Even in the APPROXIMATION version presented in Section 4.2, achieving precise accuracy with terms like $I_{W_{IN}}$ and $I_{W_{OUT}}$ in Equations 7 and 8 seems challenging. The current benchmark may very struggle to achieve 100% accuracy, raising questions about this method’s effectiveness and feasibility.
>
> Thank you for your feedback. Our FEEDER discovery process, including the approximation algorithm (i.e., Algorithm 2) involves invoking LLMs to evaluate sufficiency. To reduce the time cost, we introduce two key hyper-parameter setting: (i) limiting the tree depth to 2 (i.e., $|I_{W_{IN}}|$ = $|I_{W_{OUT}}|$ =1 and the number of iteration $K=1$) and (ii) running Algorithm 2 in a single round (i.e., $R=1$). These adjustments effectively reduce the overall computational complexity to $O(log_2|D_{TRAIN}|)$. We have clarified this further in Section 4.2 of the revised manuscript. Additionally, to provide a comprehensive understanding of the computational requirements, we detail the time complexity and performance difference of varying $K$ and $R$ in FEEDER in Figures 3, 8 and 9.
>
> > As the number of samples within each node grows, conducting few-shot tests on these samples becomes increasingly difficult. For instance, if the limit for few-shot samples is set to n=5, handling a node containing 1,000 samples would be problematic.
>
> That is right. Handling nodes with more than 1000 samples would be impractical. To address this, as mentioned earilier, we limit the tree depth to 2 in our pre-selection process, ensuring that each node contains only 1 or 2 samples. Specifically, after applying the procedures outlined in Eqs. (7) and (8), all the remaining nodes are merged to form the resulting set. This approach ensures that the process remains computationally efficient while maintaining the integrity of the selected samples. We also provid detailed algorithms for computing the exact FEEDER in Appendix A4 and discuss its deployment in Appendix A7. The results in Appendix A9.3 (Table 12) along with Tables 1, 2, 3 and 4 show that our approximation algorithm with $K=1$ and $R=1$ provides an optimal trade-off between computational efficiency and model performance.
>
> > Most of the benchmarks used in this study, aside from GSM8k, are based on sentiment classification and linguistic analysis, which are proposed before 2018. These benchmarks are somewhat outdated and do not adequately represent the benchmarks currently prioritized by the research community.
>
> We acknowledge your observation. Although our text classification datasets were proposed before 2018, some of these datasets would be valuable to revisit and investigate in light of more recent developments especially for LLMs under 10B [1]. However, as you suggest, we are open to incorporate additional datasets that align with recent research focus. If you have specific recommendations for recently proposed benchmarks or datasets, we would appreciate your advice to enhance te relevance and impact of our evaluation.
>
> [1] Text Classification via Large Language Models. 2023.
>
> > Besides Llama 7B and Gemma-2 2B, the models tested are relatively outdated (before 2020) and have limited parameter sizes. This choice of models and benchmarks may reduce the impact and relevance of the paper’s conclusions for the present-day community.
>
> We have included the Llama-3 8B model [2] as a new baseline in our experiments, which achieves impressive performance (around 80% accuracy) on the GSM89K dataset. The corresponding results are reported in Table 3, demonstrating that our approach generalizes effectively to powerful LLMs. This further validates the scalability and versatility of our method across different model architectures.
>
> [2] https://ai.meta.com/blog/meta-llama-3/

---

> ### Author Response · Authors · 2024-12-02
>
> Dear Reviewer RMS6,
>
> As the discussion deadline approaches, we are wondering whether our responses have properly addressed your concerns. Your feedback would be extremely helpful to us. If you have further comments or questions, we hope for the opportunity to respond to them.
>
> Many thanks,
>
> Authors

---

> ### Author Response · Authors · 2024-12-03
>
> Once again, we are grateful for your time and effort in reviewing our paper. Since the discussion period will end in around a day, we are very eager to get your feedback on our response.
>
> To facilitate your review, we further summarize our responses as follows:
> - (i) Regarding the complexity and feasibility of assessing sufficiency and necessity, we limit the iterations and the number of rounds to 1, making the process practical and computationally efficient. Our empirical results, as presented in Tables 1, 2, 3, 4, and 12, demonstrate that even with this simplified setting, our approach achieves strong performance.
> - (ii) Regarding our use of LLMs, we have incorporated a recently proposed, powerful LLM, LLaMA3-8B, into our experiments. The results in Table 3 further validate the superiority of our approach, especially when leveraging more advanced LLMs.
>
> Thank you again for your time and consideration. We look forward to your feedback.
>
> Sincerely,
>
> Authors

---

> > ### Author Response · Authors · 2024-12-04
> >
> > Dear reviewer RMS6,
> >
> > Since the discussion period will end in a few hours, we will be online waiting for your feedback on our rebuttal, which we believe has fully addressed your concerns.
> >
> > We would highly appreciate it if you could take into account our response when updating the rating and having discussions with AC and other reviewers.
> >
> > Thank you so much for your time and efforts. Sorry for our repetitive messages, but we're eager to ensure everything is addressed.
> >
> > Authors of # 13601

---

### Official Review · Reviewer_1pew · 2024-11-04

**Soundness:** 3
**Presentation:** 3
**Contribution:** 3
**Rating:** 6
**Confidence:** 4

**Summary:**

This paper addresses examples (a.k.a. demonstrations) selection for in-context learning (ICL) and Fine-tuning. The authors build on the hypothesis that a small set of highly informative examples can often encapsulate enough information to infer many less informative examples. This allows for the use of these smaller, more informative samples in ICL (or fine-tuning). The authors then propose a framework that defines "sufficient" and "necessary" samples for model prediction, where "sufficiency" implies that a sample enhances model prediction accuracy, and "necessity" means that a sample is required for the model to make correct predictions. The authors introduce a pre-selection framework to “select” necessary and sufficient samples from a training set (termed the "FEEDER set"), using a tree-based merging mechanism at each tree layer. Moreover, the authors propose a bi-level optimization method for fine-tuning based on FEEDER using a frozen LLM in the outer loop and optimizing LLM training in the inner loop.

Finally, authors demonstrate that the FEEDER set, across six different classification tasks, reasoning and semantic-parsing datasets and six large language models (LLMs), consistently provides superior performance.

**Strengths:**

Considering that LLMs are resource-intensive, selecting the most effective samples from a large training set is a reasonable approach to improve "efficiency and accuracy" in inference (for few-shot learning) and training (for fine-tuning). To find the right selector, the authors first define "necessary" and "sufficient" instances, extending these definitions to necessary and sufficient sets. Based on this concept, they define $D_{\text{FEEDER}}$ as a subset of $D_{\text{TRAIN}}$ that is both sufficient and necessary. A brute-force approach to find $D_{\text{FEEDER}}$ would be expensive, so the authors propose a tree-based merge technique that builds from the bottom up, merging examples into nodes based on their sufficient and necessary characteristics at each tree layer. They also claim that the approach has low complexity, yielding $D_{\text{FEEDER}}$ in a reasonable number of iterations. ICL evaluation results suggest that FEEDER is an effective pre-selector (or "compressor") of demonstrations and can benefit from various demonstration selectors. Additionally, for fine-tuning, the authors formulate sample selection as a bi-level optimization problem. In the outer layer, it identifies a “high-quality” sample set using a frozen LLM, while in the inner layer, it optimizes LLM training. In experimentation, authors show that FEEDER achieves improvements compared to fine-tuning with entire training set.

**Weaknesses:**

While the paper’s motivation is reasonable and the argument flow is sound, there are some gaps in the authors' reasoning throughout.

**Ambiguous Goal (Accuracy vs. Efficiency)**: It's unclear whether the authors are primarily focused on improving model accuracy or throughput (cost) with FEEDER. Throughout, they claim the approach improves efficiency and prediction accuracy, but for ICL, there is no evidence provided for how FEEDER enhances model efficiency.

**Computation cost for FEEDER discovery**: The FEEDER discovery algorithm requires calling the LLM at each stage to assess the sufficiency relationship between datapoints in $\mathscr{W}_{k-1}$, which appears computationally expensive and potentially impractical.

**Complexity and Scalability Concerns**: While the authors state that the FEEDER algorithm requires $O(K \log_2 |D_{\text{TRAIN}}|)$ iterations, each iteration involves comparing sufficient and necessary criteria between "every pair," resulting in an overall complexity of $O(K |D_{\text{TRAIN}}| \log_2 |D_{\text{TRAIN}}|)$.

**Questions:**

Q1: In definitions 1 and 2, can  $(x_n, y_n)$ be $(x_m, y_m)$? In other words, can a sample be considered its own sufficient and necessary sample?

Q2: Since the ICL is limited to 5 to 10 examples, could author provide the results based on brute-force approach where we search across $plug$ and $unplug$ (as defined in Definition 1 and 2) for entire $D_\text{TRAIN}$

---

> ### Author Response · Authors · 2024-11-28
> **Response to Reviewer 1pew**
>
> We appreciate reviewer 1pew’s constructive feedback and positive comments on our idea, technical contributions, and experiments contributions. We respond to each of the reviewer's questions in the following and also please see the main comments above.
>
> > In Definition 1, can a sample be considered its own sufficient and necessary sample?
>
> According to the definitions of sufficiency and necessity (as described in Definitions 1 and 2), since LLMs are capable of fully memorizing the facts presented in the input ($x_n, y_n$), any sample can be considered sufficient on its own. However, it may not be necessary, as the LLM may already posses knowledge of the fact without requiring the explicit input ($x_n, y_n$).
>
> > Since the ICL is limited to 5 to 10 examples, could author provide the brute-force approach where we search across plug and unplug.
>
> Since ICL performance is influenced not only by the selector but also by the order of demonstrations [1], implementing a brute-force approach to identify the optimal demonstration set would be impractical due to the exponential growth of possible combinations.
>
> [1] A Survey on In-context Learning. 2022.
>
> > It is unclear whether the authors are primarily focused on improving model accuracy and there is no evidence provided for how FEEDER enhances the model efficiency.
>
> Thanks for your feedback. Our primary focus is to introduce a new stage, termed the pre-selection stage, to eliminate abundant and unnecessary training samples. The benefits of this approach are twofold: (i) Efficiency: ICL requires less time for similarity-based, diversity-based, or learning-based retrieval methods due to a smaller retrieval pool (the resulting FEEDER set v.s. the entire training dataset), and fine-tuning operates on a reduced dataset, saving significant computational resources. (ii) Improved Performance: Our method enhances ICL accuracy in most cases and significantly improves fine-tuning accuracy (as evidenced in Tables 1, 2, and 3).
>
> We acknowledge that introducing a pre-selection stage incurs an additional computational cost (e.g., our pre-selection stage on the GSM8K dataset using the Llama-3 8B model takes 5 hours, while the inference stage takes 2 hours). However, we argue that the pre-selection process can be performed in a "once-for-all" manner: once the FEEDER set is computed, it can be reused to serve all test samples. As the model serves more online test samples over time, the amorized cost of pre-selection per test sample decreases. Furthermore, the time savings from operating on a smaller ICL pool grow over time.
>
> > The FEEDER discovery requires calling LLM at each stage.
>
> Our FEEDER discovery process, including the approximation algorithm (i.e., Algorithm 2), requires invoking LLMs to evaluate sufficiency, which can be computationally intensive. To mitigate this complexity, we introduce two key hyper-parameter setting: (i) limiting the tree depth to 2 (i.e., $K=1$) and (ii) running Algorithm 2 in a single round. These adjustments reduce the overall computational complexity to $O(log_2|D_{TRAIN}|)$. We have further clarfied this further in Section 4.2 of the revised manuscript. Additionally, to provide a comprehensive understanding of the computational demands, we detail the time complexity of each operation in FEEDER and analyze the relationship between model accuracy and time complexity in Appendix 6.
>
> > While the author state tht the FEEDER algorithm requires $O(log_2|D_{TRAIN}|)$ iterations, each iteration involves comparing sufficient and necessary criteria between "every pair", resulting in an overall complexity of $O(K|D_{TRAIN}|log_2|D_{TRAIN}|)$.
>
> As discussed in Section 4.2, lines 260-261, "After performing the above calculation for each pair, we remove them from $W_{k-1}$", and thereby our approximation algorithm does not examine all pairs due to this removal operation. This design reduces the overall complexity of our approximation algorithm to $O(K log_2|D_{TRAIN}|)$ for each round of computation. Additionally, we have provided detailed algorithms for computing the exact FEEDER in Appendix A4 and discuss its deployment in Appendix A7. As mentioned earlier, our empirical results demonstrate that the approximated FEEDER, with one iteration (i.e., $K=1$) and a single round of computation (i.e., $R=1$), already achieves strong performance. This highlights the efficiency and effectiveness of our approximation algorithm in practice.

---

> ### Author Response · Authors · 2024-12-02
>
> Dear Reviewer 1pew,
>
> As the discussion deadline approaches, we are wondering whether our responses have properly addressed your concerns. Your feedback would be extremely helpful to us. If you have further comments or questions, we hope for the opportunity to respond to them.
>
> Many thanks,
>
> Authors

---

### Official Review · Reviewer_Y7CH · 2024-11-04

**Soundness:** 2
**Presentation:** 3
**Contribution:** 2
**Rating:** 6
**Confidence:** 3

**Summary:**

This paper proposes a demonstration pre-selection framework. Specifically, it considers both sufficiency and necessity conditions and introduces an approximate tree-based selection method. The author conducts extensive experimental validation under both in-context learning and fine-tuning settings.

**Strengths:**

1. The paper is well written and easy to follow;
2. The idea of iterative optimization is reasonable;
3. The extension from instance level to set level and the approximation method based on transitivity are novel.

**Weaknesses:**

1. Lack of experiments with larger-scale models. For example, the  original metrics on GSM8k is quite low. Although Section 5.3 mentions this issue, the scale of the models used in the corresponding experiments is still too small. The performance of few-shot selection for larger-scale models may differ from that of smaller models, and it is uncertain whether this method can be scaled up to larger models.
2. Regarding sufficiency and necessity, more ablation studies are needed to prove that both aspects are important.
3. In the implementation of tree construction, the different design choices of merging nodes among adjacent layers requires more experimental comparisons.

**Questions:**

Please see the weaknesses.

---

> ### Author Response · Authors · 2024-11-28
> **Response to Reviewer Y7CH**
>
> We would like to thank reviewer Y7CH’s feedback on our paper is well-written, idea is reasonable and novel. We respond to each of the reviewer's questions in the following and also please see the main comments above.
>
> > Lack of experiments with larger-scale models, e.g., the original metric on GSM8K is quite low.
>
> Thanks for your feedback. In response to your concern, we applied our FEEDER to the recently proposed Llama 3 (8B) model [1], which can achieve around 80% accuracy on the GSK8K benchmark. The corresponding results are summarized in Table 3, alongside outcomes from the SMCalFlow dataset. These findings demonstrate that our FEEDER performs effectively when integrated with advanced LLMs, further validating its scalability and robustness.
>
> [1] https://ai.meta.com/blog/meta-llama-3/
>
> > Regarding sufficiency and necessity, more ablation studies, are needed to prove that both aspects are important.
>
> As discussed in Section 4.2, sufficiency and necessity are inherently connected "samples that can be removed from the set without compromising its sufficiency to represent the entire training dataset are unnecessary", Our approximation algorithm is specifically designed to filter out these unnecessary samples while maintaining the sufficiency of the remaining subset. As described in Proposition 1, our approximation algorithm (i.e., Algorithm 2) ensures the sufficiency of the resulting set but does not guarantee the necessity. In contrast, our exact FEEDER algorithm (i.e., either Algorithm 3 or 4) is capable of identifying both sufficient and necessary samples. To illustrate the performance gap between the approximation and exact methods, we conduct experiments in Appendix A9.3 (Table 12), demonstrating that the approximation approach achieves performance close to that of the exact algorithm, capturing most of its benefits.
>
> > In the implementation of tree construction, the different design choices of merging nodes among adjacent layers require more experimental comparisons.
>
> Our merging strategy is derived from the observed transitivity of LLMs, as noted in [2]. The primary design choice in this approach is the size of the corpus of data points evaluated together, which directly influences the depth of the tree in our approximation algorithm. To investigate the impact of this design choice, we conducted empirical evaluations of increasing tree depths. The results, reported in Appendices A9.2 and A10.1 (Figures 8 and 9), show that increasing the tree depth can slightly improve performance but significantly increases time complexity. This finding further validates that our approximated FEEDER, with one iteration (i.e., $K=1$) and a single round of computation (i.e., $R=1$), achieves strong performance in balancing the trade-off between the efficiency and effectiveness of our approximation algorithm in practice.
>
> [2] Consistency Analysis of ChatGPT. 2023.

---

> ### Author Response · Authors · 2024-12-02
>
> Dear Reviewer Y7CH,
>
> As the discussion deadline approaches, we are wondering whether our responses have properly addressed your concerns. Your feedback would be extremely helpful to us. If you have further comments or questions, we hope for the opportunity to respond to them.
>
> Many thanks,
>
> Authors

---

### Author Response · Authors · 2024-11-28
**General Responses to all reviewewrs**

We first summarize our response and the results of additional suggested experiments here. We have also responded to the specific concerns of each reviewer as individual comments below.

All reviewers agree that our idea is interesting and we would like to further clarify our contributions as follows.
1. FEEDER serves as a demonstration pre-selector designed to identify a subset of informative samples from the training dataset. Importantly, our FEEDER operates in an "once-for-all" manner: once constructed, it can serve any test data point without requiring additional computations.
2. To address the high computational cost of computing a FEEDER set, we designed an efficient tree-based approximation algorithm. This algorithm reduces computational overhead by selectively removing "unnecessary" parts of the dataset while preserving the "sufficiency" of the remaining part. Our empirical results further support the efficiency of this approach, showing that limiting the tree depth to two (i.e., the number of iterations $K=1$) and running the algorithm for a single round (i.e., the number of rounds $R=1$) can reduce the training dataset size by nearly 20% (see Figure 4), meanwhile improving the few-shot performance (see Tables 1, 2, 3) and the fine-tuning performance (see Table 4).

We then summarize the results of the suggested experiments as follows.
1. We appreciate the valuable suggestions from reviewers Y7CH and RMS6 to apply FEEDER to larger models to strengthen our paper. In response, we conducted additional experiments with FEEDER applied to Llama 3 (8B), a recently proposed and powerful LLM, on the GSM8K and SMCALFlow datasets. The results, presented in Table 3, demonstrate that FEEDER performs effectively when integrated with advanced LLMs.
2. In response to reviewers 1pew, RMS6, and CVSL, we have provided more detailed results on the time complexity of varying the number of iterations $K$ and the number of rounds $R$ in FEEDER, as presented in Appendix A10. Our empricial results in Tables 1, 2, 3 and 4 demonstrate that FEEDER achieves strong performance with a limited tree depth of two and a single round of running our approximation algorithm.
3. As suggested by reviewers Y7CH and CSVL, we incorporated the following additional ablations as baselines to strengthen our in-depth analysis: (i) employing a random strategy as the pre-selector to evaluate the effectiveness of our method compared to a baseline with no informed selection (see Table 11, Appendix 9.1), and (ii) examining the impact of tree depth in our approximation algorithm to better understand its influence on performance and efficiency (see Figures 8 and 9, Appendices 9.2 and 10.1).


These new experiments demonstrate that our approximation algorithm significantly reduces the time complexity while simultaneously enhancing the performances of powerful LLMs. This underscores the advantageous trade-off between computational efficiency and improved model effectiveness achieved by our approach.

For your convenience, all modifications in the updated PDF are highlighted in blue color.


[1] https://ai.meta.com/blog/meta-llama-3/

---

### Meta-Review · Area_Chair_z5iF · 2024-12-17

**Metareview:**

This paper proposes FEEDER to address examples selection for LLM in-context learning (ICL) and finetuning. The method is motivated by the hypothesis that a small set of highly informative examples can encapsulate enough information to infer many less informative examples. This is then used to define and select the sufficient and necessary samples for model prediction. The authors present a pre-selection framework for extracting essential samples from a training set, termed as FEEDER set, utilizing a tree-based merging mechanism, and propose a bi-level optimization approach for fine-tuning FEEDER and LLM training. All reviewers agreed the method is reasonable and addresses an important research problem. However, more than one reviewer found the experiments are not convincing enough, and a more detailed ablation study is required. At the end of the rebuttal, there were neither unanimous reviews nor strong recommendations for acceptance, and thus this paper can not be accepted by ICLR in its current version.

**Additional Comments On Reviewer Discussion:**

More than one reviewer found the experiments are not convincing enough, and a more detailed ablation study is required. The rebuttal did not fully address these concerns. At the end of the rebuttal, there were neither unanimous reviews nor strong recommendations for acceptance, and thus this paper can not be accepted by ICLR in its current version.

---

### Decision · Program_Chairs · 2025-01-22

Reject